# Potential and electric double-layer effect in electrocatalytic urea synthesis

Qian Wu [1], Chencheng Dai[1,2], Fanxu Meng [1], Yan Jiao [3] &
Zhichuan J. Xu [1,2,4,5] ✉

Electrochemical synthesis is a promising way for sustainable urea production, yet the exact mechanism has not been fully revealed. Herein, we explore the mechanism of electrochemical coupling of nitrite and carbon dioxide on Cu surfaces towards urea synthesis on the basis of a constant-potential method combined with an implicit solvent model. The working electrode potential, which has normally overlooked, is found influential on both the reaction mechanism and activity. The further computational study on the reaction pathways reveals that *CO-NH and *NH-CO-NH as the key intermediates. In addition, through the analysis of turnover frequencies under various potentials, pressures, and temperatures within a microkinetic model, we demonstrate that the activity increases with temperature, and the Cu(100) shows the highest efficiency towards urea synthesis among all three Cu surfaces. The electric double-layer capacitance also plays a key role in urea synthesis. Based on these findings, we propose two essential strategies to promote the efficiency of urea synthesis on Cu electrodes: increasing Cu(100) surface ratio and elevating the reaction temperature.

Urea ($CO(NH_2)_2$) is a highly valuable nitrogen fertilizer supporting approximately 27% of the world's population[1,2]. The traditional urea industry is accomplished through the reaction of ammonia ($NH_3$) and carbon dioxide ($CO_2$) operating under harsh conditions (150–200 °C,150–250 bar)[3]. This indirect method consumes approximately 80% of produced $NH_3$. Thereinto, the Haber-Bosch process is the predominant approach for industrial $NH_3$ synthesis, which alone accounts for approximately 2% of global energy consumption and releases vast amounts of green-house gas[4,5]. Therefore, great effort has been made to develop greener routes for urea synthesis.

Electrochemical urea synthesis via the direct coupling of $CO_2$ and $N_2$ under mild conditions has recently emerged as a promising alternative to conventional synthesis methods[6–9]. Nonetheless, substantial input energy is required to dissociate the inert N≡N triple bond (the bonding energy is 940.95 kj mol$^{-1}$)[10–12]. Further studies rooted in

electrocatalytic coupling of $CO_2$ with nitrogen oxides provide a more intriguing picture for direct electrocatalytic urea synthesis. In view of the lower bonding energy of N=O (204 kj mol$^{-1}$), this direct urea synthesis method exhibits potential of higher current efficiency with respect to the coupling of $CO_2$ with $N_2$[13,14]. In particular, it can bring huge economic and environmental benefits at an industrial level, as the reactants for electrocatalytic urea synthesis are cheap and environmentally unfriendly (for instance, the greenhouse gas $CO_2$ captured from point sources is priced at US$40–60 per metric ton[15,16] and nitrogen oxides are sourced from the pollutants in industrial wastewater[17]) and the price of urea is extremely high (US$650–1000 per metric ton for urea[18,19] (see the latest FOB international fertilizer prices)). Additionally, the unnecessity for complicated high-temperature-high-pressure equipment and inherent nature of the electrolyser allows the decentralized on-demand urea production,

[1]School of Material Science and Engineering, Nanyang Technological University, 50 Nanyang Avenue, Singapore 639798, Singapore. [2]The Cambridge Centre for Advanced Research and Education in Singapore, 1 CREATE way, Singapore 138602, Singapore. [3]School of Chemical Engineering, The University of Adelaide, Adelaide, SA 5005, Australia. [4]Energy Research Institute @ Nanyang Technological University, ERI@N, Interdisciplinary Graduate School, Nanyang Technological University, 50 Nanyang Avenue, Singapore 639798, Singapore. [5]Center for Advanced Catalysis Science and Technology, Nanyang Technological University, 50 Nanyang Avenue, Singapore 639798, Singapore. ✉ e-mail: xuzc@ntu.edu.sg

rendering this process economically and environmentally more attractive. Compared with the indirect method, such direct urea synthesis is significantly reforming the urea industry.

Despite the fundamental importance and huge interest, understanding the mechanism is still challenging in electrochemical coupling of $CO_2$ with nitrogen oxides (such as $NO_2^-$ in wastewater) toward urea after decades studies. There are two major questions to be addressed. One is the potential effect. Nowadays, the evaluation of the reaction mechanism and activity in electrochemical urea synthesis is mainly focused on "constant charge condition" calculations[20,21]. The relevant studies on reaction pathway selectivity and key intermediates do not consider the potential effect either, although the potential is experimentally demonstrated to be indispensable and vital in realistic catalytic reactions[22,23]. Another question is about the modulation of reaction mechanism. Up to now, theoretical principles for effectively modulating the catalytic performance in electrochemical urea synthesis remains largely unexplored[24–28]. This knowledge gap severely hampers the progress in this field, particularly given the complex nature of the urea synthesis process where the C-N coupling mechanism and key intermediates remain elusive[23,29]. The lack of consensus on modulating the urea synthesis reaction mechanism can extensively be attributed to the multitude of factors associated with the catalytic activity, which include but are not limited to applied potentials, reaction temperature, pressure, electrocatalysts, and electric double-layer[19,30–32]. Therefore, a comprehensive atomic-level understanding of the overlooked potential effect and the performance regulation factors for urea synthesis are challenging but essential.

Herein, we demonstrate the fundamental mechanism of electrochemical $NO_2^-$ and $CO_2$ coupling toward urea on various Cu surfaces under the constant-potential method combined with the implicit solvent model. Our calculations show that the previously overlooked potential is particularly important in determining the reaction mechanism and activity. The results also identify *CO-NH and *NH-CO-NH as the key intermediates in the urea formation. By analysing turnover frequencies under various potentials, pressures, and temperatures within a microkinetic model, the activity exhibits dependency on temperature and surface type. It is also worth mentioning that the capacitance of the electric double-layer plays a key role in the kinetic barrier for rate determining step. In light of these insights, we propose two strategies to promote the efficiency of urea synthesis on Cu electrodes: increasing (100) surface ratio and elevating the reaction temperature.

## Results

### Active sites and reaction mechanism for $NO_2^-$RR and $CO_2$RR

Cu is a popular electrocatalyst for the electro-reduction of carbon dioxide and nitrite ions to urea[23–28]. Therefore, we select Cu(111), Cu(110), and Cu(100) low-index single-crystal slabs as model electrocatalysts to investigate the electrochemical urea synthesis mechanism. Cu(111) and Cu(100) slabs have terrace surfaces, while the Cu(110) slab has a stepped surface (Fig. 1a). The entire reaction mechanism for electrochemical $NO_2^-$ and $CO_2$ coupling to urea can be divided into four stages: reduction of $NO_2^-$ and $CO_2$, the first C-N bond formation, the second C-N bond formation, and the final hydrogenation to urea (Fig. 1b). Among them, $NO_2^-$ ($NO_2^-$RR) and $CO_2$ ($CO_2$RR) reduction reactions are crucial competing reactions and the coupling reaction of corresponding N- and C-intermediates plays a pivotal role in urea synthesis. Consequently, examining the mechanisms of $NO_2^-$RR and $CO_2$RR on these surfaces is of utmost importance for understanding the urea synthesis process.

Previous research has already established the most favorable pathways for $CO_2$RR and $NO_2^-$RR on Cu electrodes. For $CO_2$RR, the pathway is $CO_2 \rightarrow$ *COOH $\rightarrow$ *CO[33,34], and for $NO_2^-$RR, it is $NO_2^- \rightarrow$ *$NO_2 \rightarrow$ *$HNO_2 \rightarrow$ *NO $\rightarrow$ *NOH $\rightarrow$ *N $\rightarrow$ *NH $\rightarrow$ *$NH_2 \rightarrow$ *$NH_3$[35,36]. In this work, the corresponding potential adsorption configurations of all N- and C-intermediates on various active sites of the three surfaces are investigated (Fig. 1a), with the most probable adsorption configurations and sites of intermediates on three surfaces depicted in Supplementary Figs. 1 and 2. Electrochemical reactions are widely recognized to be controlled by both kinetics and thermodynamics. Regardless of whether the reactions are of Type 1 or Type 2 (Fig. 1d), the energy difference between the transition state (TS) and the initial state (IS) is more positive than that between the final state (FS) and IS. Therefore, our study primarily concentrates on the kinetic process, which serves as the major rate-determining factor for electrochemical reactions[20].

The proton-coupled electron transfer (PCET) step for $NO_2^-$RR and $CO_2$RR can be achieved through either the Eley−Rideal (E-R; H atom from water) or the Langmuir-Hinshelwood (L-H; H atom from surface bonding) mechanisms[6], exemplified by the hydrogenation of *$NO_2$ in Fig. 1c. To accurately simulate hydrogen bonding of an H atom from a

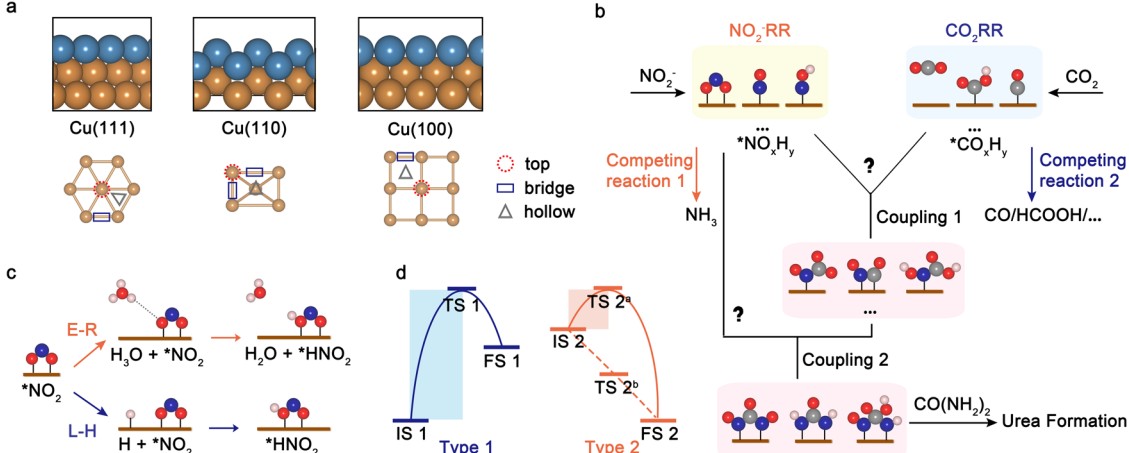

**Fig. 1 | Active sites and reaction mechanism for urea synthesis. a** Side view of the model slabs (the blue atoms represent surface atoms), and the potential active sites on the surfaces. **b** Schematic illustration of the electrochemical reaction mechanism for urea synthesis. Pink, red, brown, gray, and blue balls represent hydrogen, oxygen, copper, carbon, and nitrogen atoms, respectively. **c** Schematic diagram of Eley−Rideal (E-R) and Langmuir-Hinshelwood (L-H) mechanisms for the proton-coupled electron transfer (PCET) step. **d** Schematic illustration of kinetic barriers for electrochemical reactions. IS, TS and FS represent the initial, transition, and final states. For Type 2 reaction, there are two possibilities. The reaction may proceed spontaneously without crossing a kinetic barrier (TS $2^b$), or require a kinetic barrier as same as Type 1 (TS $2^a$).

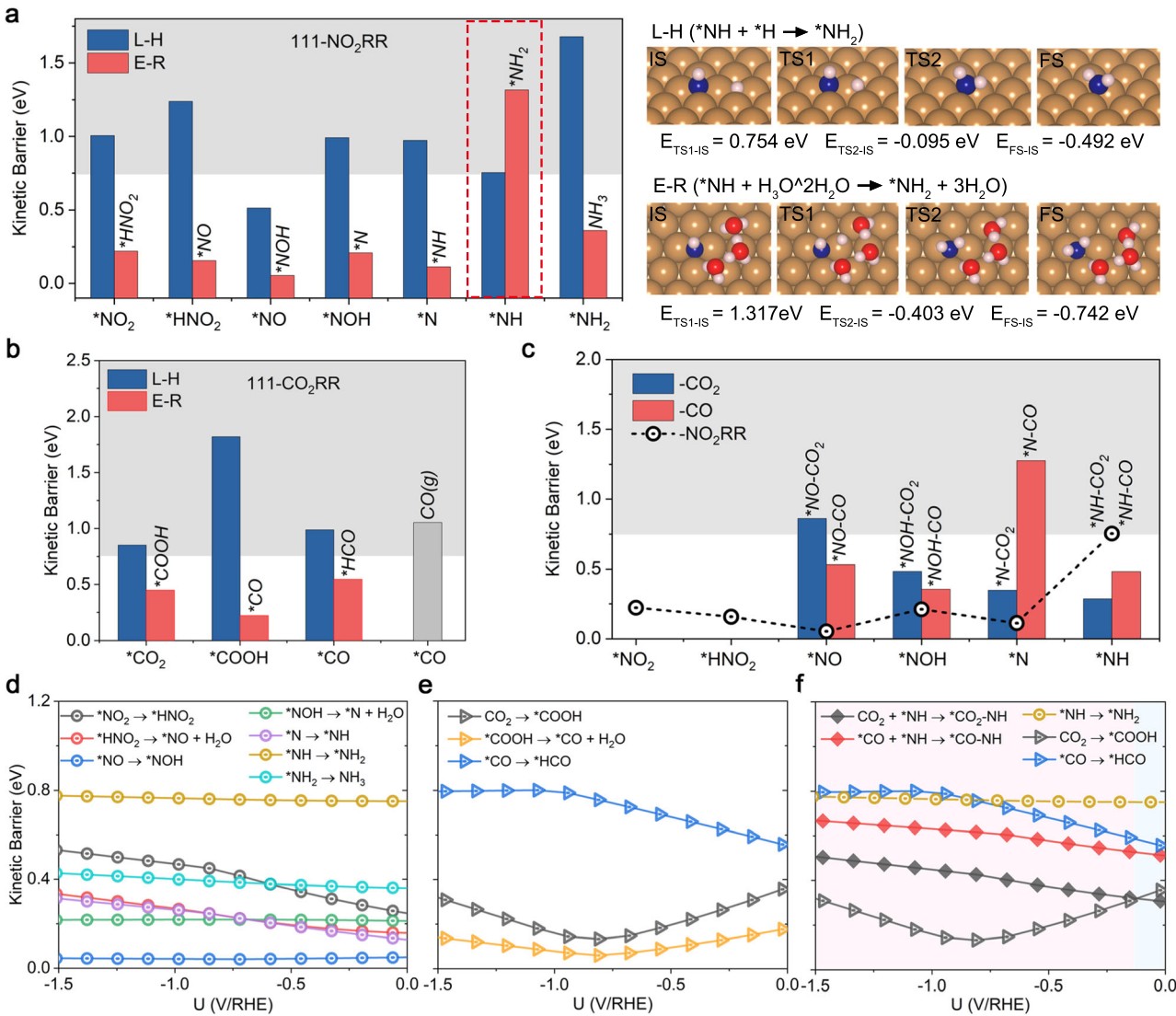

**Fig. 2 | Kinetic evaluation of NO$_2^-$RR, CO$_2$RR, and the first C-N coupling step.** A comparison of the kinetic barriers via L-H and E-R mechanisms for (**a**) NO$_2^-$RR and (**b**) CO$_2$RR on the Cu(111) surface. Right panel of (**a**) are snapshots of the kinetic process for the *NH hydrogenation step via L-H (blue bars in **a**) and E-R (pink bars in **a**) mechanisms. **c** Comparison of the kinetic barriers for the first C-N coupling and NO$_2^-$RR hydrogenation steps. Blue/Red bars stand for the kinetic barrier of CO$_2$/*CO coupling steps, and the black dashed line stands for the kinetic barrier of NO$_2^-$RR hydrogenation steps via the favorable mechanism. Gray shadows of (**a**)–(**c**) indicate

kinetically infeasible energy of 0.75 eV at room temperature (~300 K). Kinetic barriers for (**d**) NO$_2^-$RR and (**e**) CO$_2$RR hydrogenation steps on the Cu(111) surface as a function of the applied electrode potential vs reversible hydrogen electrode (U/RHE from 0 to −1.5 V). The pH is set as 6.8 for NO$_2^-$RR and 8.3 for CO$_2$RR and urea synthesis in accordance with the experimental environments[13]. **f** Kinetic barriers for CO$_2$ + *NH and *CO + *NH coupling steps as a function of the U/RHE and compared to *NH, CO$_2$, and *CO hydrogenation steps on the Cu(111) surface. Pink and blue shadows indicate the *CO-NH and *CO$_2$-NH coupling steps, respectively.

solvating water molecule within the E-R mechanism, we incorporate two explicit H$_2$O molecules and one H$_3$O$^+$ molecule[37-40], in conjunction with an implicit solvent. The calculated kinetic barriers of PCET steps for NO$_2^-$RR and CO$_2$RR via E-R and L-H mechanisms on three surfaces are summarized in Fig. 2a, b and Supplementary Fig. 3. For NO$_2^-$RR, the results reveal that on the Cu(111) surface, nearly all PECT steps favor the E-R mechanism except for the hydrogenation of *NH, which serves as the rate-determining step (RDS) for NO$_2^-$RR with a high kinetic barrier of 0.754 eV, as exemplified in Fig. 2a. On Cu(110) and Cu(100) surfaces, the majority of PECT steps also prefer the E-R mechanism, with the exception of *N hydrogenation during NO$_2^-$RR. The RDS for NO$_2^-$RR is the hydrogenation of *NH$_2$ and *NH on Cu(110) and Cu(100) surfaces with kinetic barriers of 0.346 and 1.082 eV, respectively. To pinpoint potential coupling steps, a kinetic barrier larger than 0.75 eV is set as a criterion for a fast-electrochemical process, and consider that reactions with kinetic barriers higher than 0.75 eV are kinetically unfeasible[21,41]. Therefore, the *NH$_2$ on Cu(111) and Cu(100) surfaces is

not included for the next C-N coupling step, as the formation of *NH$_2$ is hindered by a high kinetic barrier (>0.75 eV).

For CO$_2$RR, all PECT steps favor the E-R mechanism except for the hydrogenation of CO$_2$ on the Cu(110) surface. It should be noted that the *CO can be desorbed to CO(g) or further hydrogenated to *HCO experimentally[33,34]. Therefore, knowing whether *CO can be desorbed and hydrogenated on three surfaces is important for the subsequent C-N coupling step. It indicates that the hydrogenation of *CO to *HCO prefers the E-R mechanism on these surfaces, while the desorption of CO(g) is difficult to proceed due to the high kinetic barrier (>0.75 eV) (Fig. 2b and Supplementary Fig. 3). The following discussions is focused on the reaction pathway of CO$_2$ → *COOH → *CO → *HCO for CO$_2$RR.

Considering the limitations of the simple model of the computational hydrogen electrode (CHE) method under constant charge conditions[42-44], the constant-potential method to simulate potential-dependent kinetic barriers under experimental constant potential

conditions is utilized instead. The calculated total energies of IS and TS via the most favorable mechanism for $NO_2^-RR$ and $CO_2RR$ as a quadratic function of U/RHE on three charged surfaces are presented in Supplementary Figs. 4–9, and the fitted parameters are summarized in Supplementary Tables 1–6. Based on these results, kinetic barriers for electrochemical reaction steps of $NO_2^-RR$ and $CO_2RR$ with respect to the U/RHE are calculated (Fig. 2d, e and Supplementary Figs. 8–12). For $NO_2^-RR$, it can be found that kinetic barriers are influenced by the electrode potential, especially for the hydrogenation of $*NO_2/*HNO_2/*N$ on Cu(111), $*HNO_2/*NO/*NOH$ on Cu(110), and $*NO_2/*HNO_2/*N$ on Cu(100). This reveals the essential role of electrode potential on the $NO_2^-RR$ activity. As the electrode potential changes from zero to negative, kinetic barriers for $NO_2^-RR$ generally become more positive on the Cu(111) surface, suggesting $NO_2^-RR$ more sluggish at a more negative potential. In contrast, kinetic barriers for $NO_2^-RR$ are mostly more negative on Cu(110) and Cu(100) surfaces, indicating that $NO_2^-RR$ activity will increase as the electrode potential increases.

For $CO_2RR$, the kinetic barrier for the hydrogenation step of $CO_2$ to $*COOH$ is more positive than that of $*COOH$ to $*CO$ over the entire potential range on Cu(111) and Cu(110) surfaces, which means that $*COOH$ will rapidly convert to $*CO$ without coupling with N-intermediates on these two surfaces (Fig. 2e and Supplementary Fig. 8). Furthermore, on Cu(111) and Cu(110) surfaces, the kinetic barrier for the hydrogenation step of $*CO$ to $*HCO$ is more positive than that of $CO_2$ to $*COOH$ and $*COOH$ to $*CO$, indicating that $CO_2$ and $*CO$ are more facile to participate in the C-N coupling step rather than the protonation. On the Cu(100) surface, the kinetic barriers for the hydrogenation of $CO_2$ to $*COOH$ and $*COOH$ to $*CO$ cross over the entire potential range (Supplementary Fig. 9), implying that $CO_2$, $*COOH$, and $*CO$ may all be involved in the C-N coupling. However, it should be noted that below a potential of −0.60 V, the kinetic barrier for the hydrogenation of $CO_2$ to $*COOH$ is higher than that of $*CO$ to $*HCO$ on the Cu(100) surface. Consequently, once $CO_2$ is hydrogenated to $*COOH$, the $*COOH$ will be effectively reduced to $*CO$ and then to $*HCO$, and $*COOH$ and $*CO$ will not participate in the C-N coupling step on the Cu(100) surface above the electrode potential of −0.60 V. Based on these considerations, the $CO_2/*CO$ coupling reaction on Cu(111) and Cu(110) surfaces, and $CO_2/*COOH/*CO$ coupling reaction on Cu(100) surface are then investigated in the following section.

In addition, the adsorption energies of reactants $NO_2$ and the corresponding competing species with respect to the applied potential are shown in Supplementary Fig. 13. The adsorption energy of $*NO_2$ is significantly influenced by the applied potential, followed by $*H$ and $*CO_2$. Throughout the entire potential range, $*NO_2$ exhibits the most negative adsorption energy, implying that the active site will be primarily occupied by $*NO_2$. The adsorption energy of $*NO_2$ under different coverages are shown in Supplementary Fig. 14, they display the similar trends, and $*NO_2$ exhibits the more negative adsorption energy at the lowest coverage. In the following discussions, we will focus on the lowest coverage of $*NO_2$.

## The first C-N coupling step

After disclosing the electrochemical behavior of $NO_2^-RR$ and $CO_2RR$, the feasibility of the C-N coupling step on three Cu surfaces towards urea synthesis are needed to be assessed. For this purpose, kinetic barriers of potential C-N coupling steps and N-intermediates hydrogenation steps on three surfaces are compared (Fig. 2c and Supplementary Fig. 15). Figure 2c suggests the C-N coupling ($CO_2$ and $*NH/*CO$ and $*NH$ coupling step) more favorable than the hydrogenation step of $*NH$ to $*NH_2$ on the Cu(111) surface. The coupling of $CO_2$ with $*N$ is kinetically preferable to the hydrogenation step of $*N$ to $*NH$ on the Cu(110) surface (Supplementary Fig. 15). For the Cu(100) surface, the coupling of $*CO$ with $*N$ is more favorable than the hydrogenation step of $*N$, and the coupling of $CO_2/*COOH/*CO$ with $*NH$ is more favorable

than the hydrogenation step of $*NH$ (Supplementary Fig. 15). Snapshots of reactive trajectories for potential first coupling steps on these three surfaces are displayed in Supplementary Fig. 16. The total energies of IS and TS states for the possible C-N coupling steps as a function of electrode potential are presented in Supplementary Figs. 17–19, with the fitted parameters of the parabolic functions summarized in Supplementary Tables 7–9.

Based on the calculation results, the derived kinetic barrier curves of $CO_2/*CO$ and $*NH$ coupling steps on various Cu surfaces as a function of applied electrode potential, and the corresponding hydrogenation step of $*NH$ and $CO_2/*CO$ are also shown for comparison: i) On the Cu (111) surface (Fig. 2f), two regions can be classified in this framework: between 0.00 and −0.15 V vs RHE, $CO_2$ first couples with $*NH$ due to the lower kinetic barrier of the C-N coupling compared to $*NH$ and $CO_2$ hydrogenation; below −0.15 V vs RHE, $CO_2$ preferentially hydrogenates to $*CO$ before coupling with $*NH$ and then form $*CO$-NH. ii) On the Cu(110) surface, $CO_2$ cannot efficiently couple with $*N$ under applied electrode potentials ranging from 0.00 to −1.50 V, as $CO_2$ tends to hydrogenate to $*CO$ rather than coupling with $*N$ (Supplementary Fig. 18). Meanwhile, the kinetic barrier of the $*CO$-N coupling step is higher than the hydrogenation of $*N$ to $*NH$ (Supplementary Fig. 15). Overall, C-N bond formation is challenging on the Cu(110) surface, which favors $CO_2RR$ and $NO_2^-RR$ over C-N coupling for urea synthesis. iii) On the Cu(100) surface, three C-N ($*N$) coupling pathways can suit different electrode potential ranges (Supplementary Fig. 20): in the region of 0.00 to −0.42 V vs RHE, $*CO$-N coupling is favored; in the region of −0.42 to −1.30 V vs RHE, $*CO_2$-N coupling is preferred; below −1.30 V vs RHE, no C-N coupling occurs, as $*N$ species prefers to hydrogenate to $*NH$ rather than undergo C-N coupling with $CO_2/*CO$. While for the C-N ($*NH$) coupling step on the Cu(100) surface, $*CO$-NH coupling can occur at −0.20 to −0.35 V vs RHE, and $*COOH$-NH coupling can occur at 0.00 to −0.20 V vs RHE. Given that $*CO$-N and $*CO$-NH coupling steps have overlapped potential intervals, $*CO$-N can be successively reduced to $*CO$-NH. As demonstrated in Supplementary Figs. 21 and 22, the protonation of $*CO$-N to $*CO$-NH is highly facile, with a kinetic barrier lower than 0.11 eV under 0.00 to −1.30 V. Therefore, the $*CO$-NH intermediate forms at electrode potential between 0 and −1.30 V on the Cu(100) surface. It can be concluded that the low kinetic barriers for N-intermediates hydrogenation (below 0.4 eV) on the Cu(110) surface block the C-N coupling reactions (Supplementary Fig. 11), while the high kinetic barriers for the hydrogenation step of $*NH$ on Cu(111) and Cu(100) surfaces provide a window for the C-N coupling reaction (Fig. 2d and Supplementary Fig. 12). Keeping these in mind, we recognize that a proper potential window is required to enable N- and C-intermediate coupling rather than undergoing further hydrogenation separately on the electrocatalyst surface.

## The second C-N coupling and final hydrogenation steps

Having established the feasibility of the first C-N intermediates on Cu(111) and Cu(100) surfaces during the urea production process, the second concern regarding urea production is whether the second C-N bond can be further formed under experimental conditions. To address this concern, the possibility of N-intermediates ($*NOH/*NH$ of Cu(111) and $*N/*NH$ of Cu(100) surface) coupling with previously identified C-N intermediates are explored. Here, N-intermediates of $*NO_2/*HNO_2/NO/*N$ on Cu(111) surface and $*NO_2/*HNO_2/NO/*NOH$ on Cu(100) surface are exluded due to their relatively low kinetic barrier for hydrogenation reactions, making them more likely to undergo hydrogenation rather than the second coupling reactions. For the second C-N coupling on the Cu(111) surface, our calculation results indicate that the $*CO_2$-NH could not couple with the second N-intermediates (Supplementary Fig. 23). Furthermore, the possibility of the $*CO_2$-NH protonation to $*CO$-NH is investigated. As shown in Supplementary Fig. 25, the protonation of $*CO_2$-NH to $*CO$-NH is not

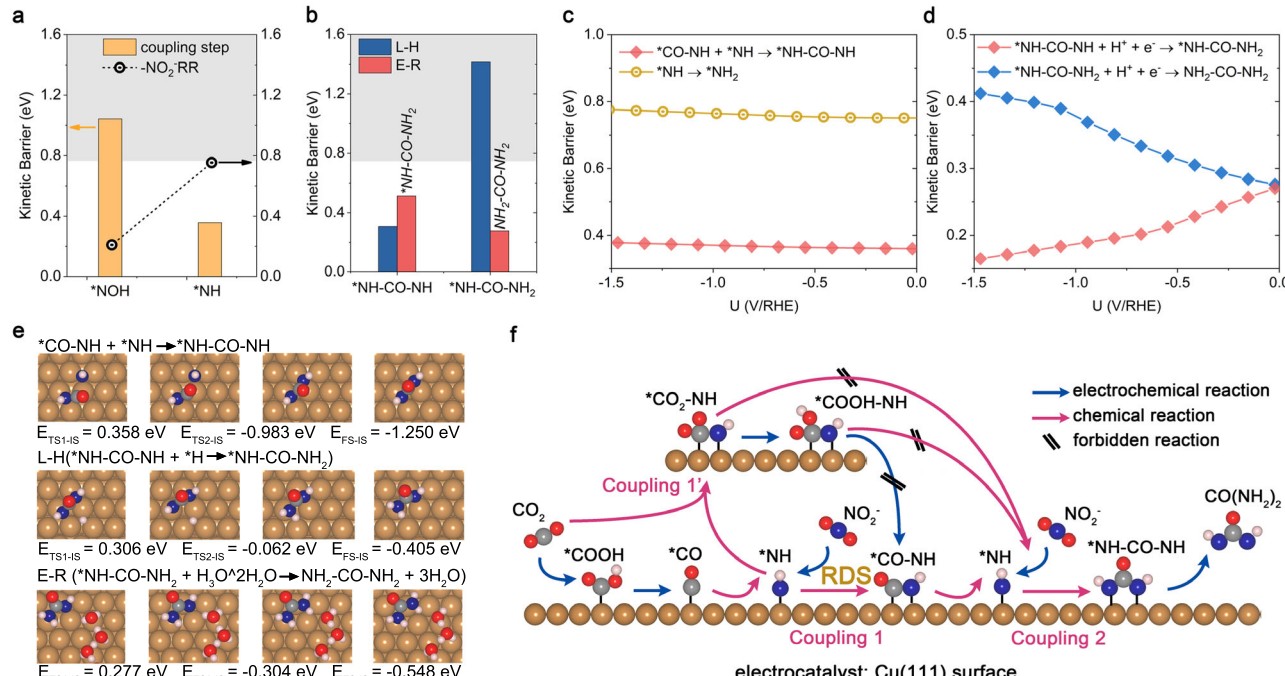

**Fig. 3 | Reaction mechanism for the second C-N coupling and final hydrogenation steps. a** Comparison of kinetic barriers for the second C-N coupling and the corresponding NO$_2^-$RR hydrogenation steps on the Cu(111) surface. **b** Comparison of the kinetic barriers via L-H and E-R mechanisms for the hydrogenation step of *NH-CO-NH and *NH-CO-NH$_2$ on the Cu(111) surface. **c** Kinetic barriers for *CO-NH and *NH coupling step as a function of the U/RHE, compared to the *NH hydrogenation step. **d** Kinetic barriers for *NH-CO-NH and *NH-CO-NH$_2$ hydrogenation steps as a function of the U/RHE. **e** Snapshots of the kinetic process for *CO-NH and *NH coupling and *NH-CO-NH and *NH-CO-NH$_2$ hydrogenation steps. **f** Schematic illustration of urea synthesis on the Cu(111) surface. The pH is set as 8.3 for urea synthesis in accordance with the experimental environments[13].

favorable due to the large kinetic barrier for *COOH-NH protonation to *CO-NH. The corresponding fitted parameters are summarized in Supplementary Table 10. Notably, *COOH-NH could not couple with the second N-intermediates either (Supplementary Fig. 23). On the Cu(111) surface, *CO-NH can only couple with *NH, presenting a relatively lower kinetic barrier of 0.357 eV as compared to the hydrogenation of *NH to *NH$_2$ (Fig. 3a). The energies of IS and TS and the corresponding kinetic barriers for the second C-N coupling reaction as functions of the applied electrode potential are shown in Supplementary Fig. 25 and Fig. 3c, with fitted parameters of the parabolic functions summarized in Supplementary Table 11. Obviously, the associated kinetic barrier for the *NH-CO-NH coupling is always lower than *NH hydrogenation, thus favoring coupling instead of protonation. Once *NH-CO-NH is formed, it will experience two hydrogenation steps to form urea (CO(NH$_2$)$_2$) with kinetic barriers of 0.306 eV and 0.278 eV via the L-H and E-R mechanisms, respectively (Fig. 3b), the corresponding configurations for the kinetic processes are presented in Fig. 3e. Moreover, the kinetic barriers for the subsequent two consecutive hydrogenation steps of *NH-CO-NH are lower than 0.45 eV among the applied electrode potential of 0.00 to −1.50 V (Fig. 3d), indicating the fast kinetics.

As for Cu(100) surface, *CO$_2$-N cannot directly couple with the second N-intermediates either. However, *CO$_2$-N can be rapidly hydrogenated to *COOH-N/*CO-N and subsequently to *CO-NH with low kinetic barriers within the applied electrode potential range of −0.42 to −1.30 V (Supplementary Figs. 21 and 22). While for *COOH-N and *CO-N, the kinetic barriers for the second C-N coupling are relatively high and less favorable than the hydrogenation of *N and *NH (kinetic barriers are larger than 1.00 eV), as displayed in Supplementary Fig. 26a. Therefore, *CO$_2$-N, *COOH-N, and *CO-N are not reactive toward urea synthesis. For *CO-NH, it could efficiently couple with *NH intermediates to form *NH-CO-NH, exhibiting a lower kinetic barrier than *NH protonation within the applied electrode potential range of

0.00 to −1.30 V (details can be seen in Supplementary Figs. 26a, c and 27). After that, the *NH-CO-NH could be further hydrogenated to *NH-CO-NH$_2$ and CO(NH$_2$)$_2$ with fast kinetics via the E-R mechanism (more details could be found in Supplementary Figs. 26 and 27). The fitted parameters of the parabolic functions are summarized in Supplementary Table 12.

From above, the reaction mechanism for urea synthesis is proved to be potential-dependent. On the Cu(111), it proceeds through the following mechanism under an applied electrode potential of −0.15 to −1.50 V, NO$_2^-$ → *NO$_2$ → *HNO$_2$ → *NO → *HNO → *N → *NH → *NH + *COOH → *NH + *CO → *CO-NH → *CO-NH + *NH → *NH-CO-NH → *NH-CO-NH$_2$ → CO(NH$_2$)$_2$. The RDS in this case is the coupling of *NH and *CO. For the Cu(100) surface, the reaction mechanism under the high applied electrode potential (0.00 to -0.42 V) follows NO$_2^-$ → *NO$_2$ → *HNO$_2$ → *NO → *HNO → *N → *N + *COOH → *N + *CO → *CO-N → *CO-NH → *CO-NH + *NH → *NH-CO-NH → *NH-CO-NH$_2$ → CO(NH$_2$)$_2$, with the *NH and *CO-NH coupling step as the RDS. Under the low applied electrode potential (-0.42 to -1.30 V), the mechanism is NO$_2^-$ → *NO$_2$ → *HNO$_2$ → *NO → *HNO → *N → *N + CO$_2$ → *CO$_2$-N → *COOH-N → *CO-N → *CO-NH → *CO-NH + *NH → *NH-CO-NH → *NH-CO-NH$_2$ → CO(NH$_2$)$_2$, where the *NO$_2$ hydrogenation step serves as the RDS. Hence, *CO-NH and *NH-CO-NH are the two key intermediates for urea synthesis. The corresponding schematics illustration for urea synthesis on Cu(111) and Cu(100) surfaces are depicted in Fig. 3f and Supplementary Fig. 26f, respectively.

## Microkinetic simulations
To further explore the selectivity and efficiency of urea production, microkinetic analysis is conducted to estimate the turnover frequencies (TOF) of urea and ammonia synthesis on Cu(111) and Cu(100) surfaces under experimental conditions. The microkinetic equations for urea and ammonia synthesis are summarized in Supplementary Tables 13, 15, 16, 18, and 20. The evolution of TOF for urea and

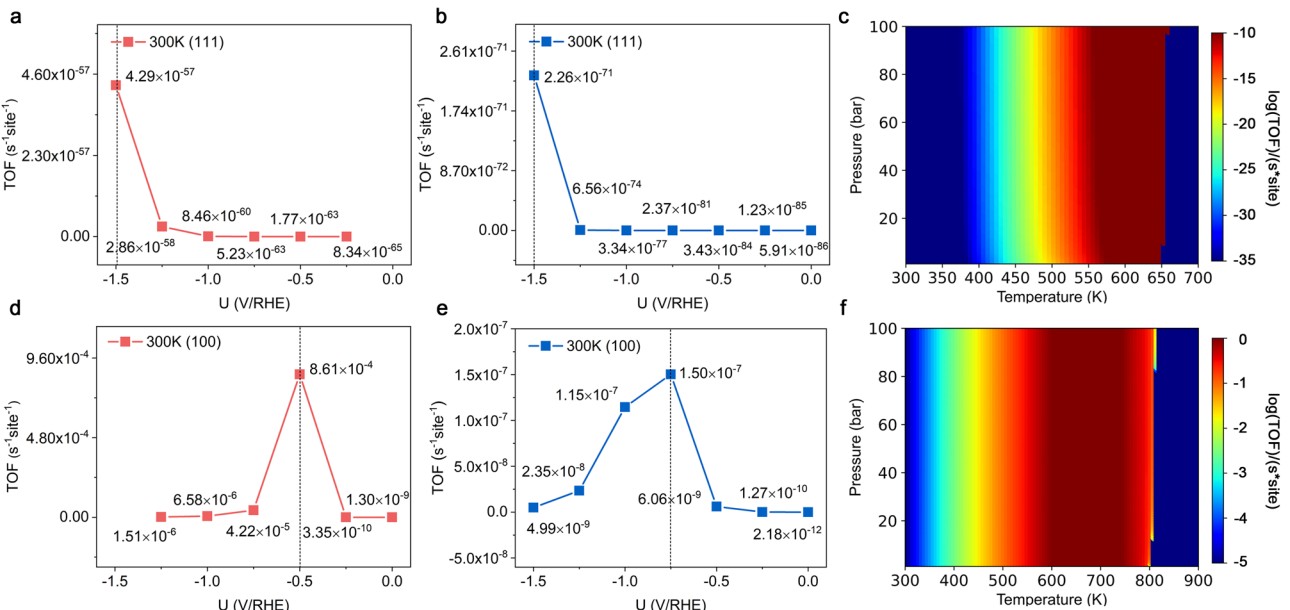

**Fig. 4 | Microkinetic simulations.** Turnover frequencies (TOF) per site for urea synthesis on (**a**) Cu(111) and (**d**) Cu(100) surfaces as functions of applied electrode potential vs RHE at 300 K and 1 bar. TOFs per site for NH₃ synthesis on (**b**) Cu(111) and (**e**) Cu(100) surfaces as a function of applied electrode potential vs RHE at 300 K and 1 bar. TOFs per site for urea synthesis on (**c**) Cu(111) and (**f**) Cu(100) surfaces mapped with pressure (1–100 bar) and temperature (300–700 K and 300–900 K, respectively).

ammonia synthesis on Cu (111) and Cu(100) surfaces as functions of applied electrode potential under mild conditions (300 K and 1 pa) is disclosed in Fig. 4a,d and Fig. 4b,e, respectively. Here, the applied electrode potentials ranging from 0.00 to −1.50 V are considered in steps of 0.25 V. It is worth noting that the applied electrode potential is a critical parameter for both urea and ammonia synthesis. The results suggest that Cu(100) surface is more active than the Cu(111) surface. The TOF for urea synthesis reaches its maximum value at −1.50 and −0.50 V vs RHE on Cu(111) and Cu(100) surfaces with values of $4.29 \times 10^{-57}$ s⁻¹site⁻¹ and $8.61 \times 10^{-4}$ s⁻¹site⁻¹ (Fig. 4a, d), respectively. Additionally, the TOF for ammonia synthesis reaches its maximum value at −1.50 and −0.75 V vs RHE on Cu(111) and Cu(100) surfaces with values of $2.26 \times 10^{-71}$ s⁻¹site⁻¹ and $1.50 \times 10^{-7}$ s⁻¹site⁻¹ (Fig. 4b, e), respectively. The TOF for CO₂RR on Cu(111) and Cu(100) surfaces are also shown in Supplementary Fig. 28. Within the applied potential range, both surfaces exhibit negative TOF values, signifying the reversibility of the reaction. The results corroborate our findings and are consistent with previous studies which suggest that *CO struggles with desorption and tends to further reactions on Cu surfaces[45,46]. Notably, the Cu(100) surface presents a more challenging environment for *CO desorption than the Cu(111) surface. Therefore, increasing the Cu(100) surface ratio is a strategy to enhance the efficiency of electro-catalysis on the Cu electrocatalyst for urea synthesis. To elucidate the huge differences in TOF on Cu(111) and Cu(100) surfaces during urea synthesis, the comprehensive degree of rate control (DRC, $X_i$) (Supplementary Tables 14 and 17) and coverage analysis (Supplementary Fig. 29) are conducted. A positive (negative) Xi value signifies that an increase in reaction rate (r) need to further stabilization (destabilization) of the corresponding surface state. On Cu(111), CO-NH coupling is the RDS (Supplementary Table 14); the surface is covered fully with *NH but with very little *CO (Supplementary Fig. 29). On Cu(100), NO₂-H hydrogenation is the RDS (Supplementary Table 17); the surface is covered fully with *NO₂ (Supplementary Fig. 29). Cu(111) struggles to get *CO, while Cu(100) gets H⁺ mainly from the electrolyte (E-R step). Thus, the urea synthesis reaction can proceed smoothly on Cu(100) surface, but is difficult on Cu(111) surface. Moreover, the *NH with $X_i = -1$ negatively impacts the r on Cu(111), while the surface is covered fully with *NH. On Cu(100), *NO₂ with $X_i = 1$ positively influences r and *NO₂ is the predominant surface-

covering species during the urea synthesis on Cu(100). Therefore, the r for urea synthesis is further weaken on Cu(111) surface and enhanced on Cu(100) surface. Consequently, the reason for large difference in urea TOFs between Cu(100) and Cu(111) is not kinetic barriers alone. It need to be attributed to the different RDS and the availability of reactants (*NH/*CO for Cu(111), *NO₂/H⁺ for Cu(100)).

Next, the reaction rate for urea synthesis on Cu(111) and Cu(100) surfaces under a pressure range of 1 to 100 bar and a temperature range of 300 to 1000 K (Fig. 4c, f) are calculated. With the increase in temperature, the TOF for urea synthesis on both Cu(111) and Cu(100) surfaces increases, while it remains unaltered by pressure variations. A pronounced decrease in TOF values for urea synthesis is observed, plummeting to nearly zero above 650 K on Cu(111) surfaces and 900 K on Cu(100) surface. Consequently, we restrict our TOF display to the range of 300 to 700 K for Cu(111) and 300 to 900 K for Cu(100) surface. Temperature is a pivotal factor affecting the adsorption of intermediates, thereby affecting the progress of reactions. To explore the temperature effect on species adsorption, the coverage curves of adsorbed species at varying temperature are conducted. Supplementary Fig. 29 reveals the dominant adsorbed intermediates are *NH and *NO₂ on Cu(111) and Cu(100) surfaces within the aforementioned temperature ranges. This can be attributed to the RDS on the Cu(111) surface being the coupling of *CO and *NH, while it is the hydrogenation of *NO₂ on the Cu(100) surface. Above 650 and 900 K, other species dominate adsorption on Cu(111) and Cu(100) surfaces, respectively, leading to alterations in the reaction mechanism. The corresponding ab initio molecular dynamics (AIMD) simulations in Supplementary Fig. 30 also confirm the significant impact of temperature on the adsorption behavior of intermediates. Elevating the temperature too high might make reactants dissociation, potentially hindering the progress of the reaction. Based on the above microkinetic analysis of urea synthesis, another strategy to improve the efficiency of urea synthesis on the Cu is to appropriately increase the reaction temperature. Moreover, even though the calculated TOF for urea synthesis on the Cu(111) surface is enhanced at 1 bar and 600 K, it remains about five orders of magnitude lower than that of the Cu(100) surface at 1 bar and 300 K. Therefore, the Cu(100) surface should be the best among other surfaces for urea synthesis.

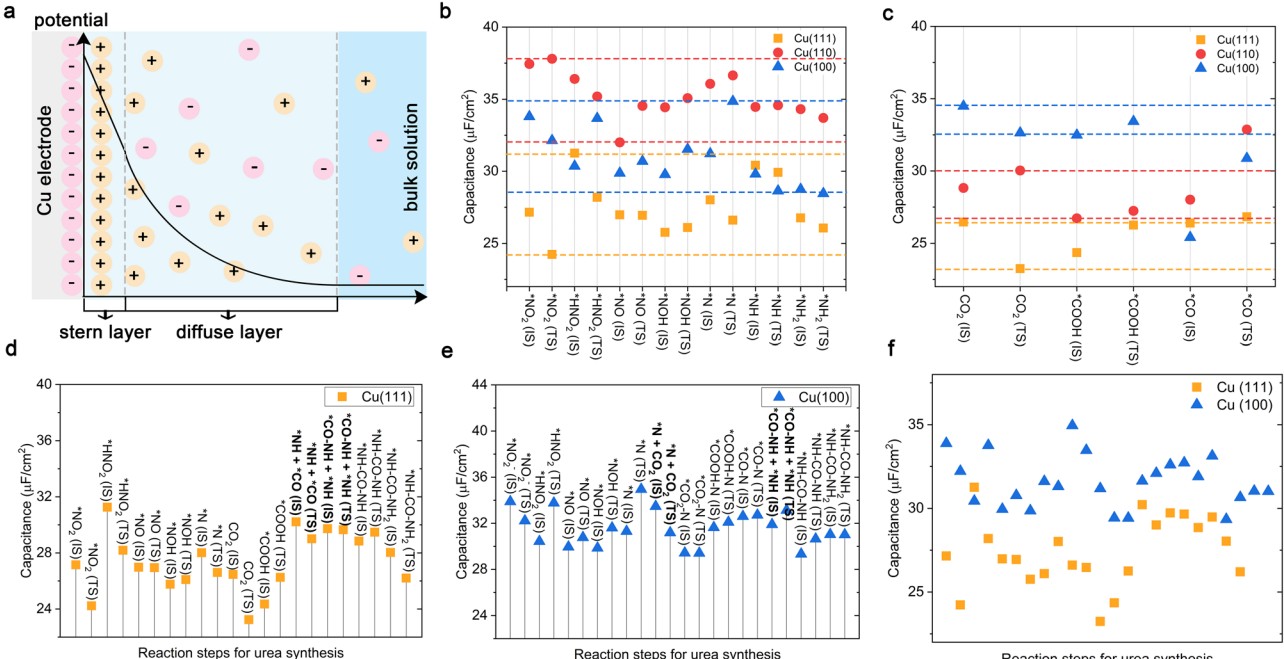

**Fig. 5 | Electric double-layer (EDL) capacitances. a** Schematic of the EDL on the negatively charged Cu electrode surface, with the corresponding potential distribution varying with distance from the electrode surface. Capacitances of the electrocatalyst surfaces (Cu(111), Cu(110), Cu(100)) with different intermediates during (**b**) $NO_2^-$RR and (**c**) $CO_2$RR processes in the EDL. IS and TS indicate the initial and transition states of intermediates during the PCET steps. The horizontal dotted lines in (**b**) and (**c**) represent the capacitance range. During the urea synthesis process, the EDL capacitances on (**d**) Cu(111) and (**e**) Cu(100) surfaces with different intermediates. Bold indicates the coupling reaction steps, and non-bold indicates the PCET steps. **f** Comparison the capacitances of Cu(111) and Cu(100) surfaces during the urea synthesis.

Additionally, the relationships between the TOF values on Cu(111)/Cu(100) surfaces and reactant concentrations are investigated (Supplementary Fig. 31): $NO_2^-$/$H^+$ concentration and $CO_2$ pressure. The TOF dependencies on reactants for Cu(111) and Cu(100) surfaces differ, a variation intrinsically linked to differences in reaction mechanisms. On the Cu(111) surface, the RDS is the coupling of *NH and *CO, making TOF values particularly sensitive to $NO_2^-$ concentration and $CO_2$ pressure. In contrast, the RDS on the Cu(100) surface is the hydrogenation of *$NO_2$, thus TOF values exhibit a strong correlation with $NO_2^-$ concentration and $H^+$ concentration, while $CO_2$ pressure has a minimal effect. Intriguingly, the C-intermediates display a positive $X_i$ value during urea synthesis on the Cu(111) surface (Supplementary Table 14), indicating that challenges in the CO-NH coupling on the Cu(111) surface stem primarily from the adsorption constraints of C-intermediate, which is well consistent with the results of Supplementary Fig. 31. The $NO_2^-$/$H^+$ concentration and $CO_2$ pressure impact of the adsorbate coverages on Cu(111) and Cu(100) surfaces are also accessed. A notable finding on the Cu(111) surface is the consistent adsorbate coverages within the pH range of 1 to 7 (acidic environment), while as pH transitions to 8 to 14 (alkaline environment), the coverages fluctuate with pH, as depicted in Supplementary Fig. 32. These signify that pH not only modulates the TOF but also alters the reaction mechanism on the Cu(111) surface. Interestingly, alterations in $NO_2^-$ concentration also will alter the adsorbate coverages on Cu(111) and Cu(100) surfaces as shown in Supplementary Fig. 33.

## Discussion

In electrochemical reactions, an electric double-layer (EDL) arises due to the interaction between ions in the electrolyte bulk solution and the charged surface of the electrode[47,48]. The EDL is comprised of two charged layers: the inner Stern layer and the outer diffuse layer (Fig. 5a). Generally, thinner EDL is preferred in experiments to yield a larger electric field intensity and larger capacitance, which facilitate charge migration and ion diffusion[47,49]. Therefore, it is of critical importance to understand the role of EDL in electrocatalysis, for the design and optimization of the urea synthesis process. A comprehensive analysis of EDL on Cu(111), Cu(110), and Cu(100) surfaces is conducted.

Figure 5b,c show the capacitances of Cu(111), Cu(110), Cu(100) surfaces with different intermediates during $NO_2^-$RR and $CO_2$RR processes in the EDL. The interaction of various intermediates with the electrode surface leads to different EDL capacitances. For $NO_2^-$RR, Cu(110) surface exhibits the highest capacitances (ranging from 32.003 to 37.803 $\mu F/cm^2$), followed by the Cu(100) surface (ranging from 28.454 to 34.854 $\mu F/cm^2$)), and finally the Cu(111) surface (ranging from 24.228 to 31.258 $\mu F/cm^2$)) (Fig. 5b and Supplementary Tables 1–3). For $CO_2$RR to *COOH and *CO, the capacitance order is Cu(100) (between 32.482 and 34.477 $\mu F/cm^2$) > Cu(110) (between 26.722 and 30.032 $\mu F/cm^2$) > Cu(111) surface (between 23.241 and 26.465 $\mu F/cm^2$) (Fig. 5c and Supplementary Table 4-6). The kinetics and favored reaction pathways of electrochemical reactions are primarily influenced by the structure of EDL and the intrinsic interactions between the electrode and the electrolyte[32]. Several fundamental interactions within the EDL, including electrostatic forces, covalent bonds, and non-covalent bonds, together shape the dynamic nature of the interfacial structure. Specifically, during the urea synthesis, covalent interactions involve direct bond formations, which encompass orbital overlaps and chemical interactions between the adsorbates, which are strongly dependent on the interfacial field, the specific nature of the electrode surface. Notably, the EDL capacitances for intermediates involved in the $NO_2^-$RR and $CO_2$RR are comparable on Cu(111) and Cu(100) surfaces. However, Cu(110) surface presents substantial difference in capacitances between the $NO_2^-$RR and $CO_2$RR processes. This suggests that the coupling behaviors of N/C-intermediates on Cu(111) and Cu(100) surfaces might be enhanced, owing to their resembling physicochemical attributes and interfacial behaviors. However, Cu(110) surface might hinder the interaction of N-intermediate with C-intermediate due to the distinctions in

capacitances. To better illustrate the physical origin of the relationship between EDL capacitance and electrokinetic, we studied the surface-charge density ($\sigma$) of intermediates adsorbed surfaces. $\sigma$ could be an appropriate descriptor for electrostatic effects of the double layer on electrokinetic, since it describes the variations of the interfacial field local to the reaction site[50,51].

Supplementary Fig. 34 presents the changes in $\sigma$ for surfaces with adsorbed C- and N-intermediates. The C- and N-intermediates adsorbed Cu(111) and Cu(100) surfaces are characterized by positive $\sigma$, showing identical interfacial fields. This uniformity indicates that C- and N-intermediates can coexist on these surfaces, thereby providing steric possibilities for the coupling steps. In contrast, C- and N-intermediates adsorbed Cu(110) surface shows opposite interfacial fields, which may hinder coupling steps. Especially, the negative $\sigma$ of *CO$_2$ adsorbed Cu(110) surface could be responsible for the decrease in EDL capacitance of C-intermediate adsorbed Cu(110) surface (Fig. 5). As a result, the interaction among electrocatalyst surface, intermediates, and electrolyte alters the reaction mechanism pathways, ultimately determining the preferred direction of the reaction toward specific products. This finding further illustrates the poor electrocatalytic performance for urea synthesis on Cu(110) surface. The calculated C$_{dl}$ values shown in Supplementary Tables 1-12 are closely aligning with experimentally reported values[52,53], which further validates our methodology for catalyst-electrolyte interface.

To get further insights into the influence of EDL, the capacitances change on Cu(111) and Cu(100) surfaces during the urea synthesis are summarized. As shown in Fig. 5d, e, the capacitances on Cu(111) and Cu(100) surfaces vary from 24.228 to 31.258 μF/cm$^2$ and 29.337 to 34.945 μF/cm$^2$, respectively. Notably, all intermediates adsorbed Cu(100) surface exhibit a larger capacitance than the Cu(111) surface (Fig. 5f). This variation can be attributed to alterations in $\sigma$ as the larger capacitances indicate more electrons on the electrode. As shown in Supplementary Fig. 35a–c, the $\sigma$ follows the order Cu(100) > Cu(111), which is consistent with the order of EDL capacitance (Fig. 5f). Supplementary Fig. 35 also show the $\sigma$-dependent kinetic barriers of RDS for urea synthesis on both Cu(111) and Cu(100) surfaces, there is an increase in the kinetic barrier as $\sigma$ decreases. By regulating the EDL with the larger capacitances, we can efficiently regulate the kinetic barrier of RDS. It should be noted that the capacitance of the EDL can be influenced by several factors and can be experimentally modulated through various approaches, such as the electrolyte concentration, electrolyte type, temperature, and pH[54]. Therefore, tuning the capacitance of the EDL could be a way to optimize the electrochemical synthesis of urea. However, it needs to be evaluated with other factors to comprehensively assess the impact on the reaction rate, due to the TOF is not influenced by the kinetic barriers alone as discussed above.

To summarize, the mechanism of urea synthesis on Cu electrode is investigated with the constant-potential method. The reaction mechanism and urea production activity are found strongly related to the previously overlooked potential. The *CO-NH and *NH-CO-NH are identified as the two key intermediates in urea synthesis. In addition, the calculations employing a microkinetic model reveals that the activity increases with temperature, and Cu(100) surface is the most effective one for urea synthesis. Moreover, the capacitance of EDL is proved to be critical for urea synthesis on Cu surfaces. It is worth noting that while the EDL capacitance is effective in regulating the kinetic barrier of RDS, its impact on the reaction rate of complex reactions should be evaluated in conjunction with other factors such as coverage. Based on these findings, we propose the design principles for promoting the efficiency of urea synthesis, i.e., increasing (100) surface ratio and elevating the reaction temperature. This study offers a unique and foundational insight into electrochemical urea synthesis. The techniques employed can be further applied to gain essential understanding and catalyst designs for various electrochemical reactions.

## Methods

### DFT computations

We perform spin-polarized density functional theory (DFT) within the Vienna ab initio simulation package (VASP) to calculate the reaction energies of single crystal Cu[55]. The ion-electron interaction is described using the projector-augmented plane-wave[56] and the exchange-correlation interaction is described using the Perdew–Burke–Ernzerhof functional of the generalized gradient approximation[57]. A $4 \times 4 \times 1$ unit cell of Cu(111) and Cu(100) slabs with three layers (bottom layer fixed) and a $4 \times 3 \times 1$ unit cell of Cu(110) slab with four layers (bottom two layers fixed) are constructed as substrate electrocatalysts and $3 \times 3 \times 1$ Monkhorst−Pack k-point grids are used. The vacuum layer larger than 15 Å is implemented to prevent the interaction between periodical slabs. Grimme's D3 method is adopted to consider the van der Waals (vdW) interactions[58]. The cut-off energy is set as 500 eV, and all the systems are optimized until energy and force were less than $10^{-5}$ eV and 0.01 eV/Å. To locate transition states and kinetic barriers, we use the climbing-image nudged elastic band (CINEB) method[59], with the force convergence criterion of 0.1 eV/Å. And the computational results are post-processed by the VASPKIT code[60]. The free energy changes for the electrochemical urea synthesis steps are following the computational hydrogen electrode (CHE) model developed by Nørskov et al.[61].

### Constant potential method

To more realistically simulate the influence of the experimental reaction condition on the intrinsic catalytic activity of single-crystal Cu slabs, we adopt the constant-potential method[62]. Different from the original implementation of the double-reference method[63], which employs explicit water molecules to represent the metal/aqueous interface, the aqueous environment is modeled as a continuum dielectric by the VASPsol code with a relative permittivity of 80[64,65]. The effective surface tension parameter is set to 0 in VASPsol to neglect the cavitation energy contribution. The linear Poisson-Boltzmann model with a Debye screening length of 3.0 Å mimics the compensating charge, allowing for a more realistic description of the EDL. Modeling the charged species transport in the electrolyte is important while complex due to the interactions between the charged species and the local environment at the electrode-electrolyte interface. A fully explicit description of the electrolyte is needed to provide the most accurate description of the transporting properties. Such a method would involve complex AIMD simulations incorporating both water molecules and ions explicitly[66–69]. However, complex gradients study is more expensive and not affordable for a large system, particularly when exploring complex reaction mechanisms like urea synthesis in our study, it exceeds the capabilities of our current computational resources. The charges for each system are added from −1.5 e to +1.5 e in steps of 0.5 e to clarify the electrode potential function. The potential-dependent energy of the slabs can be calculated as[63,70]

$$E = E_{scf} + E_{corr} - q\varphi_q \qquad (1)$$

where $E_{scf}$ is the self-consistent energy of DFT calculations, $E_{corr}$ is the correction energy of the homogeneous background charge, $q$ is the added number of electrons, and $-\varphi_q$ is the work function of the charged slab. $E_{corr}$ is the correction energy of background charge and is obtained by the average electrostatic potential of the systems $<\overline{V_{tot}}>$

$$E_{corr} = \int_0^q \langle \overline{V}_{tot} \rangle dQ \qquad (2)$$

The electrode potential ($U_q$) of the charged systems referenced to the standard hydrogen electrode (SHE) is calculated as

$$U_q = -4.6 - \varphi_q/e\text{V} \qquad (3)$$

where 4.6 V is the absolute potential relative to the SHE benchmarked in the VASPsol[71]. The energy varies with the electrode potential as a quadratic function, which takes the form

$$E(U_q) = -\frac{1}{2}C(U_q - U_0)^2 + E_0 \qquad (4)$$

where $C$ is the capacitance of the system, $U_0$ is the potential of zero charges (PZC), and $E_0$ is the energy at the zero charges.

The pH can also affect the energies of the systems. Upon changing the pH value, the electrode potential under the SHE will change to a fixed potential of reversible hydrogen electrode (RHE) as

$$U_{RHE} = U_{SHE} + k_b T ln(10)pH/e \qquad (5)$$

$k_b$ is the Boltzmann constant. The pH is set as 6.8 for $NO_2^-$RR and 8.3 for $CO_2$RR and urea synthesis in accordance with the experimental environments[13].

## Microkinetic model

Microkinetic simulation is employed to estimate the reaction rate of urea and ammonia synthesis and the $CO_2$RR on single-crystal Cu surfaces under the quasi-equilibrium approximation[72,73]. That is all the reaction steps are in equilibrium states except for the RDS to identify the coverages of the intermediate species. The equilibrium constant ($K_i$), forward ($k_i$), and backward ($k_{-i}$) rates constants are calculated by the Arrhenius equation[74,75]:

$$K_i = \frac{-\Delta G_i}{e^{k_b T}} \qquad (6)$$

$$k_i = \frac{k_b T}{h} e^{\frac{-\Delta G_{TS}}{k_b T}} \qquad (7)$$

$$K_i = \frac{k_i}{k_{-i}} \qquad (8)$$

where $\Delta G_i$, $\Delta G_{TS}$, $k_b$, $T$, and $h$ are the free energy change between the final and initial state, the kinetic barrier calculated by CI-NEB, Boltzmann constant, temperature, and Planck constant, respectively. The TOF is obtained as the reaction rate of RDS.

The reaction rate of an elementary step $r_j$ is computed as

$$r_j = k_j \prod_i \theta_i^{v_i^j} \qquad (9)$$

where $\theta_i$ is the coverage of species i, and $v_i^j$ is the stoichiometry of species i in the elementary step j. The coverage of all the reaction species equal to one.

In this work, the concentration of solvated $CO(NH_2)_2$ and $NO_2^-$ are set at $1 \times 10^{-4}$ M and 0.1 M, respectively[39]. $C_{H2O}$ denotes the bulk concentration of $H_2O$ and equals to 1[38]. The concentration of $H^+$ is calculated by: $C_{H+} = 10^{-pH}$[76].

The generalized DRC ($X_i$) is used to distinguish the relative importance for each species i[77,78],

$$X_i = \left( \frac{-\partial lnr}{\partial \left( \frac{G_i^0}{k_b T} \right)} \right)_{G_j^0 \neq i} \qquad (10)$$

where $r$ is the net reaction rate to the product of interest, $G_i$ is the free energy change of each intermediate and transition state. The partial derivative is now taken holding constant the free energy change of all other species (intermediates and transition states), j.

## Data availability

The data supporting the findings of this study are available within the article and its Supplementary Information. Additional data are available from the corresponding author upon request.

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

## Acknowledgements

This work is supported by Singapore Ministry of Education Tier 1 Grant (RG78/22) and A*STAR (Agency for Science, Technology and Research) under its LCERFI program (Award No. U2102d2002).

## Author contributions

Q.W. conceived this research, contributed calculations, analysis of data and writing.; Z.J.X. acquired research and computational resources; C.D. and F.M. assisted with the analysis of results from experimental perspectives; Z.J.X., Y.J., Q.W. and C.D. reviewed and revised the manuscript. All authors contributed towards discussion on the manuscript.

## Competing interests

The authors declare no competing interests.
