## [Peer Review File · Nature Communications]

REVIEWER COMMENTS

Reviewer #1 (Remarks to the Author):

Review attached

Reviewer #2 (Remarks to the Author):

Reviewer #3 (Remarks to the Author):

The authors present a computational study of urea synthesis on three copper surfaces. Their results show that Cu(100) presents the most favorable energetics for urea synthesis of the models considered. The authors very nicely and thoroughly analyze the system, with a well-rationalized mechanism proposed. Although this work is very thorough and careful, I believe there is a significant flaw in the analysis, along with some substantial limitations to the microkinetic modeling approach; these items limit my ability to fully interpret the conclusions. I believe that as a result a major revision is needed before considering its publication.

Most significant: when calculating activation energies of E-R steps, what is the validity of using uncharged H₃O as a hydrogen source? If this species is not charged, it creates an artificial driving force for H transfer and thereby will artificially lower the barriers for H transfer. This is true for all the considered E-R steps, and likely is a reason why the barriers for E-R steps are significantly lower than those for L-H steps. More physically representative would be, e.g., a mechanism by which transfer of an electron is made to the surface-bound species, followed by transfer of a proton (not charge-neutral H). I fear that this has substantial impacts on the calculated reaction energetics and will affect the conclusions of this work, as I'm unable to ascertain what the effect on the mechanisms may be.

Microkinetic modeling plays an important role in this work. I commend the authors for applying microkinetic modeling in a complex environment, though the limitations of microkinetic modeling in complex electrochemical environments are significant and will affect the quantitative conclusions, if not the qualitative conclusions. Among the limitations that I believe need to be more carefully considered are (i) adsorbate interactions at higher coverages, including with applied potential; (ii) adsorption of charged species like NO₂⁻ with respect to potential and quantifying their coverage/competitive adsorption, and (in general) determination of appropriate chemical potential references in the liquid phase; and (iii) transport of species at the interface, which is more complex than in gas-phase contexts. The associated impacts need to be discussed, with the results appropriately placed in the context of these limitations. For example, how might the conclusions with respect to temperature effects be affected when considering that higher temperature will further alter the adsorption of species (in the context of the above points)?

Relatively minor points:

In Figure 1d, it isn't clear what is being conveyed by the "a" and "b" versions of TS 2 in type 2. This should be clarified for readers.

It is standard to provide the calculated energetics of adsorbed species in a supplementary table.

The authors provide a nice introduction as to how urea synthesis from NO_x and CO₂. There seems to be a potential mismatch in scale relative to urea production from N₂, in that substantial sources of NO_x species would be required. Is industrial wastewater capture sufficient relative to the overall scale of urea production? The authors do not need to perform a techno-economic analysis, but a comment related to this aspect would be appreciated to better contextualize the nice discussion they already provide.

The authors may wish to correct the spelling of "Eley-Rideal".

Review

In this work, Wu et al. used a “double-reference method” with DFT to study the potential-dependent mechanism of electrocatalytic C-N coupling to produce urea from CO₂ and NO₂⁻. The authors identified and addressed all the important and interesting questions on this topic. However, I find their methodology lacks justifications for some important model choices, giving the impression that the “double-reference” method was employed without being closely examined. The section about reaction mechanisms (lines 95 to 306) proves that the authors investigated this complex reaction in a detail-oriented and physically reasonable manner. Yet, the later sections (below line 308) about the microkinetic model and electric double-layer capacitance are rather unsatisfactory with many questionable physical interpretations of their results. I, ultimately, recommend not publishing this work in its current state.

Positives and takeaways:

- Credits are due for studying a complex reaction mechanism at varying electrode potentials in careful detail.
- The authors clearly elucidated abundant C-derived and N-derived species (prior to C-N coupling) on each Cu facet at different potentials (0.0 V to -1.5 V RHE).
- They identified key C-derived and N-derived intermediates whose first coupling step is more favorable or competitive to the non-coupling reduction paths.
 - For Cu(111), it is either *CO+*NH or *CO₂+*NH (depending on the applied potential).
 - For Cu(110), non-coupling reduction paths are preferred.
 - For Cu(100), multiple coupling pairs can be feasible depending on the applied potential (perhaps the most significant finding of this manuscript).
- They identified intermediates for the second C-N coupling step, which are *CO-NH + *NH on both Cu(111) and Cu(100) (while ruling out *CO₂-NH and *COOH-NH).

Negatives/needs improvement:

Major

- “Not-double-reference” DFT:
 - The term “double-reference” in this manuscript is a misnomer. The original method by Neurock (Phys. Rev. B 73, 165402, 2006; citation 61 in this manuscript) entails referencing the electrode work function (i.e., the absolute electrode potential) to the potential of a static explicit water layer and the potential of a cleaved vacuum region (hence, double reference). The method that the authors herein employed from the Henkelman paper (ACS Catal., 9, 5567-5573, 2019; citation 58 in this manuscript) instead references the electrode work function only to a continuum slab (no explicit water) and hence, is not “double-reference”. The Henkelman paper, in my opinion, also used the term “double-reference” incorrectly. The similar equations seen in the Neurock paper and Henkelman paper (and this manuscript) pertain to using a homogeneous background counter-charge and correcting the DFT-determined energies, not how the electrode potential is referenced (which the term “double-reference” references).
 - Although the Henkelman paper reported intriguing results for ORR on Au(100), this “not-double-reference” method still has serious physical assumptions that the authors should at least acknowledge and comment on. Firstly, this potential-referencing approach is

physically questionable because a dielectric continuum (modeled by VASPsol) does not necessarily have the same electrostatic potential as a “true vacuum”, and therefore should not be used as an absolute reference to determine the electrode potential. Secondly, using a homogenous counter-charge means the counter-charge is spread evenly throughout the unit-cell volume. It means that counter-charge also permeates inside the metal slab which is unphysical as it should ideally represent only ionic counter-charge in the electrolyte region.

- Questionable VASPsol setting:
 - A dielectric constant of 80 is greatly overestimated. Something between 3 to 10 is more widely accepted as the dielectric constant of the interfacial region.
 - What is the justification for a 3 Å Debye screening length? I assume the authors use a temperature of 300 K, electrolyte concentration of 1M, and dielectric constant of 80 (again not representative of the interfacial region).
 - What is the justification for setting the surface tension term to zero, i.e., neglecting the surface tension term that encompasses short-range solvent interaction and entropic contributions? These terms may differ significantly between reaction intermediates, e.g., *CONH likely forms hydrogen bonds with water and *N likely not; and their sizes are very different.
- Microkinetic model: missed opportunities and questionable methodology.
 - The microkinetic result for urea TOF in Fig. 4a and 4d is quite baffling, which the authors did not satisfactorily explain/elaborate. Despite the slightly varying mechanism on Cu(111) and Cu(100) at different potentials, the kinetic barriers of the RDS are not that different (around 0.7 eV for both facets at most potentials). However, at 300K the Cu(100) facet is something like 55 orders of magnitude more active than Cu(111) for urea synthesis. If this is only because of their intrinsic kinetics, that would suggest something like a 3 eV difference in the activation barrier, which we don't see. This means the difference in TOF between the two facets must come from competition with side reactions (NO₂RR and CO₂RR). The authors only show TOF for NO₂RR on the two facets, which have similar orders of magnitude. Therefore, NO₂RR alone unlikely explains the difference in urea production TOF between the two facets. Would the TOF of CO₂RR (not done in the manuscript) help to explain it?
 - The authors should give the degree of rate control (DRC) to conclude the rate-determining step (to corroborate with the DFT-calculated kinetic barriers) and other relevant slow steps (which can also be informative).
 - Tables 1 and S13-S15 describe aqueous-phase species like H⁺ and NO₂⁻ with a pressure with a unit of bar in the rate equations. What do 1 or 10 bars of H⁺ and NO₂⁻ ions mean? For neutral gaseous species like CO₂ and NH₃, it is reasonable to expect a simple equilibrium relationship for gaseous pressure and surface coverage; it is not as clear for ions.
- Electric double-layer capacitance analysis: No clear correlation is observed, and physical interpretations are unsatisfactory.
 - The correlation between EDL capacitance and urea synthesis TOF is established unclearly in Fig. 5 and paragraph lines 373 to 395. Simply stating the range of capacitance measured

- in each facet (lines 378 to 383) and then concluding that Cu(111) and Cu(100) surfaces are more active than Cu(110) for urea synthesis (lines 383 to 385) is unconvincing.
- Physical interpretations for such a correlation – even though not clearly established – are also unsatisfactory. In paragraph lines 396 to 407, the authors attribute the “higher capacitance equals higher urea synthesis activity” relationship to two possibilities: “rapid diffusion of intermediates across the diffuse layer” (line 401) or “stronger electric field to support the charge transfer” (lines 402-403). Both explanations have holes.
 - Larger EDL capacitance leading to thinner EDL and easier diffusion of species across the diffuse layer to the surface is generally reasonable. However, the capacitance determined using the current approach arises from the response of a homogenous background counter-charge to excess electrode charges, not the electrolytic ions counter-charge that forms the EDL (the authors also use Poisson-Boltzmann with a 3 Å Debye length consistently, not variably to match electrode charges). Therefore, ascribing the capacitance values here to the EDL capacitance is questionable. Furthermore, the employed DFT model captures none of the diffusion kinetics (adsorption of for example NO_2^- to $^*\text{NO}_2$ is not diffusion across the diffuse layer).
 - A stronger electric field does not universally improve kinetics (i.e., lower the activation barriers). Generally, an interfacial electric field interacts with adsorbates differently depending on the adsorbate’s dipole moment, meaning it can increase or decrease an activation barrier based on the difference in dipole moments between the initial and transition states. The employed microkinetic model accounts for many reaction intermediates and transition states, which all interact with an electric field differently. It is therefore questionable to state that a stronger electric field can automatically facilitate faster urea synthesis.

Minor

- Line 67: Nitrogen oxides are often interpreted as gaseous NO, NO_2 , or N_2O . The authors could specify that they are looking at aqueous nitrite ions.
- Line 140: Justification for the rather arbitrary 0.75 eV cutoff should be given.
- Line 180: “higher electrode potential” indicates more negative potential in this sentence, which can be confusing.
- Line 192 and 195: “above a potential of -0.60 V”, should be “below” or “more negative” according to Fig. S9.
- Does Figure 2a report total energies (units not given) below each image? The total energies from DFT have no meaning and ideally should be referenced to show differences and be more readily interpretable.

Response to the comments of Reviewers

We would like to thank the reviewers for giving us comments as well as valuable suggestions to our manuscript. We have revised the manuscript according to the reviewers' comments and all the changes are highlighted in red color in the revised manuscript. Below please find a point-to-point response to the reviewers' comments.

Reviewer #1

In this work, Wu et al. used a “double-reference method” with DFT to study the potential dependent mechanism of electrocatalytic C-N coupling to produce urea from CO₂ and NO₂⁻. The authors identified and addressed all the important and interesting questions on this topic. However, I find their methodology lacks justifications for some important model choices, giving the impression that the “double-reference” method was employed without being closely examined. The section about reaction mechanisms (lines 95 to 306) proves that the authors investigated this complex reaction in a detail oriented and physically reasonable manner. Yet, the later sections (below line 308) about the microkinetic model and electric double-layer capacitance are rather unsatisfactory with many questionable physical interpretations of their results. I, ultimately, recommend not publishing this work in its current state.

Positives and takeaways:

- Credits are due for studying a complex reaction mechanism at varying electrode potentials in careful detail.
- The authors clearly elucidated abundant C-derived and N-derived species (prior to C-N coupling) on each Cu facet at different potentials (0.0 V to -1.5 V RHE).
- They identified key C-derived and N-derived intermediates whose first coupling step is more favorable or competitive to the non-coupling reduction paths.
 - For Cu(111), it is either *CO+*NH or *CO₂+*NH (depending on the applied potential).
 - For Cu(110), non-coupling reduction paths are preferred.
 - For Cu(100), multiple coupling pairs can be feasible depending on the applied potential (perhaps the most significant finding of this manuscript).
- They identified intermediates for the second C-N coupling step, which are *CO-NH + *NH on both Cu(111) and Cu(100) (while ruling out *CO₂-NH and *COOH-NH).

We appreciate the reviewer for his/her positive comments and valuable suggestions. We have updated the term from “double-reference method” to “constant potential method”. Additionally, we have provided further details on the microkinetic model and the electric double-layer capacitance in our revised version. Kindly refer to the updated version below.

Negatives/needs improvement:

Major

- “Not-double-reference” DFT:
 - The term “double-reference” in this manuscript is a misnomer. The original method by Neurock (Phys. Rev. B 73, 165402, 2006; citation 61 in this manuscript) entails referencing the electrode work function (i.e., the absolute electrode potential) to the potential of a static explicit water layer and the potential of a cleaved vacuum region (hence, double reference). The method that the authors herein employed from the Henkelman paper (ACS Catal., 9, 5567-5573, 2019; citation 58 in this manuscript) instead references the electrode work function only to a continuum slab (no explicit water) and hence, is not “double-reference”. The Henkelman paper, in my opinion, also used the term “double-reference” incorrectly. The similar equations seen in the Neurock paper and Henkelman paper (and this manuscript) pertain to using a homogeneous background counter-charge and correcting the DFT-determined energies, not how the electrode potential is referenced (which the term “double-reference” references).
 - Although the Henkelman paper reported intriguing results for ORR on Au(100), this “not double-reference” method still has serious physical assumptions that the authors should at least acknowledge and comment on. Firstly, this potential-referencing approach is physically questionable because a dielectric continuum (modeled by VASPsol) does not necessarily have

the same electrostatic potential as a “true vacuum”, and therefore should not be used as an absolute reference to determine the electrode potential. Secondly, using a homogenous counter-charge means the counter-charge is spread evenly throughout the unit-cell volume. It means that counter-charge also permeates inside the metal slab which is unphysical as it should ideally represent only ionic counter-charge in the electrolyte region.

Response: Thanks for the questions. We would like to address these questions raised point-to-point to provide a comprehensive clarification:

- The term “double-reference” in this manuscript is a misnomer.

Thanks for pointing out the potential ambiguity regarding the term “double-reference.” Upon reviewing the context, we realized that our use of the term “double-reference” may have caused confusion. We noted that the “double-reference” developed by Neurock et al. (*Phys. Rev. B* 2006, 73, 165402) including a layer of water far from the solid surface and fixed in position. By holding the electrostatic potential of the water layer (ϕ_w) constant irrespective of the solid charge, it serves as a reference for determining the relative Fermi level position. Since the relative difference between the ϕ_w and the vacuum level can be obtained from a net-neutral electronic system, this allows the Fermi level to be referenced against the vacuum level in a charged solid. As elucidated in *Phys. Rev. B* 2006, 73, 165402, this method indeed established two reference potentials: one related to the potential of the free electron in vacuum, and the other associated with distant H₂O species from the electrode.

The Henkelman group has applied the “double-reference method” to assess the impact of the applied potential on reaction energetics at the metal/solution interface. As highlighted in their publication (*Langmuir* 2018, 34, 15268): “Different from the original implementation of the double-reference method, where explicit water molecules are used to model the metal/aqueous interface, the aqueous environment in this study is treated by the continuum solvation model developed by the Hennig group as implemented in the VASPsol code. The electric potential of the slab referenced to the SHE is calculated as $U_q = -4.6 - \varphi_q/eV$, where $-\varphi_q$ is the work function of the charged slab and 4.6 V is the work function of the H₂/H⁺ couple under standard conditions. To calculate φ_q , the ascalculated Fermi energy of the charged system φ'_q needs to be referenced twice to be properly referenced to the vacuum level. The first reference is the electrostatic potential in the middle of the solution phase far from the electrode. The absolute electrostatic potential in the middle of the solution is considered constant as a function of charge. The second reference point is the vacuum level in the uncharged calculation.”.

We agree with the reviewer’s observation that this approach does not constitute a “double-reference” in the strict sense. Therefore, we revised the “double-reference method” to the more acceptable “constant-potential method” in the revised version.

To address this point in the revised manuscript, the statement of “Different from the original implementation of the double-reference method,⁶⁶ which employs explicit water molecules to represent the metal/aqueous interface.” has been added in Page 12.

- Although the Henkelman paper reported intriguing results for ORR on Au(100), this “not double-reference” method still has serious physical assumptions that the authors should at least acknowledge and comment on.

Thanks for reviewer's insights regarding the physical assumptions in the implicit solvation methods as implemented by VASPsol. Indeed, the implicit solvation approach, while computationally efficient, does introduce approximations, notably the potential-referencing method and the usage of a homogeneous counter-charge. We acknowledged that a fully explicit description of the electrolyte would provide the most accurate description of the effects of solvation, interfacial fields, and applied bias. Such a method would involve ab initio molecular dynamics (AIMD) simulations incorporating both water molecules and ions explicitly. However, due to the computational demands posed by such a method, especially when considering complex reaction mechanisms like urea synthesis in this study, it remains prohibitive for our current computational resources. We would have preferred to delve into this level of detail, but such simulations are beyond our computational budget at present. Furthermore, as underscored by Nørskov’s research (*ACS Catal.* 2019, 9, 920), solvent effects remain a challenging arena to accurately probe even when employing explicit water or MD simulations.

The VASPsol-based implicit solvation method has been successfully applied in a number of recent mechanistic studies (*Nat. Commun.* 2023, 14, 936; *J. Am. Chem. Soc.* 2023, 145, 7030; *J. Am. Chem. Soc.* 2023, 145, 14335). It also has been applied as part of a scheme for grand canonical potential kinetics method, which is developed by Goddard et al., calculates the reaction energetics at different charges and then transforms the free energy to that under constant potential. (*Nat. Commun.* 2020, 11, 2256; *J. Am. Chem. Soc.* 2018, 140, 16773). Especially, in this manuscript, for species exhibiting strong interactions with water molecules (those favoring E-R mechanism over L-H mechanism) like hydrogen bond formations, we included explicit water molecules to accurately represent the solvation effect at various states of our calculations. This microsolvation approach, combining minimal explicit solvent molecules within a continuum model, has proven effective and accurate in predicting pKa values of acids in aqueous solutions (*ACS Catal.* 2019, 9, 920; *ACS Catal.* 2018, 8, 2188). Such an approach enables us to bypass the extensive calculations involved in explicit solvent and ion inclusion while maintaining an ab initio consideration of solvent effects at the interface. Therefore, the implicit solvation method employed VASPsol in this manuscript is also reasonable.

We appreciate the reviewer's emphasis on these aspects, and we have made an effort to acknowledge these limitations transparently in the revised version, stressing the importance of such considerations for future studies. The discussions of "A fully explicit description of the electrolyte would provide the most accurate description of the effects of solvation, interfacial fields, and applied bias. Such a method would involve AIMD simulations incorporating both water molecules and ions explicitly.⁶⁹ However, given the significant computational requirements of such an approach, particularly when exploring complex reaction mechanisms like urea synthesis in our study, it exceeds the capabilities of our current computational resources." have been added in Page 12.

- Questionable VASPsol setting:
 - A dielectric constant of 80 is greatly overestimated. Something between 3 to 10 is more widely accepted as the dielectric constant of the interfacial region.
 - What is the justification for a 3 Å Debye screening length? I assume the authors use a temperature of 300 K, electrolyte concentration of 1M, and dielectric constant of 80 (again not representative of the interfacial region).
 - What is the justification for setting the surface tension term to zero, i.e., neglecting the surface tension term that encompasses short-range solvent interaction and entropic contributions? These terms may differ significantly between reaction intermediates, e.g., *CONH likely forms hydrogen bonds with water and *N likely not; and their sizes are very different.

Response: Thanks for raising these valuable concerns. We feel sorry that the reasons for choosing parameters values were not explained well in the previous version. We would like to clarify on these points:

- A dielectric constant of 80 is greatly overestimated. Something between 3 to 10 is more widely accepted as the dielectric constant of the interfacial region.

In our study, we employed the implicit solvation model of VASPsol to account for the solvation effect—the influence of the aqueous solution environment on the catalytic reaction. While a dielectric constant of 80 corresponds the permittivity of bulk water, we note that water at the metal/aqueous interface and the existence of electrolyte ions can cause the relative permittivity to deviate from the bulk value. However, previous works (*Nat. Commun.* 2023, 14, 936; *J. Am. Chem. Soc.* 2022, 144, 39; *J. Am. Chem. Soc.* 2022, 144, 16524) indicated that the thermodynamics of proton-coupled electron transfer (PCET) are captured reliably within this computational approach using the dielectric constant of bulk water without a low dielectric interfacial region.

For an optimal depiction, the relative permittivity of the electrolyte would involve simulations with explicit water molecules and ions in the solution. However, the computational demands for such ab initio molecular dynamics simulations, which would effectively capture the interfacial electrolyte structures, currently exceed our available resources. Nevertheless, for species having pronounced water molecule interactions, we did incorporate explicit water molecules in our calculations. This microsolvation method, a blend of minimal explicit solvent molecules with a continuum model, has been demonstrated to be both effective and precise in earlier research (*Nat. Commun.* 2020, 11, 2256; *ACS Catal.* 2019, 9, 920; *ACS Catal.* 2018, 8, 2188).

In addition, Norskov et al. (*ACS Catal.* 2019, 9, 920) have discussed the impact of a decreased bulk

dielectric constant (ϵ_b). When the ϵ_b decreased from the 78.4 to 4, the associated capacitance is lowered from 13.7 to 5.4 $\mu\text{F}/\text{cm}^2$. The decrease in capacitance from lowering the bulk dielectric constant synergizes with the decrease from pushing the countercharge density away from the surface; with both modifications made, the predicted capacitance drops to about 2.7 $\mu\text{F}/\text{cm}^2$, which is about an order of magnitude lower than experimental capacitances. Therefore, a lower dielectric constant is not conducive to obtaining a more accurate capacitance value, and a dielectric constant of 80 is suitable for the scope of this study.

- What is the justification for a 3 Å Debye screening length?

Our utilization of a 3.0 Å Debye screening length is rooted in the linear Poisson–Boltzmann implicit solvation model. This model assumes a 1M aqueous solution containing monovalent cations at 300 K. Adopting this screening length ensures neutralization of the non-zero charge in the simulation cell and realistically simulate water and the electrolyte, providing an authentic depiction of the electrochemical double layer, which is paramount for our study. Importantly, this short Debye length enables the counter charge to remain proximal to the electrode surface, resulting in a uniform electrostatic potential in the solution region's center due to the electrolyte's screening effect (*ACS Catal.* 2021, 11, 14439). Moreover, the 3 Å Debye screening length has garnered widespread application in pioneer researches (*Nat. Catal.* 2021, 4, 1024; *Nat. Commun.* 2020, 11, 3844; *Nat. Commun.* 2019, 10, 3340; *ACS Catal.* 2019, 9, 8197).

To address this point in the revised manuscript, the statement of “The linear Poisson-Boltzmann model with a Debye screening length of 3.0 Å mimics the compensating charge, allowing for a more realistic description of the EDL.” has been added in Page 12.

- What is the justification for setting the surface tension term to zero, i.e., neglecting the surface tension term that encompasses short-range solvent interaction and entropic contributions?

The decision to set the surface tension term to zero was initially a simplification strategy, as reflected in earlier studies (*Nat. Catal.* 2021, 4, 1024; *J. Am. Chem. Soc.* 2022, 144, 39; *ACS Catal.* 2020, 10, 7826). As noted above, we have incorporated explicit water molecules for species likely to form hydrogen bonds, ensuring an accurate representation of the solvation effect. A study by Norskov et al. (*ACS Catal.* 2019, 9, 920) highlighted that modifications to the surface tension term induced only minor shifts in capacitance and potentials of zero charges. Therefore, setting the surface tension term to zero does not change the conclusion of this study, and we believe that this setting is appropriate for our study.

- Microkinetic model: missed opportunities and questionable methodology.

- The microkinetic result for urea TOF in Fig. 4a and 4d is quite baffling, which the authors did not satisfactorily explain/elaborate. Despite the slightly varying mechanism on Cu(111) and Cu(100) at different potentials, the kinetic barriers of the RDS are not that different (around 0.7 eV for both facets at most potentials). However, at 300K the Cu(100) facet is something like 55 orders of magnitude more active than Cu(111) for urea synthesis. If this is only because of their intrinsic kinetics, that would suggest something like a 3 eV difference in the activation barrier, which we don't see. This means the difference in TOF between the two facets must come from competition with side reactions (NO₂RR and CO₂RR). The authors only show TOF for NO₂RR on the two facets, which have similar orders of magnitude. Therefore, NO₂RR alone unlikely explains the difference in urea production TOF between the two facets. Would the TOF of CO₂RR (not done in the manuscript) help to explain it?
- The authors should give the degree of rate control (DRC) to conclude the rate-determining step (to corroborate with the DFT-calculated kinetic barriers) and other relevant slow steps (which can also be informative).
- Tables 1 and S13-S15 describe aqueous-phase species like H⁺ and NO₂⁻ with a pressure with a unit of bar in the rate equations. What do 1 or 10 bars of H⁺ and NO₂⁻ ions mean? For neutral gaseous species like CO₂ and NH₃, it is reasonable to expect a simple equilibrium relationship for gaseous pressure and surface coverage; it is not as clear for ions.

Response: Thanks for raising these valuable concerns. We acknowledge the concerns raised and agree that additional analysis is necessary to clarify the methodologies used in our microkinetic model. We would like to share our opinions with the reviewer.

- NO₂RR alone unlikely explains the difference in urea production TOF between the two facets. Would

the TOF of CO₂RR (not done in the manuscript) help to explain it?

Indeed, the kinetic barriers of the rate determining step (RDS) for Cu(111) and Cu(100) surfaces are not that different. The significant variance in TOF between the two surfaces can be attributed to their distinct reaction mechanisms and RDSs. According to the reviewer's recommendation, we have revised the pressure of solvated species like NO₂, H₂O, and H⁺ to the concentration of the reactants, which significantly influence the turnover frequencies (TOF) differences on Cu(111) and Cu(100) surfaces. These modifications successfully minimized the difference, as illustrated in Figure R1 and the method section in the revised version.

Figure R1 Microkinetic simulations. Turnover frequencies (TOF) per site for urea synthesis on **a** Cu(111) and **d** Cu(100) surfaces as functions of applied electrode potential vs RHE at 300 K and 1 bar. TOFs per site for NH₃ synthesis on **b** Cu(111) and **e** Cu(100) surfaces as a function of applied electrode potential vs RHE at 300 K and 1 bar. TOFs per site for urea synthesis on **c** Cu(111) and **f** Cu(100) surfaces mapped with pressure (1-100 bar) and temperature (300-600 K and 300-900K, respectively).

We feel sorry that the TOF calculations of CO₂RR were not provided in the previous version. The reactions involving CO₂RR on Cu surfaces are notably complex. Our initial focus was on CO₂RR to CO and the next hydrogenation step, which is directly related to urea synthesis. To offer a more comprehensive analysis, we hypothesized that CO is the main product of CO₂RR on Cu surfaces, and subsequent calculations and analysis of the TOF of CO₂RR on both Cu(111) and Cu(100) surfaces were undertaken. Interestingly, our data suggests that for the applied potential range, the TOF values for both surfaces are negative. This indicates that the reaction is reversible, corroborating our findings and aligning with previous research (*Nat. Commun.* 2017, 8, 15438; *J. Am. Chem. Soc.* 2017, 139, 130; *J. Am. Chem. Soc.* 2016, 138, 483), indicating that *CO struggles with desorption and tends to further reactions on Cu surfaces. Specifically, the Cu(100) surface presents a more challenging environment for *CO desorption in comparison to Cu(111) surface.

Figure R2 TOF for CO₂RR to CO on Cu(111) and Cu(100) surfaces as functions of applied electrode potential vs RHE at 300 K and 1 bar.

To incorporate these insights into the work, below three modifications have been added in the revised version:

- ◆ Figure R1 has been revised and shown in the manuscript as **Figure 4**.
- ◆ Figure R2 has been incorporated as **Supplementary Figure 28**.
- ◆ The discussions of “The TOF for CO₂RR on Cu(111) and Cu(100) surfaces are also shown in **Supplementary Fig. 28**. Within the applied potential range, both surfaces exhibit negative TOF

values, signifying the reversibility of the reaction. The results corroborate our findings and are consistent with previous studies which suggest that *CO struggles with desorption and tends to further reactions on Cu surfaces.^{39,50,51} Notably, the Cu(100) surface presents a more challenging environment for *CO desorption than the Cu(111) surface.” have been added in Page 9.

- The authors should give the degree of rate control (DRC) to conclude the rate-determining step (to corroborate with the DFT-calculated kinetic barriers) and other relevant slow steps (which can also be informative).

We feel sorry that the degree of rate control (DRC) was not provided in the previous version. We have included the DRC in our revised version. For a clearer understanding, we've added computational details of how DRC was determined in our revised methods section. A summary of the computed DRC values for all intermediates and transition states related to the urea synthesis process on both Cu(111) and Cu(100) surfaces, can be found in Tables R1 and R2.

A positive (negative) X_i value signifies that an increase in reaction rate r need to further stabilization (destabilization) of the corresponding surface state (*ACS Catal.* 2017, 7, 2770; *J. Am. Chem. Soc.* 2009, 131,8077). Our DRC analysis corroborates that, for the Cu(111) surface, the RDS in urea synthesis is the coupling of *CO and *NH, with a value of $X_{(\text{CO-NH})} = 1.0$. Meanwhile, on the Cu(100) surface, the RDS is determined to be the hydrogenation step of *NO₂, with $X_{(\text{NO}_2\text{-H})} = 1.0$. This interpretation is consistent with the observed irreversible nature of the RDS and the steady-state nature of all subsequent steps. Further analysis also highlighted the notable coverage of *NH on the Cu(111) surface and *NO₂ on the Cu(100) surface, as represented in Figure R3. This resulted in negative X_i values for N-intermediates and *NO₂ on Cu(111) and Cu(100) surfaces, respectively. Intriguingly, the C-intermediate displayed a positive X_i value during urea synthesis on the Cu(111) surface, indicating that challenges in the CO-NH coupling on the Cu(111) surface stem primarily from the adsorption constraints of the C-intermediate, which is well consistent with the results of Figure R4.

Table R1 Degrees for rate control of all intermediates and transition states of urea synthesis process on the Cu(111) surface at 300 K under the applied electrode potential of -0.25 V.

Species (intermediates)	X_i	Species (transition-states)	X_i
*NO ₂	-1.0	*-NO ₂	0.0
*HNO ₂	-1.0	NO ₂ -H	0.0
*NO	-1.0	NOOH-H	0.0
*NOH	-1.0	NO-H	0.0
*N	-1.0	NOH-H	0.0
*NH	-1.0	N-H	0.0
*COOH	1.0	COO-H	0.0
*CO	1.0	COOH-H	0.0
*CO-NH	1.69×10^{-26}	CO-NH	1.0
*NH-CO-NH	1.69×10^{-26}	NHCO-NH	0.0
*NH-CO-NH ₂	1.69×10^{-26}	NHCONH-H	0.0
*	1.69×10^{-26}	*-CO(NH ₂) ₂	0.0

Table R2 Degrees for rate control of all intermediates and transition states of urea synthesis process on the Cu(100) surface at 300 K under the applied electrode potential of -0.5 V.

Species (intermediates)	X_i	Species (transition-states)	X_i
*NO ₂	3.27×10^{-23}	*-NO ₂	0.0
*HNO ₂	0.0	NO ₂ -H	1.0
*NO	0.0	NOOH-H	0.0
*NOH	0.0	NO-H	0.0
*N	0.0	NOH-H	0.0
*NH	3.73×10^{-30}	N-H	0.0
*CO ₂ -N	-3.73×10^{-30}	CO ₂ -N	0.0
*COOH-N	-3.73×10^{-30}	NCOO-H	0.0

*CO-N	-3.73×10^{-30}	CO-N	0.0
*CO-NH	-3.73×10^{-30}	CON-H	0.0
*NH-CO-NH	3.73×10^{-30}	NHCO-NH	0.0
*NH-CO-NH ₂	3.73×10^{-30}	NHCONH-H	0.0
*	3.73×10^{-30}	*-CO(NH ₂) ₂	0.0

Figure R3 Simulated coverage curves of adsorbed species as a function of temperature on Cu(111) (300 to 900K) and Cu(100) (300 to 1000K) under the applied electrode potential of -0.25 and -0.5 V, respectively.

Figure R4 TOF of urea synthesis as a function of NO₂⁻ concentration (0 to 1 M), CO₂ pressure (0 to 100 bar) and H⁺ concentration (0 to 0.1 M) for Cu(111) and Cu(100) under the temperature of 300 K and the applied potential of -0.25 and -0.5 V, respectively.

To summaries, below changes have been made in the revised version to address the concerns about DRC in this work:

- ◆ Table R1 has been incorporated as Supplementary Table 14.
- ◆ Table R2 has been incorporated as Supplementary Table 17.
- ◆ Figure R3 has been incorporated as Supplementary Figure 29.
- ◆ Figure R4 has been incorporated as Supplementary Figure 31.
- ◆ The discussions of “To fully understand the microkinetic mechanisms, the degree of rate control (DRC, X_i) is conducted for all intermediates and transition states (Supplementary Tables 14,17). Our DRC analysis corroborates that, on the Cu(111) surface, the RDS in urea synthesis is the coupling of *CO and *NH, with a value of $X_{(CO-NH)} = 1.0$. Meanwhile, on the Cu(100) surface, the RDS is the hydrogenation step of *NO₂ with $X_{(NO2-H)} = 1.0$. This interpretation is consistent with the observed irreversible nature of the RDS and the steady-state nature of all subsequent steps.” and “Consequently, negative X_i values can be observed for N-intermediates and *NO₂ on Cu(111) and Cu(100) surfaces, respectively.” have been added in Page 9.
- ◆ The discussions of “Intriguingly, the C-intermediates display a positive X_i value during urea synthesis on the Cu(111) surface, indicating that challenges in the CO-NH coupling on the Cu(111) surface stem primarily from the adsorption constraints of the C-intermediates, which is well consistent with the results of Supplementary Fig. 31.” have been added in Page 10.

- ♦ The discussions of “The generalized DRC (X_i) is used to distinguish the relative importance for each species i ,^{78,79}

$$X_i = \left(\frac{-\partial \ln r}{\partial \left(\frac{G_i^0}{k_b T} \right)} \right)_{G_{j \neq i}^0}$$

where r is the net reaction rate to the product of interest, G_i is the free energy change of each intermediate and transition state. The partial derivative is now taken holding constant the free energy change of all other species (intermediates and transition states), j .^{78,79} have been added in the method section.

- What do 1 or 10 bars of H_+ and NO_2^- ions mean? For neutral gaseous species like CO_2 and NH_3 , it is reasonable to expect a simple equilibrium relationship for gaseous pressure and surface coverage; it is not as clear for ions.

Thanks for the reviewer’s comment. In our revised approach, we aligned our methodology with earlier studies to represent the pressure of solvated species (including NO_2^- , H_2O , and H^+), in terms of reactant concentration. The corresponding microkinetic equations have been revised.

To address this point in the revised manuscript, below discussions have been added in Page 14 in the revised version: “In this work, the concentration of solvated $CO(NH_2)_2$ and NO_2^- are set at 1×10^{-4} M and 0.1 M, respectively.⁴⁴ C_{H_2O} denotes the bulk concentration of H_2O and equals to 1.⁷⁶ The concentration of H^+ is calculated by: $C_{H^+} = 10^{-pH}$.⁷⁷”.

- Electric double-layer capacitance analysis: No clear correlation is observed, and physical interpretations are unsatisfactory.
 - The correlation between EDL capacitance and urea synthesis TOF is established unclearly in Fig. 5 and paragraph lines 373 to 395. Simply stating the range of capacitance measured in each facet (lines 378 to 383) and then concluding that Cu(111) and Cu(100) surfaces are more active than Cu(110) for urea synthesis (lines 383 to 385) is unconvincing.
 - Physical interpretations for such a correlation – even though not clearly established – are also unsatisfactory. In paragraph lines 396 to 407, the authors attribute the “higher capacitance equals higher urea synthesis activity” relationship to two possibilities: “rapid diffusion of intermediates across the diffuse layer” (line 401) or “stronger electric field to support the charge transfer” (lines 402-403). Both explanations have holes.
 - Larger EDL capacitance leading to thinner EDL and easier diffusion of species across the diffuse layer to the surface is generally reasonable. However, the capacitance determined using the current approach arises from the response of a homogenous background counter-charge to excess electrode charges, not the electrolytic ions counter-charge that forms the EDL (the authors also use Poisson-Boltzmann with a 3 Å Debye length consistently, not variably to match electrode charges). Therefore, ascribing the capacitance values here to the EDL capacitance is questionable. Furthermore, the employed DFT model captures none of the diffusion kinetics (adsorption of for example NO_2^- to $*NO_2$ is not diffusion across the diffuse layer).
 - A stronger electric field does not universally improve kinetics (i.e., lower the activation barriers). Generally, an interfacial electric field interacts with adsorbates differently depending on the adsorbate’s dipole moment, meaning it can increase or decrease an activation barrier based on the difference in dipole moments between the initial and transition states. The employed microkinetic model accounts for many reaction intermediates and transition states, which all interact with an electric field differently. It is therefore questionable to state that a stronger electric field can automatically facilitate faster urea synthesis.

Response: Thanks for raising these valuable concerns. We are glad to share our opinions with the reviewer.

- The correlation between EDL capacitance and urea synthesis TOF is established unclearly in Fig. 5 and paragraph lines 373 to 395.

We are sorry for that the correlation between electric double layer (EDL) capacitance and urea synthesis TOF is not clearly stated in previous manuscript. We further refine this part in the revised

manuscript. The discussions of “On Cu(111), Cu(110) and Cu(100) surfaces, electrochemical energy conversion occurs via charge transfer reactions at the EDL. The kinetics and favored reaction pathways of electrochemical reactions are primarily influenced by the structure of EDL and the intrinsic interactions between the electrode and the electrolyte.⁵⁸ Several fundamental interactions within the EDL, including electrostatic forces, covalent bonds, and non-covalent bonds, together shape the dynamic nature of the interfacial structure. Specifically, during the coupling step of urea synthesis, covalent interactions involve direct bond formations, which encompass orbital overlaps and chemical interactions between the adsorbates. Reaction intermediates predominantly reside within the inner Helmholtz plane (IHP). The adsorption energy of these intermediates is strongly dependent on the interfacial field, the specific nature of the electrode surface, and thus resulting in a distinct electrocatalytic reaction rate. Notably, the EDL capacitances for intermediates involved in the NO₂⁻RR and CO₂RR are comparable on Cu(111) and Cu(100) surfaces. This suggests that the distribution, adsorption, and migration behaviors of N/C-intermediates on these electrode surfaces might be analogous. Owing to their resembling physicochemical attributes and interfacial behaviors, the interactions between these intermediates might be enhanced, facilitating their coupling and subsequent reactions. However, the situation on the Cu(110) surface presents challenges regarding the substantial difference in capacitances between the NO₂⁻RR and CO₂RR processes. Such distinctions in behavior on Cu(110) surface might hinder the interaction of N-intermediate with C-intermediate, potentially compromising the efficiency of the coupling reaction.” have been added in Page 11.

- Physical interpretations for such a correlation—even though not clearly established—are also unsatisfactory. In paragraph lines 396 to 407, the authors attribute the “higher capacitance equals higher urea synthesis activity” relationship to two possibilities: “rapid diffusion of intermediates across the diffuse layer” (line 401) or “stronger electric field to support the charge transfer” (lines 402-403). Both explanations have holes.

Referring to the work of Norskov et al. (*ACS Catal.* 2019, 9, 920), it has been demonstrated that the VASPsol methodology provides reliable estimates for capacitances and potentials of zero charges (PZCs) on single-crystal transition-metal surfaces. As described earlier (*Nat. Commun.* 2020, 11, 2256; *J. Am. Chem. Soc.* 2022, 144, 39; *ACS Catal.* 2020, 10, 7826), the constant-potential method could directly give the double layer capacitance (C_{dl}), the results can be compared with experiment. Various experimental studies have observed the C_{dl} to lie between 20 and 25 $\mu\text{F}/\text{cm}^2$ for a majority of metal surfaces. Ringe et al. (*Energy Environ. Sci.* 2019, 12, 3001) assumed a value of 25 $\mu\text{F}/\text{cm}^2$ in their model to address double layer effects for CO₂ reduction reaction on Ag (111) single crystal electrode. As seen in Figure 5 and detailed in Supplementary Tables 1-12 of our study, the calculated C_{dl} values are closely align with these experimentally reported values (*J. Electroanal. Chem. Interfacial Electrochem.* 1983, 145, 225; *J. Electroanal. Chem. Interfacial Electrochem.* 1982, 138, 37). Therefore, the capacitance determined using the constant-potential method in this study is at least within a reasonable range.

To address this point in the revised manuscript, the statement of “The calculated C_{dl} values shown in Supplementary Tables 1-12 are closely aligning with experimentally reported values,^{55,56} which further validates our methodology for catalyst-electrolyte interface.” has been added in Page 11.

We agree with the reviewer's perspective on the distinct interactions of the interfacial electric field with varying adsorbates, depending on their respective dipole moments. It's crucial to recognize that shifts in the activation barrier depends on the change in dipole moments between the initial states and the transition phases. Acknowledging this, we have removed the expression regarding the effect of the electric field on the dynamics.

Minor

- Line 67: Nitrogen oxides are often interpreted as gaseous NO, NO₂, or N₂O. The authors could specify that they are looking at aqueous nitrite ions.

Response: Thanks for the reviewer's nice reminder. The “nitrogen oxides” has been specified as “NO₂⁻” in the revised version.

Despite the fundamental importance and huge interest, understanding the mechanism is still challenging in electrochemical coupling of CO₂ with nitrogen oxides (such as NO₂⁻ in wastewater) toward urea after decades studies.¹⁸⁻²²

- Line 140: Justification for the rather arbitrary 0.75 eV cutoff should be given.

Response: Thanks for the reviewer's nice reminder. A value of 0.75 eV is generally regarded as kinetically feasible in electrocatalytic reactions (*ACS Catal.* 2022, 12, 11494), and at room temperature, the hard limit for a surmountable barrier is 0.75 eV (*Nature* 2022, 611, 702). We have included relevant references (*Nature* 2022, 611, 702 and *ACS Catal.* 2022, 12, 11494) to support this claim in the revised manuscript.

- Line 180: "higher electrode potential" indicates more negative potential in this sentence, which can be confusing.

Response: Thanks for pointing the typo out. We are sorry for the confusion caused, the "higher electrode potential" has been corrected as "a more negative potential" in the revised version.

- Line 192 and 195: "above a potential of -0.60 V", should be "below" or "more negative" according to Fig. S9.

Response: Thanks for pointing this out. We are sorry for the confusion caused, the "above a potential of -0.60 V" has been corrected to "below a potential of -0.60 V" in the revised version.

- Does Figure 2a report total energies (units not given) below each image? The total energies from DFT have no meaning and ideally should be referenced to show differences and be more readily interpretable.

Response: We appreciate the reviewer for pointing out this issue. We agree and have made necessary changes in the revised manuscript. The total energies below each image in Fig. 2a and Fig. 3e have been replaced by the energy differences in the revised version, marked in red:

Fig. 2 Kinetic evaluation of NO₂RR, CO₂RR, and the first C-N coupling step.

Fig. 3 Reaction mechanism for the second C-N coupling and final hydrogenation steps.

Reviewer #2 (Remarks to the Author):

Response: We would like to express our gratitude to Reviewer #2 for participating in the peer-review process as a co-reviewer. Thanks for valuable suggestions from reviewer. We have made the additional note as advised to improve the clarity and readability of this manuscript.

Reviewer #3 (Remarks to the Author):

The authors present a computational study of urea synthesis on three copper surfaces. Their results show that Cu(100) presents the most favorable energetics for urea synthesis of the models considered. The authors very nicely and thoroughly analyze the system, with a well-rationalized mechanism proposed. Although this work is very thorough and careful, I believe there is a significant flaw in the analysis, along with some substantial limitations to the microkinetic modeling approach; these items limit my ability to fully interpret the conclusions. I believe that as a result a major revision is needed before considering its publication.

Most significant: when calculating activation energies of E-R steps, what is the validity of using uncharged H₃O as a hydrogen source? If this species is not charged, it creates an artificial driving force for H transfer and thereby will artificially lower the barriers for H transfer. This is true for all the considered E-R steps, and likely is a reason why the barriers for E-R steps are significantly lower than those for L-H steps. More physically representative would be, e.g., a mechanism by which transfer of an electron is made to the surface-bound species, followed by transfer of a proton (not charge-neutral H). I fear that this has substantial impacts on the calculated reaction energetics and will affect the conclusions of this work, as I'm unable to ascertain what the effect on the mechanisms may be.

Response: Thanks for the reviewer's comments. We feel sorry that the reason for choosing the H₃O as a hydrogen source was not explained well in the previous version. We would like to clarify on this point. We chose the H₃O as a hydrogen source based on the following reasons:

- In the pioneer theoretical studies (*Nat. Commun.* 2023, 14, 112; *Nat. Commun.* 2023, 14, 3256; *J. Am. Chem. Soc.* 2022, 144, 12800 and *J. Am. Chem. Soc.* 2021, 143, 9423), a hydrogen atom was generally placed on a water molecule nearby the adsorbate to form an H₃O unit, which acted as a proton source in various electrochemical reactions, including NORR, NRR and ORR. To address this point in the manuscript, the corresponding references are cited in the revised version (*Nat. Commun.* 2023, 14, 112; *Nat. Commun.* 2023, 14, 3256; *J. Am. Chem. Soc.* 2022, 144, 12800 and *J. Am. Chem. Soc.* 2021, 143, 9423).
- In our work, we presented that the E-R steps were generally more favorable than the L-H steps for most of the reaction steps. These results align well with previous simulation results for electrochemical synthesis of urea, where the E-R mechanism was also shown to be more favorable than the L-H mechanism for the hydrogenation steps (*Nat. Commun.* 2022, 13, 5471 and *Nat. Commun.* 2021, 12, 4080).

Therefore, chose the H₃O as a hydrogen source is suitable in this work.

Microkinetic modeling plays an important role in this work. I commend the authors for applying microkinetic modeling in a complex environment, though the limitations of microkinetic modeling in complex electrochemical environments are significant and will affect the quantitative conclusions, if not the qualitative conclusions. Among the limitations that I believe need to be more carefully considered are (i) adsorbate interactions at higher coverages, including with applied potential;

Response: Thanks for the important points raised by the reviewer. This brings up an interesting discussion. We would like to share our opinions with the reviewer. We investigated the relationship between the turnover frequency (TOF) values on Cu(111) and Cu(100) surfaces and three parameters: NO₂⁻ concentration (0 to 1 M), CO₂ pressure (0 to 100 bar), and pH (1 to 14). These results are illustrated in Figure R1. Our findings are under conditions where the optimal applied potential is -0.25 V for Cu(111) and -0.5 V for Cu(100). When the NO₂⁻ concentration or CO₂ pressure increases, or the pH value decreases, it indicates that the reactants NO₂⁻, CO₂ and H⁺ could be adequately adsorbed onto the electrocatalysts.

Figure R1 TOF of urea synthesis as a function of NO_2^- concentration (0 to 1 M), CO_2 pressure (0 to 100 bar) and H^+ concentration (0 to 0.1 M) for Cu(111) and Cu(100) under the temperature of 300 K and the applied potential of -0.25 and -0.50 V, respectively.

From Figure R1, we observed the following trends:

- For the Cu(111) surface:
 - ◆ TOF values increase with a decrease in NO_2^- concentration or an increase in CO_2 pressure.
 - ◆ TOF values in alkaline environments are higher than in acidic ones.
- For the Cu(100) surface:
 - ◆ TOF values increase with an increase in NO_2^- concentration, plateauing once the concentration reaches 0.1 M.
 - ◆ CO_2 pressure (1-100 bar) doesn't significantly alter the TOF values.
 - ◆ TOF values increase with an increase H^+ concentration.

It should be noted that the dependencies of TOF on reactants between Cu(111) and Cu(100) surfaces are intrinsically linked to the variances in their reaction mechanisms. For the Cu(111) surface, the rate-determining step (RDS) is the coupling of $^*\text{NH}$ and $^*\text{CO}$, making TOF values particularly sensitive to NO_2^- concentration and CO_2 pressure. In contrast, the RDS on the Cu(100) surface is the hydrogenation of $^*\text{NO}_2$, thus TOF values exhibit a strong correlation with NO_2^- concentration and H^+ concentration, while CO_2 pressure has a minimal effect.

Moreover, we also accessed how the NO_2^- concentration, CO_2 pressure, and pH impact the adsorbate coverages on Cu(111) and Cu(100) surfaces. A noteworthy observation on the Cu(111) surface is that adsorbate coverages remain consistent within the pH range of 1 to 8 (acidic environment), while as pH transitions from 9 to 14 (alkaline environment), the coverages fluctuate with pH, as depicted in Figure R2. This signifies that pH not only modulates the TOF but also alters the reaction mechanism on the Cu(111) surface. Intriguingly, variations in NO_2^- concentration and CO_2 pressure will not change the adsorbate coverages on the Cu(111) surface. In contrast, on the Cu(100) surface, both NO_2^- and H^+ concentration (pH = 9 to 14) will profoundly alter the adsorbate coverages, with the results shown in Figures R2,R3.

Figure R2 Simulated coverage curves of adsorbed species for urea synthesis as a function of temperature on **a** Cu(111) and **b** Cu(100) under different pH values (pH = 9 to 14).

Figure R3 Simulated coverage curves of adsorbed species for urea synthesis as a function of temperature on Cu(100) under different NO_2^- concentrations ($C_{NO_2^-} = 0.0001$ to 0.5 M).

To address these points, the following changes have been made in the revised version:

- Figure R1 has been incorporated as **Supplementary Figure 31**.
- Figure R2 has been incorporated as **Supplementary Figure 32**.
- Figure R3 has been incorporated as **Supplementary Figure 33**.
- The discussions of “Additionally, the relationships between the TOF values on Cu(111)/Cu(100) surfaces and reactant concentrations are investigated (Supplementary Fig. 31): NO_2^-/H^+ concentration and CO_2 pressure. The TOF dependencies on reactants for Cu(111) and Cu(100) surfaces differ, a variation intrinsically linked to discrepancies in reaction mechanisms. On the Cu(111) surface, the RDS is the coupling of $*NH$ and $*CO$, making TOF values particularly sensitive to NO_2^- concentration and CO_2 pressure. In contrast, the RDS on the Cu(100) surface is the hydrogenation of $*NO_2$, thus TOF values exhibit a strong correlation with NO_2^- concentration and H^+ concentration, while CO_2 pressure has a minimal effect.” and “The NO_2^-/H^+ concentration and CO_2 pressure impact of the adsorbate coverages on Cu(111) and Cu(100) surfaces are also accessed. A notable finding on the Cu(111) surface is the consistent adsorbate coverages within the pH range of 1 to 8 (acidic environment), while as pH transitions

to 9 to 14 (alkaline environment), the coverages fluctuate with pH, as depicted in Supplementary Fig. 32. These signify that pH not only modulates the TOF but also alters the reaction mechanism on the Cu(111) surface. Interestingly, alterations in NO_2^- concentration and CO_2 pressure will not change the adsorbate coverages on the Cu(111) surface. In contrast, on the Cu(100) surface, both NO_2^- and H^+ concentration (pH = 9 to 14) will profoundly alter the adsorbate coverages, with the results shown in Supplementary Figs. 32,33.” have been added in Page 10.

(ii) adsorption of charged species like NO_2^- with respect to potential and quantifying their coverage/competitive adsorption, and (in general) determination of appropriate chemical potential references in the liquid phase;

Response: Thanks for the reviewer’s comments. We feel sorry that the adsorption of NO_2^- with respect to potential, the coverage/competitive adsorption of NO_2^- and the chemical potential references were not provided in the previous version. The adsorption energies of NO_2^- and the corresponding competitive adsorption energies of CO_2 and H^+ with respect to potential are provided below in Figures R4,R5 and the chemical potential references for the NO_2^- , NH_3 , CO_2 , CO , H_2O and H^+ are given in the Table R1 in the revised version.

Figure R4 Adsorption energies of $^*\text{NO}_2$, $^*\text{CO}_2$, and $^*\text{H}$ as a function of U/RHE (from 0 to -1.5V) on a **a** Cu(111), **b** Cu(110) and **c** Cu(100) surfaces. The pH is set as 8.3 for urea synthesis in accordance with the experimental environments.

Figure R5 Adsorption energies of $^*\text{NO}_2$ at varying coverages as a function of U/RHE (from 0 to -1.5V) on the Cu(111) surface. The pH is set as 6.8 for NO_2^- RR in accordance with the experimental environments. The ML indicates the monolayer.

Table R1 Calculated energy, free energy correction and corresponding Gibbs free energy values for HNO_2 , NH_3 , CO_2 , CO , H_2O and H_2 .

	E (eV)	ZPE-TS (eV)	G (eV)
HNO_2	-23.125	0.451	-22.674
NH_3	-19.542	0.327	-19.215
CO_2	-22.952	-0.349	-23.301
CO	-14.780	-0.468	-15.248
H_2O	-14.220	-0.110	-14.330
H_2	-6.780	-0.120	-6.900

Noted: The energy of a proton and electron pair ($\text{H}^+ + \text{e}^-$) is calculated as: $G(\text{H}^+ + \text{e}^-) = 1/2 G(\text{H}_2)$, and the energy of NO_2^- is obtained by the: $G(\text{NO}_2^- - \text{e}^-) = G(\text{HNO}_2) - 1/2G(\text{H}_2)$.

To address these points, the following changes have been made in the revised version:

- Figure R4 has been incorporated as **Supplementary Figure 13**.
- Figure R5 has been incorporated as **Supplementary Figure 14**.
- Table R1 has been incorporated as **Supplementary Table 24**.
- The discussions of “In addition, the adsorption energies of reactants NO_2 and the corresponding competing species with respect to the applied potential are shown in Supplementary Figure 13. The adsorption energy of *NO_2 is significantly influenced by the applied potential, followed by *CO_2 and *H . Throughout the entire potential range, *NO_2 exhibits the most negative adsorption energy, implying that the active site will be primarily occupied by *NO_2 . The adsorption energy of *NO_2 under different coverages are shown in Supplementary Figure 14, they display the similar trends, and *NO_2 exhibits the more negative adsorption energy at the lower coverage. In the following discussions, we will focus on the lowest coverage of *NO_2 .” have been added in **Page 5**.

and (iii) transport of species at the interface, which is more complex than in gas-phase contexts. The associated impacts need to be discussed, with the results appropriately placed in the context of these limitations. For example, how might the conclusions with respect to temperature effects be affected when considering that higher temperature will further alter the adsorption of species (in the context of the above points)?

Response: Thanks for raising this valuable concern. The temperature effect is indeed important for the adsorption of species and therefore the discussions about this point are necessary. To explore the temperature effect for the adsorption of species, we conducted the coverage curves of adsorbed species with varying temperature. The results are provided below in Figure R6.

Figure R6 Simulated coverage curves of adsorbed species as a function of temperature on Cu(111) (300 to 900K) and Cu(100) (300 to 1000K) under the applied electrode potential of -0.25 and -0.5 V, respectively.

Figure R7 Microkinetic simulations. Turnover frequencies (TOF) per site for urea synthesis on **a** Cu(111) and **d** Cu(100) surfaces as functions of applied electrode potential vs RHE at 300 K and 1 bar. TOFs per site for NH_3 synthesis on **b** Cu(111) and **e** Cu(100) surfaces as a function of applied electrode potential vs RHE at 300 K and 1 bar. TOFs per site for urea synthesis on **c** Cu(111) and **f** Cu(100) surfaces mapped

with pressure (1-100 bar) and temperature (300-600 K and 300-900K, respectively).

An abundant adsorbed intermediate of *NH and *NO_2 on Cu(111) Cu(100) surfaces within the temperature ranging of 300 to 600K and 300 to 900 K can be observed, respectively. This can be attributed to the fact that the RDS on the Cu(111) surface is the coupling of *CO and *NH , while the RDS on the Cu(100) surface is the hydrogenation of *NO_2 . Notably, above the temperature of 600 and 900K, the Cu(111) and Cu(100) surfaces are dominated by other adsorbed species, thus the reaction mechanism will be altered on these surfaces under the higher temperature. This observation aligns with the significant drop in the TOF values of urea synthesis, approaching nearly zero above 600 K and 900 K of Cu(111) and Cu(100) surfaces, as illustrated in Figure R7 of the revised manuscript. To further comprehend the underlying reasons for these observations, we performed the ab initio molecular dynamics (AIMD) simulations for *NO_2 adsorbed Cu(111) and Cu(100). The results shown in the Figure R8 indicate that *NO_2 maintains stable adsorption on the Cu(111) surface at 300 and 500K. However, at elevated temperatures of 700 and 900K, it undergoes dissociation into *NO and *O . In contrast, on the Cu(100) surface, *NO_2 exhibits stability across the simulated temperature range (300 to 900 K). Therefore, we can conclude that the temperature plays a pivotal role in altering the adsorption of species, which will further change the reaction mechanism. At elevated temperatures, the reactants are prone to dissociation, which can potentially hinder the progress of the reaction.

Figure R8 Snapshot of atomic configuration (left panels) and the corresponding variations of energy against the time (right panels) for AIMD simulation of *NO_2 adsorbed **a** Cu(111) and **b** Cu(100) at 300/500/700/900 K for 10 ps with a time step of 2 fs.

To address these points, the following changes have been made in the revised version:

- Figure R6 has been incorporated as **Supplementary Figure 29**.
- Figure R7 has been revised and shown in the manuscript as **Figure 4**.
- Figure R8 has been incorporated as **Supplementary Figure 30**.
- The discussions of “With the increase in temperature, the TOF for urea synthesis on both Cu(111) and Cu(100) surfaces increases, while it remains unaltered by pressure variations. A pronounced decrease in TOF values for urea synthesis is observed, plummeting to nearly zero above 600 K on Cu(111) surfaces and 900 K on Cu(100) surface. Consequently, we restrict our TOF display to the range of 300 to 600 K for Cu(111) and 300 to 900 K for Cu(100) surface. Temperature is a pivotal factor affecting the adsorption of intermediates, thereby affecting the progress of reactions. To explore the temperature effect on species adsorption, the coverage curves of adsorbed species at varying temperature are conducted. Supplementary Fig. 29 reveals the dominant adsorbed intermediates are *NH and *NO_2 on Cu(111) and Cu(100) surfaces within the aforementioned temperature ranges.” and “This can be attributed to the RDS on the Cu(111) surface being the coupling of *CO and *NH , while it is the hydrogenation of *NO_2 on the Cu(100) surface. Above 600 and 900 K, other species dominate adsorption on Cu(111) and Cu(100) surfaces, respectively, leading to alterations in the reaction mechanism. This finding consistent with the results presented in Fig. 4c,f. To deepen our understanding of the mechanisms, ab initio molecular dynamics (AIMD) simulations are conducted for *NO_2 adsorbed Cu(111) and Cu(100). Supplementary Fig. 30 indicates that *NO_2 maintains stable

adsorption on the Cu(111) surface at 300 and 500 K. However, at higher temperatures (700 and 900 K), *NO₂ dissociates into *NO and *O on the Cu(111) surface, while it remains stable across the simulated temperature range (300 to 900 K) on the Cu(100) surface. This underlines the significant impact of temperature on the adsorption behavior of intermediates. Elevating the temperature too high might make reactants dissociation, potentially hindering the progress of the reaction.” have been added in Page 9.

Relatively minor points:

In Figure 1d, it isn't clear what is being conveyed by the “a” and “b” versions of TS2 in type 2. This should be clarified for readers.

Response: Thanks for the reviewer's nice reminder. To clarify the description of “TS 2^a” and “TS 2^b” for Type 2 in the revised version, the statement of “For Type 2 reaction, there are two possibilities. The reaction may proceed spontaneously without crossing a kinetic barrier (TS 2^b), or it require a kinetic barrier as same as Type 1 (TS 2^a).” has been added in the Figure caption of Fig.1.

It is standard to provide the calculated energetics of adsorbed species in a supplementary table.

Response: Thanks for the reviewer's comments. The calculated energetics of adsorbed species have been incorporated as Supplementary Tables 20-23 in the revised version.

Supplementary Table 20 Calculated energy, free energy correction and corresponding Gibbs free energy values for the adsorbed species during the NO₂RR on Cu(111), Cu(110) and Cu(100) surfaces without extra charges added.

Cu(111)	E (eV) (L-H)	ZPE-TS (eV) (L-H)	G (eV) (L-H)	E (eV) (E-R)	ZPE-TS (eV) (E-R)	G (eV) (E-R)
*NO ₂	-201.391	0.325	-201.066	-245.929	2.122	-243.807
*HNO ₂	-204.748	0.547	-204.201	-249.058	2.355	-246.703
*NO	-195.160	0.287	-194.873	-239.844	2.157	-237.687
*NOH	-198.748	0.570	-198.178	-243.474	2.449	-241.025
*N	-188.514	0.248	-188.266	-233.225	2.070	-231.155
*NH	-193.384	0.537	-192.847	-237.950	2.394	-235.556
*NH ₂	-197.593	0.835	-196.758	-242.026	2.635	-239.391
*NH ₃	-198.215	0.914	-197.301	-243.052	2.710	-240.342
Cu(110)	E (eV) (L-H)	ZPE-TS (eV) (L-H)	G (eV) (L-H)	E (eV) (E-R)	ZPE-TS (eV) (E-R)	G (eV) (E-R)
*NO ₂	-193.521	0.325	-193.196	-238.042	2.122	-235.92
*HNO ₂	-196.505	0.547	-195.958	-240.919	2.355	-238.564
*NO	-186.766	0.287	-186.479	-230.979	2.157	-228.822
*NOH	-190.149	0.570	-189.579	-234.497	2.449	-232.048
*N	-180.503	0.248	-180.255	-224.733	2.070	-222.663
*NH	-184.882	0.537	-184.345	-229.223	2.394	-226.829
*NH ₂	-189.617	0.835	-188.782	-233.960	2.635	-231.325
*NH ₃	-190.077	0.914	-189.163	-235.074	2.710	-232.364
Cu(100)	E (eV) (L-H)	ZPE-TS (eV) (L-H)	G (eV) (L-H)	E (eV) (E-R)	ZPE-TS (eV) (E-R)	G (eV) (E-R)
*NO ₂	-196.701	0.325	-196.376	-241.291	2.122	-239.169
*HNO ₂	-199.825	0.547	-199.278	-244.269	2.355	-241.914
*NO	-189.950	0.287	-189.663	-234.548	2.157	-232.391
*NOH	-194.027	0.570	-193.457	-238.730	2.449	-236.281
*N	-184.248	0.248	-184.000	-228.682	2.070	-226.612
*NH	-188.844	0.537	-188.307	-233.223	2.394	-230.829
*NH ₂	-192.838	0.835	-192.003	-237.229	2.635	-234.594
*NH ₃	-193.401	0.914	-192.487	-238.041	2.710	-235.331

Supplementary Table 21 Calculated energy, free energy correction and corresponding Gibbs free energy values for the adsorbed species during the CO₂RR on Cu(111), Cu(110) and Cu(100) surfaces without extra charges added.

Cu(111)	E (eV)	ZPE-TS (eV)	G (eV)	E (eV)	ZPE-TS (eV)	G (eV)
---------	--------	-------------	--------	--------	-------------	--------

	(L-H)	(L-H)	(L-H)	(E-R)	(E-R)	(E-R)
*CO ₂	-204.584	0.334	-204.250	-248.996	2.207	-246.789
*COOH	-207.959	0.642	-207.317	-252.370	2.513	-249.857
*CO	-197.060	0.251	-196.809	-241.754	2.125	-239.629
*HCO	-196.310	0.383	-195.927	-241.150	2.181	-238.969
CO (g)	-192.431	0.094	-192.337			
Cu(110)	E (eV) (L-H)	ZPE-TS (eV) (L-H)	G (eV) (L-H)	E (eV) (E-R)	ZPE-TS (eV) (E-R)	G (eV) (E-R)
*CO ₂	-196.165	0.334	-195.831	-240.464	2.207	-238.257
*COOH	-199.805	0.642	-199.163	-244.196	2.513	-241.683
*CO	-188.981	0.251	-188.730	-233.107	2.125	-230.982
*HCO	-188.182	0.383	-187.799	-233.637	2.181	-231.456
CO (g)	-184.032	0.094	-183.938			
Cu(100)	E (eV) (L-H)	ZPE-TS (eV) (L-H)	G (eV) (L-H)	E (eV) (E-R)	ZPE-TS (eV) (E-R)	G (eV) (E-R)
*CO ₂	-199.561	0.334	-199.227	-243.886	2.207	-241.679
*COOH	-203.040	0.642	-202.398	-247.527	2.513	-245.014
*CO	-192.145	0.251	-191.894	-236.515	2.125	-234.390
*HCO	-191.609	0.383	-191.226	-236.679	2.181	-234.498
CO (g)	-187.469	0.094	-187.375			

Supplementary Table 22 Calculated energy, free energy correction and corresponding Gibbs free energy values for the possible coupling species (the first coupling step) during the urea synthesis on Cu(111), Cu(110) and Cu(100) surfaces without extra charges added.

Cu(111)	E (eV) (*CO ₂)	ZPE-TS (eV) (*CO ₂)	G (eV) (*CO ₂)	E (eV) (*CO)	ZPE-TS (eV) (*CO)	G (eV) (*CO)
*NO	-214.351	0.346	-214.005	-206.510	0.240	-206.270
*NOH	-218.743	0.694	-218.049	-211.018	0.586	-210.432
*N	-208.714	0.327	-208.387	-202.482	0.229	-202.253
*NH	-213.427	0.636	-212.791	-205.703	0.525	-205.178
Cu(110)	E (eV) (*CO ₂)	ZPE-TS (eV) (*CO ₂)	G (eV) (*CO ₂)	E (eV) (*CO)	ZPE-TS (eV) (*CO)	G (eV) (*CO)
*NO ₂	-212.932	0.476	-212.456			
*HNO ₂	-216.837	0.764	-216.073			
*NO	-207.020	0.346	-206.674	-198.625	0.240	-198.385
*NOH	-211.071	0.694	-210.377	-203.113	0.586	-202.527
*N	-201.169	0.327	-200.842	-194.410	0.229	-194.181
*NH	-205.566	0.636	-204.930	-197.835	0.525	-197.310
*NH ₂	-209.933	0.912	-209.021	-201.818	0.844	-200.974

Cu(100)		*NO ₂	*HNO ₂	*NO	*NOH	*N	*NH
*CO ₂	E (eV)	-215.623	-219.627	-209.701	-214.426	-203.938	-208.537
	ZPE-TS (eV)	0.476	0.764	0.346	0.694	0.327	0.636
	G (eV)	-215.147	-218.863	-209.355	-213.732	-203.611	-207.901
*COOH	E (eV)	-219.424	-223.185	-214.338	-218.121	-208.519	-213.079
	ZPE-TS (eV)	0.759	1.055	0.730	1.006	0.660	0.950
	G (eV)	-218.665	-222.13	-213.608	-217.115	-207.859	-212.129
*CO	E (eV)			-202.208	-206.622	-197.598	-201.262
	ZPE-TS (eV)			0.240	0.586	0.229	0.525
	G (eV)			-201.968	-206.036	-197.369	-200.737

Supplementary Table 23 Calculated energy, free energy correction and corresponding Gibbs free energy values for the possible coupling species (the second coupling step) during the urea synthesis on Cu(111) and Cu(100) surfaces without extra charges added.

Cu(111)	E (eV)	ZPE-TS (eV)	G (eV)
*NH-CO-NOH	-224.126	1.031	-223.095

*NH-CO-NH	-218.984	1.053	-217.931
Cu(100)	E (eV)	ZPE-TS (eV)	G (eV)
*N-COOH-N	-213.258	0.678	-212.580
*N-COOH-NH	-218.698	1.032	-217.666
*N-CO-N	-204.428	0.311	-204.117
*N-CO-NH	-209.603	0.689	-208.914
*NH-CO-NH	-214.814	1.053	-213.761

The authors provide a nice introduction as to how urea synthesis from NO_x and CO₂. There seems to be a potential mismatch in scale relative to urea production from N₂, in that substantial sources of NO_x species would be required. Is industrial wastewater capture sufficient relative to the overall scale of urea production? The authors do not need to perform a technoeconomic analysis, but a comment related to this aspect would be appreciated to better contextualize the nice discussion they already provide.

Response: We appreciate the reviewer for his/her positive comments and valuable suggestions. We would like to provide more information about the chosen of NO_x in urea synthesis.

- Industrial wastewater is a viable source of NO_x, while its availability may not be sufficient to satisfy the entire demand for urea. Therefore, additional substantial sources of NO_x such as flue gas from power plants and certain industrial processes, which are typically rich in NO_x, would likely be necessary to supplement the urea production process.
- Our primary goal in discussing the urea synthesis from NO_x and CO₂ was to introduce an alternative method that might improve the efficiency of traditional methods for urea synthesis and additionally facilitate the denitrification of wastewater and balances the perturbed nitrogen cycle. This approach could significantly mitigate environmental impact by repurposing pollutants into beneficial products.
- As reported in many studies (*Nat. Catal.* 2023, 6, 402; *Nat. Nanotechnol.* 2022, 17, 759; and *Nat. Commun.* 2022, 13, 7958), NO_xRR can maintain an over 90% Faradaic efficiency from an industrial wastewater level of 2,000 ppm. Given that NO_xRR is essentially a half-reaction to urea synthesis, there is potential for synthesizing urea using NO_x derived from industrial wastewater levels in the future work.

The authors may wish to correct the spelling of "Eley-Rideal".

Response: Thanks for the reviewer's nice reminder. We are sorry for this typo. The Eley-Rideal has been corrected in the manuscript.

REVIEWER COMMENTS

Reviewer #1 (Remarks to the Author):

The authors have addressed most of my comments, but I still have two main concerns (denoted by *) that require further revision before I believe this manuscript is acceptable for publication in a high impact journal like nature comm.

- Microkinetic model:

The authors implemented a correction to the microkinetic-based TOF analysis by changing the pressure of solvated species into concentrations in the model. The new TOF results look to have the same trend as before but with different magnitudes, yet, the TOF of urea on Cu(111) (Fig. 4a) is still many orders of magnitude lower than that on Cu(100) (Fig. 4d). I'm still confused as to what is responsible for the much lower TOF of urea on Cu(111). To reiterate my confusion with bullet points:

- o The barriers to urea production RDS (at both facets and any potential) are similar enough in magnitude to likely not cause this many orders of magnitude difference in TOF. So the intrinsic barriers of the different reaction paths can not explain the huge difference in urea TOF (as the reviewer suggested in the rebuttal).

- o NO₂RR (produce NH₃) has similar TOF on both Cu(111) and Cu(100). So competition with NO₂RR does not explain the huge difference in urea TOF.

- o As the authors pointed out, the full CO₂RR path was not included in the microkinetic model (*CO is indeed not the final product on Cu electrodes, it is known to further reduce to CH₄ or C-C coupling to C₂ species). So the partial CO₂RR path can not explain the much lower urea TOF on Cu(111) either.

- o The most noticeable difference between the two facets is the RDS based on DRC analysis, i.e., the coupling of *CO-NH on Cu(111) and *NO₂-H on Cu(100). But how this different RDS translate into a huge difference in urea TOF is not clearly elaborated.

*The higher TOF of urea on Cu(100) versus Cu(111) is one of the key results in this manuscript, yet in my opinion, is still rather nebulously explained and rationalized to be fitted for high-impact publication. Please explain further.

- Electric double-layer capacitance analysis:

After reading the revision, I accept the following assertion from the authors:

- o The magnitude of EDL capacitance is reasonable with experiments and other work (20 to 25 $\mu\text{F}/\text{cm}^2$), so I concede that the method used must be getting that somewhat right.

- o The magnitude of capacitance on Cu(111) and Cu(100) is similar, but on Cu(110) it's very different.

The authors have established more clearly that a difference in capacitance seems to correlate with a difference in urea TOF, but still did not convincingly answer *why*. If they assert an "electric double layer-tuned mechanism" in the title and state "tuning the capacitance of the EDL can be a way to optimize the electrochemical synthesis of urea." (line 472-473 in revised manuscript), readers would

expect a convincing physical explanation of why the capacitance correlates with urea TOF. Is it an interfacial electric field effect? Is it an electrode charge density effect? Etc.

Although the authors have added their physical conjectures in the revision (all physically sound), they have not proven any of such concepts. They simply observed that the capacitances are different and the urea TOFs are also different on different Cu facets. Such observations are interesting and perhaps important, but asserting that the EDL capacitance can be tuned to improve urea synthesis based on that alone seems premature.

*I'm only inclined to accept this manuscript for high-impact publication if the underlying physics of the EDL capacitance is convincingly connected to the electrokinetics.

- “Not-double-reference” DFT:

The authors corrected the “double-reference” terminology to “constant-potential,” which is indeed more sensible.

A small additional comment: The method in this paper is not technically constant-potential since the electrode potential is only inferred/determined after running constant-charge calculations, as opposed to setting a target potential and letting the charge fluctuate. I do realize that the literature uses the term constant-potential (or grand canonical) loosely for both constant-charge (change q then determine V) and constant-potential (target V and let q fluctuate) DFT calculations. Therefore, I will not press further but invite the authors to assess their precise terminology in the future.

- VAPSol setting:

The authors laid out their arguments for the VAPSol settings (dielectric constant, Debye length, and surface tension terms) in good detail. I agree that these common settings are widely used in literature, but that does not exempt them from being physically questionable. The Noskov paper (ACS Catal. 2019, 9, 920) actually suggested that a dielectric constant of 80 is unphysical. They suggested using a lower dielectric constant (like 4) and modifying the default electron density cutoff to create a physical solvent and ion distribution. Although doing so indeed lowers the capacitance, it could entirely be the limitation of the implicit solvent to begin with.

Having said so, I'm aware that the impact of changing physical parameters like dielectric constant on the energetics of adsorbed species is not well documented. It is perhaps outside the scope of this manuscript to study the sensitivity of these VAPSol settings. I, therefore, accept using the commonly used settings and again invite the authors to study these parameters further in future work.

Reviewer #2 (Remarks to the Author):

Reviewer #3 (Remarks to the Author):

The authors have presented a substantially revised manuscript addressing the comments from reviewers. I appreciate the authors' efforts to improve their work; I however feel that some of the changes still leave a gap between the results and the conclusions. Most important among these is the treatment of charged species in calculating reaction barriers and adsorption energies. Some changes were additionally somewhat nonresponsive to aspects of my comments, which I further detail below. I believe the logical gaps in this version are too still substantial to justify the conclusions without significant qualification of the results.

(1) I still disagree that the authors' approach to modeling activation energies of E-R steps will give physically meaningful results. The authors reference several papers purported to use a similar approach. However, two of the papers (J. Am. Chem. Soc. 2022, 144, 12800 and J. Am. Chem. Soc. 2021, 143, 9423) appear to be using grand canonical DFT/MD in which the number of electrons fluctuates and allows treatment of species such as H_3O^+ . This H_3O^+ is an important distinction from the H_3O (charge neutral) employed by the authors in this study.

In the present submitted work, the use of charge-neutral H_3O , which has already accepted an electron, creates an unstable initial state for the NEB calculation and thereby lowers the barrier calculated with NEB. This is indeed similar to the published work in Nat. Commun. 2023, 14, 3256, and perhaps also Nat. Commun. 2023, 14, 112 (the authors state they are working with H_3O^+ , though it appears to simply be H_3O). However, these treatments of H_3O^+ as H_3O still suffer from the same problems of underestimating barriers. The most significant issue is the use of these results to conclude that E-R is more favorable than L-H, since no artificially unstable initial state is being used for the L-H mechanism.

This issue is discussed in literature; one such example is work by Janik and collaborators (Catal Today 2017, 288, 63) that nicely summarizes various treatments in its introduction. More recent work includes J Phys Chem A 2022, 126, 7382 or work by Voth's group (J Chem Phys 2021, 154, 194506 or J Am Chem Soc 2021, 143, 18672).

At the very least, the authors need to acknowledge the limitations of their approach to using uncharged H_3O in their system. They should qualify that the barriers estimated with H_3O can be significantly underestimated due to the use of unstable H_3O rather than H_3O^+ , and that this may impact the qualitative mechanistic conclusions of this study.

(2) The authors present additional microkinetic modeling results in Figures R1-R3 to discuss coverage effects. The data are interesting, although the captions of these figures must be improved to include basic details such as which panels correspond to the items in the captions. The resolution after PDF conversion is also somewhat poor and difficult to read.

My question was intended to discuss adsorbate interactions in the context of high coverages. The authors in Figure R2 show very high coverages by adsorbed species. Presumably there are no adsorbate interactions that are considered in these models, which leads to model-predicted coverages on the order of 1 ML by species. In practice, these coverages will be much lower due to repulsion of species, and this will significantly impact calculated reaction energies as well as the calculated rates. (To a point of reviewer 1, I suspect that the high coverages may play a role in the unphysical 50+ orders of magnitude

difference in TOFs still being calculated in this study, which I do not believe can be explained on the energetics alone.)

My second question related to the adsorption of charged species also seems to not have been addressed. The authors present figures R4 and R5, which are interesting data. It is however very surprising to me that NO_2^- would bind more strongly at more negative potentials, given the negative charge on NO_2^- . It seems based on Table R1 that the authors are not accounting for the negative charge on NO_2^- when calculating these adsorption energies, which represents a significant issue as the potential-dependent behavior depends on the treatment of this charge. The quantitative and qualitative behaviors of NO_2^- adsorption will differ significantly from that for NO_2 (neutral) as shown.

Further, my last point about transport of species to the surface is not necessarily related to the temperature effects and adsorption, but rather to the treatment of the complex gradients involved with transporting charged species from the bulk electrolyte to the local environment of the surface. Such modeling is significantly more involved than for comparable gas-phase contexts, and the quantitative nature of results will be affected by choices of appropriate chemical potentials and, e.g., collision theory in the condensed phase context.

Again, at the very least these items need to be discussed as limitations of the present study.

(3) Minor point, but it would be most typical to present the free energies of Supplementary Table 20-20 relative to a uniform reference (e.g., clean surface and reactants), rather than using the arbitrary energy reference provided by the DFT code. Such an approach would enable easier comparison of energies across the various intermediates and surfaces.

Response to the comments of Reviewers

We would like to express our deep gratitude to all the reviewers for their great effort, precious time, and specialized comments, which have greatly helped us to modify the science and story present in our paper. Each review comment has been carefully addressed, together with many additional calculations results newly obtained during this round of revision, all the changes are highlighted in red color in the revised manuscript. Below please find a point-to-point response to the reviewers' comments.

Reviewer #1:

The authors have addressed most of my comments, but I still have two main concerns (denoted by *) that require further revision before I believe this manuscript is acceptable for publication in a high impact journal like nature comm.

- Microkinetic model:

The authors implemented a correction to the microkinetic-based TOF analysis by changing the pressure of solvated species into concentrations in the model. The new TOF results look to have the same trend as before but with different magnitudes, yet, the TOF of urea on Cu(111) (Fig. 4a) is still many orders of magnitude lower than that on Cu(100) (Fig. 4d). I'm still confused as to what is responsible for the much lower TOF of urea on Cu(111). To reiterate my confusion with bullet points:

- The barriers to urea production RDS (at both facets and any potential) are similar enough in magnitude to likely not cause this many orders of magnitude difference in TOF. So the intrinsic barriers of the different reaction paths can not explain the huge difference in urea TOF (as the reviewer suggested in the rebuttal).
- NO₂RR (produce NH₃) has similar TOF on both Cu(111) and Cu(100). So competition with NO₂RR does not explain the huge difference in urea TOF.
- As the authors pointed out, the full CO₂RR path was not included in the microkinetic model (*CO is indeed not the final product on Cu electrodes, it is known to further reduce to CH₄ or C-C coupling to C₂ species). So the partial CO₂RR path can not explain the much lower urea TOF on Cu(111) either.
- The most noticeable difference between the two facets is the RDS based on DRC analysis, i.e., the coupling of *CO-NH on Cu(111) and *NO₂-H on Cu(100). But how this different RDS translate into a huge difference in urea TOF is not clearly elaborated.

*The higher TOF of urea on Cu(100) versus Cu(111) is one of the key results in this manuscript, yet in my opinion, is still rather nebulously explained and rationalized to be fitted for high-impact publication. Please explain further.

Response: This is a really nice and important point indeed deserving a detailed discussion and analysis in our Manuscript and Supplementary Information (SI), which we believe will also be highly useful for numerous related studies in the future. We recognize that our initial manuscript lacked a clear elucidation of this aspect, and we are grateful for the opportunity to clarify this in the revised version.

We first need to acknowledge the limitation in our previous microkinetic model, which ignored proton sourcing from the solution (H₃O⁺) and adsorbed hydrogen (*H) during proton-coupled electron transfer steps. In the revised version, we have optimized our microkinetic modeling for NO₂·RR, CO₂RR, and urea synthesis processes, as detailed in Tables R1-11. These modifications substantially alter the TOF trends and values for urea synthesis and NO₂·RR on the Cu(111) surface, as well as the degree of rate control (DRC) values. The revised modeling shows that both the NO₂·RR and urea synthesis exhibit similar low TOF values on the Cu(111) surface, contrasting with higher TOF values on Cu(100), as depicted in Figure R1. The results can be well explained via their DRC values as follows.

For urea synthesis on Cu(111), TOF is significantly influenced by various intermediates and transition states, as summarized in Table R2. A positive (negative) Xi value signifies that an increase in reaction rate *r* need to further stabilization (destabilization) of the corresponding surface state. The N-

intermediates such as *NO_2 , *HNO_2 , *NO , *NOH , *N , *NH negatively impact the r , while C-intermediates such as *COOH , *CO , and the transition state of $^*CO-NH$ positively influence it. Table R12 and Figure 2 illustrate how these intermediates behave under varying applied electrode potentials. Specifically, intermediates *NO_2 , *HNO_2 , *N , *COOH , *CO , and the transition state $^*CO-NH$ become more destabilized with increasingly negative electrode potentials. Conversely, intermediates *NO and *NOH become more stabilize with increasingly negative electrode potentials. The most notable change occurs in the free energy of *NO_2 , which varies by approximately 1 eV across an applied electrode potential range of 0 to -1.5V, whereas the changes in other intermediates and the $^*CO-NH$ transition state are less than 0.3 eV. These behaviors directly correlated with the TOF trends. As the electrode potential becomes more negative, the increase in TOF can be attributed primarily to the rising free energy of *NO_2 during urea synthesis on the Cu(111) surface. The relationship also suitable for the TOF trends during the NO_2 -RR on the Cu(111) surface (Tables R6,7,12).

On the Cu(100) surface, the TOF values are significantly influenced by the intermediates *NO_2 and *H , as well as the transition state of $^*NO_2-H$. Specifically, the intermediate *NO_2 and the transition state $^*NO_2-H$ positively impact the r , while the intermediate *H has a negative impact. As shown in Table R12 and Supplementary Figure 12, the intermediate *NO_2 and the transition state $^*NO_2-H$ become more destabilized as the electrode potential becomes more negative. In contrast, the intermediate *H stabilizes under more negative electrode potentials. This interplay of stabilization and destabilization leads to an increase in TOF as the applied electrode potential becomes more negative. The trend is consistent with the NO_2 -RR (Figure R1).

A key factor contributing to the distinct TOF values on Cu(111) and Cu(100) surfaces is the difference in their DRC values, as detailed in Tables R2 and R5. On the Cu(111) surface, *NO_2 negatively impacts TOF, while C-intermediates have a positive effect. Conversely, *NO_2 exhibits a positive impact on TOF on the Cu(100) surface. Figure R2 illustrates that *NO_2 is stably adsorbed on both Cu(111) and Cu(100) surfaces. In contrast, *CO_2 is physically adsorbed on these surfaces. This distinct adsorption behavior leads to a high TOF on Cu(100) due to the strong adsorption of *NO_2 , and a lower TOF on Cu(111) owing to the combined effects of strong *NO_2 adsorption and weak *CO_2 adsorption. Moreover, the coverage curves (Figure R2) indicate that *NH and *NO_2 are the predominant surface-covering species during the urea synthesis on Cu(111) and Cu(100), respectively. The *NH negatively impacts the reaction rate on Cu(111), while *NO_2 positively influences it on Cu(100) (Tables R2,R5). This further results in the distinct TOF trends on Cu(111) and Cu(100) surfaces.

To address these points, the following changes have been made in the revised version:

- Figure R1 has been revised and shown in the manuscript as **Figure 4**.
- Figure R2 has been revised and shown in the SI as **Supplementary Figure 29**.
- Figure R3 has been revised and shown in the SI as **Supplementary Figure 13**.
- Tables R1-11 have been incorporated as **Supplementary Tables 13-23**.
- Table R12 has been incorporated as **Supplementary Table 28**.
- The discussions of “To elucidate the huge differences in TOF on Cu(111) and Cu(100) surfaces during urea synthesis, a comprehensive degree of rate control (DRC) analysis for all intermediates and transition states is conducted, as detailed in Supplementary Tables 14,17. A positive (negative) ξ value signifies that an increase in reaction rate r need to further stabilization (destabilization) of the corresponding surface state. On the Cu(111) surface, the *NO_2 adversely affects TOF, while C-intermediates enhance it. Conversely, *NO_2 exhibits a positive impact on TOF on the Cu(100) surface. Supplementary Fig. 13 illustrates that *NO_2 can be stably adsorbed on both Cu(111) and Cu(100) surfaces while *CO_2 is physically adsorbed. This distinct adsorption behavior leads to a higher TOF on Cu(100) than Cu(111). Moreover, the coverage curves (Supplementary Fig. 29) indicate that *NH and *NO_2 are the predominant surface-covering species during the urea synthesis on Cu(111) and Cu(100), respectively. The *NH negatively impacts the reaction rate on Cu(111), while *NO_2 positively influences it on Cu(100) (Supplementary Tables 14,17). This further results in a lower TOF for Cu(111) and a higher TOF for Cu(100).” have been added in **Page 9** in the revised manuscript.
- The discussions of “For urea synthesis on Cu(111) and Cu(100) surfaces, we have identified significant impacts from various intermediates and transition states on turnover frequency (TOF), as summarized in Supplementary Table 14. For the Cu(111) surface, N-intermediates such as *NO_2 , *HNO_2 , *NO , *NOH , *N , and *NH adversely affect the reaction rate, while C-intermediates like *COOH , *CO , and the transition state of $^*CO-NH$ positively influence it. The

behaviors of these intermediates and transition state under various applied electrode potentials are detailed in Supplementary Table 28 and Fig. 2 in the manuscript. Notably, intermediates *NO₂, *HNO₂, *N, *COOH, *CO, and the transition state *CO-NH become increasingly destabilized with more negative electrode potentials. In contrast, intermediates *NO and *NOH stabilize under such conditions. The most significant variation is observed in the free energy of *NO₂, which alters by approximately 1 eV across applied electrode potential range of 0 to -1.5 V. In comparison, the changes in other intermediates and the transition state *CO-NH are less pronounced, remaining below 0.3 eV. These behaviors correlate directly with the observed TOF trends, where a more negative electrode potential leads to an increase in TOF, primarily due to the rising free energy of *NO₂ during urea synthesis on the Cu(111) surface. The relationship is also consistent with the TOF trends during the NO₂-RR on the Cu(111) surface, as demonstrated in Supplementary Tables 18,19,28.

On the Cu(100) surface, the TOF values are significantly influenced by intermediates *NO₂, *H, and the transition state *NO₂-H. Here, intermediate *NO₂ and the transition state *NO₂-H positively impact the reaction rate, while the *H intermediate has a negative effect. As shown in Supplementary Table 28 and Supplementary Fig.12, both the intermediate *NO₂ and the transition state *NO₂-H become more destabilized as the electrode potential becomes more negative, while the intermediate *H stabilizes under these conditions. This dynamic between stabilization and destabilization leads to an increase in TOF as the electrode potential being more negative, a trend consistent with the NO₂-RR on the Cu(100) surface, as illustrated in Fig. 4 in the manuscript." have been added in Page 24 in the revised SI.

Table R1 Microkinetic equations of urea synthesis process on the Cu(111) surface. * denotes active site. θ_i and P_i represent the coverage and pressure of reactants respectively. C_i is the concentrations of aqueous-phase species.

Reaction steps	Reaction rate (r)
$\text{NO}_2^- + * = \text{*NO}_2 + \text{e}^-$	$\theta(\text{*NO}_2) = K_1 C_{\text{NO}_2} \theta(*)$
$\text{*NO}_2 + \text{H}^+ + \text{e}^- = \text{*HNO}_2$	$r_2 = k_2 \theta(\text{*NO}_2) C_{\text{H}^+} - k_{-2} \theta(\text{*HNO}_2)$
$\text{*HNO}_2 + \text{H}^+ + \text{e}^- = \text{*NO} + \text{H}_2\text{O}$	$r_3 = k_3 \theta(\text{*HNO}_2) C_{\text{H}^+} - k_{-3} \theta(\text{*NO}) C_{\text{H}_2\text{O}}$
$\text{*NO} + \text{H}^+ + \text{e}^- = \text{*NOH}$	$r_4 = k_4 \theta(\text{*NO}) C_{\text{H}^+} - k_{-4} \theta(\text{*NOH})$
$\text{*NOH} + \text{H}^+ + \text{e}^- = \text{*N} + \text{H}_2\text{O}$	$r_5 = k_5 \theta(\text{*NOH}) C_{\text{H}^+} - k_{-5} \theta(\text{*N}) C_{\text{H}_2\text{O}}$
$\text{*N} + \text{H}^+ + \text{e}^- = \text{*NH}$	$r_6 = k_6 \theta(\text{*N}) C_{\text{H}^+} - k_{-6} \theta(\text{*NH})$
$\text{CO}_2 + * + \text{H}^+ + \text{e}^- = \text{*COOH}$	$r_7 = k_7 \theta(*) C_{\text{H}^+} P_{\text{CO}_2} - k_{-7} \theta(\text{*COOH})$
$\text{*COOH} + \text{H}^+ + \text{e}^- = \text{*CO} + \text{H}_2\text{O}$	$r_8 = k_8 \theta(\text{*COOH}) C_{\text{H}^+} - k_{-8} \theta(\text{*CO}) C_{\text{H}_2\text{O}}$
$\text{*NH} + \text{*CO} = \text{*CO-NH} + *$ (RDS)	$r_9 = k_9 \theta(\text{*NH}) \theta(\text{*CO}) - k_{-9} \theta(\text{*CO-NH}) \theta(*)$
$\text{*CO-NH} + \text{*NH} = \text{*NH-CO-NH} + *$	$r_{10} = k_{10} \theta(\text{*CO-NH}) \theta(\text{*NH}) - k_{-10} \theta(\text{*NH-CO-NH}) \theta(*)$
$\text{*NH-CO-NH} + \text{*H} = \text{*NH-CO-NH}_2 + *$	$r_{11} = k_{11} \theta(\text{*NH-CO-NH}) \theta(\text{*H}) - k_{-11} \theta(\text{*NH-CO-NH}_2) \theta(*)$
$\text{*NH-CO-NH}_2 + \text{H}^+ + \text{e}^- = \text{CO}(\text{NH}_2)_2 + *$	$r_{12} = k_{12} \theta(\text{*NH-CO-NH}_2) C_{\text{H}^+} - k_{-12} C_{\text{CO}(\text{NH}_2)_2} \theta(*)$
$\text{H}^+ + \text{e}^- + * = \text{*H}$	$\theta(\text{*H}) = K_{13} C_{\text{H}^+} \theta(*)$

Table R2 Degrees for rate control of all intermediates and transition states of urea synthesis process on the Cu(111) surface at 300 K under the applied electrode potential of -1.50 V.

Species (intermediates)	X_i	Species (transition-states)	X_i
*NO ₂	-1	*-NO ₂	0
*HNO ₂	-1	NO ₂ -H	0
*NO	-1	NOOH-H	0
*NOH	-1	NO-H	0
*N	-1	NOH-H	0
*NH	-1	N-H	0
*COOH	1	COO-H	0
*CO	1	COOH-H	0
*CO-NH	0	CO-NH	1
*NH-CO-NH	0	NHCO-NH	0
*NH-CO-NH ₂	9.19×10^{-61}	NHCONH-H	0

*	3.15×10^{-60}	*-CO(NH ₂) ₂	0
*H	-3.64×10^{-59}	*-H	0

Table R3 Microkinetic equations of urea synthesis process on the Cu(100) surface under the high applied electrode potential (0.00 to -0.42 V). * indicates active site. θ_i and P_i represent the coverage and pressure of reactants respectively. C_i is the concentrations of aqueous-phase species.

$\text{NO}_2^- + * = *\text{NO}_2 + e^-$	$\theta(*\text{NO}_2) = K_1 C_{\text{NO}_2^-} \theta(*)$
$*\text{NO}_2 + \text{H}^+ + e^- = *\text{HNO}_2$	$r_2 = k_2 \theta(*\text{NO}_2) C_{\text{H}^+} - k_{-2} \theta(*\text{HNO}_2)$
$*\text{HNO}_2 + \text{H}^+ + e^- = *\text{NO} + \text{H}_2\text{O}$	$r_3 = k_3 \theta(*\text{HNO}_2) C_{\text{H}^+} - k_{-3} \theta(*\text{NO}) C_{\text{H}_2\text{O}}$
$*\text{NO} + \text{H}^+ + e^- = *\text{NOH}$	$r_4 = k_4 \theta(*\text{NO}) C_{\text{H}^+} - k_{-4} \theta(*\text{NOH})$
$*\text{NOH} + \text{H}^+ + e^- = *\text{N} + \text{H}_2\text{O}$	$r_5 = k_5 \theta(*\text{NOH}) C_{\text{H}^+} - k_{-5} \theta(*\text{N}) C_{\text{H}_2\text{O}}$
$*\text{N} + *\text{H} = *\text{NH} + *$	$r_6 = k_6 \theta(*\text{N}) \theta(*\text{H}) - k_{-6} \theta(*\text{NH}) \theta(*)$
$\text{CO}_2 + *\text{H} = *\text{COOH}$	$r_7 = k_7 \theta(*\text{H}) P_{\text{CO}_2} - k_{-7} \theta(*\text{COOH})$
$*\text{COOH} + \text{H}^+ + e^- = *\text{CO} + \text{H}_2\text{O}$	$r_8 = k_8 \theta(*\text{COOH}) C_{\text{H}^+} - k_{-8} \theta(*\text{CO}) C_{\text{H}_2\text{O}}$
$*\text{N} + *\text{CO} = *\text{CO-N} + *$	$r_9 = k_9 \theta(*\text{N}) \theta(*\text{CO}) - k_{-9} \theta(*\text{CO-N}) \theta(*)$
$*\text{CO-N} + \text{H}^+ + e^- = *\text{CO-NH}$	$r_{10} = k_{10} \theta(*\text{CO-N}) C_{\text{H}^+} - k_{-10} \theta(*\text{CO-NH})$
$*\text{CO-NH} + *\text{NH} = *\text{NH-CO-NH} + *$ (RDS)	$r_{11} = k_{11} \theta(*\text{CO-NH}) \theta(*\text{NH}) - k_{-11} \theta(*\text{NH-CO-NH}) \theta(*)$
$*\text{NH-CO-NH} + \text{H}^+ + e^- = *\text{NH-CO-NH}_2$	$r_{12} = k_{12} \theta(*\text{NH-CO-NH}) C_{\text{H}^+} - k_{-12} \theta(*\text{NH-CO-NH}_2)$
$*\text{NH-CO-NH}_2 + \text{H}^+ + e^- = \text{CO}(\text{NH}_2)_2 + *$	$r_{13} = k_{13} \theta(*\text{NH-CO-NH}_2) C_{\text{H}^+} - k_{-13} C_{\text{CO}(\text{NH}_2)_2} \theta(*)$
$\text{H}^+ + e^- + * = *\text{H}$	$\theta(*\text{H}) = K_{14} C_{\text{H}^+} \theta(*)$

Table R4 Microkinetic equations of urea synthesis process on the Cu(100) surface under the low applied electrode potential (-0.42 to -1.30 V). * indicates active site. θ_i and P_i represent the coverage and pressure of reactants respectively. C_i is the concentrations of aqueous-phase species.

$\text{NO}_2^- + * = *\text{NO}_2 + e^-$	$\theta(*\text{NO}_2) = K_1 C_{\text{NO}_2^-} \theta(*)$
$*\text{NO}_2 + \text{H}^+ + e^- = *\text{HNO}_2$ (RDS)	$r_2 = k_2 \theta(*\text{NO}_2) C_{\text{H}^+} - k_{-2} \theta(*\text{HNO}_2)$
$*\text{HNO}_2 + \text{H}^+ + e^- = *\text{NO} + \text{H}_2\text{O}$	$r_3 = k_3 \theta(*\text{HNO}_2) C_{\text{H}^+} - k_{-3} \theta(*\text{NO}) C_{\text{H}_2\text{O}}$
$*\text{NO} + \text{H}^+ + e^- = *\text{NOH}$	$r_4 = k_4 \theta(*\text{NO}) C_{\text{H}^+} - k_{-4} \theta(*\text{NOH})$
$*\text{NOH} + \text{H}^+ + e^- = *\text{N} + \text{H}_2\text{O}$	$r_5 = k_5 \theta(*\text{NOH}) C_{\text{H}^+} - k_{-5} \theta(*\text{N}) C_{\text{H}_2\text{O}}$
$*\text{N} + *\text{H} = *\text{NH} + *$	$r_6 = k_6 \theta(*\text{N}) \theta(*\text{H}) - k_{-6} \theta(*\text{NH}) \theta(*)$
$*\text{N} + \text{CO}_2 = *\text{CO}_2\text{-N}$	$r_7 = k_7 \theta(*\text{N}) P_{\text{CO}_2} - k_{-7} \theta(*\text{CO}_2\text{-N})$
$*\text{CO}_2\text{-N} + \text{H}^+ + e^- = *\text{COOH-N}$	$r_8 = k_8 \theta(*\text{CO}_2\text{-N}) C_{\text{H}^+} - k_{-8} \theta(*\text{COOH-N})$
$*\text{COOH-N} + \text{H}^+ + e^- = *\text{CO-N} + \text{H}_2\text{O}$	$r_9 = k_9 \theta(*\text{COOH-N}) C_{\text{H}^+} - k_{-9} \theta(*\text{CO-N}) C_{\text{H}_2\text{O}}$
$*\text{CO-N} + \text{H}^+ + e^- = *\text{CO-NH}$	$r_{10} = k_{10} \theta(*\text{CO-N}) C_{\text{H}^+} - k_{-10} \theta(*\text{CO-NH})$
$*\text{CO-NH} + *\text{NH} = *\text{NH-CO-NH} + *$	$r_{11} = k_{11} \theta(*\text{CO-NH}) \theta(*\text{NH}) - k_{-11} \theta(*\text{NH-CO-NH}) \theta(*)$
$*\text{NH-CO-NH} + \text{H}^+ + e^- = *\text{NH-CO-NH}_2$	$r_{12} = k_{12} \theta(*\text{NH-CO-NH}) C_{\text{H}^+} - k_{-12} \theta(*\text{NH-CO-NH}_2)$
$*\text{NH-CO-NH}_2 + \text{H}^+ + e^- = \text{CO}(\text{NH}_2)_2 + *$	$r_{13} = k_{13} \theta(*\text{NH-CO-NH}_2) C_{\text{H}^+} - k_{-13} C_{\text{CO}(\text{NH}_2)_2} \theta(*)$
$\text{H}^+ + e^- + * = *\text{H}$	$\theta(*\text{H}) = K_{14} C_{\text{H}^+} \theta(*)$

Table R5 Degrees for rate control of all intermediates and transition states of urea synthesis process on the Cu(100) surface at 300 K under the applied electrode potential of -0.50 V.

Species (intermediates)	X_i	Species (transition-states)	X_i
*NO ₂	2.17×10^{-23}	*-NO ₂	0
*HNO ₂	0	NO ₂ -H	1
*NO	0	NOOH-H	0
*NOH	0	NO-H	0
*N	0	NOH-H	0
*NH	3.12×10^{-32}	N-H	0
*CO ₂ -N	-3.12×10^{-32}	CO ₂ -N	0
*COOH-N	-3.12×10^{-32}	NCOO-H	0
*CO-N	-3.12×10^{-32}	CO-N	0
*CO-NH	-3.12×10^{-32}	CON-H	0
*NH-CO-NH	3.12×10^{-32}	NHCO-NH	0

*NH-CO-NH ₂	2.33×10 ⁻²⁷	NHCONH-H	0
*	2.33×10 ⁻²⁷	*-CO(NH ₂) ₂	0
*H	-1.02×10 ⁻²³	*-H	0

Table R6 Microkinetic equations of NO₂-RR process on Cu(111) surface. * indicates active site. θ_i and P_i represent the coverage and pressure of reactants respectively. C_i is the concentrations of aqueous-phase species.

NO ₂ ⁻ + * = *NO ₂ + e ⁻	θ(*NO ₂) = K ₁ C _{NO₂⁻} θ(*)
*NO ₂ + H ⁺ + e ⁻ = *HNO ₂	r ₂ = k ₂ θ(*NO ₂)C _{H⁺} - k ₋₂ θ(*HNO ₂)
*HNO ₂ + H ⁺ + e ⁻ = *NO + H ₂ O	r ₃ = k ₃ θ(*HNO ₂)C _{H⁺} - k ₋₃ θ(*NO)C _{H₂O}
*NO + H ⁺ + e ⁻ = *NOH	r ₄ = k ₄ θ(*NO)C _{H⁺} - k ₋₄ θ(*NOH)
*NOH + H ⁺ + e ⁻ = *N + H ₂ O	r ₅ = k ₅ θ(*NOH)C _{H⁺} - k ₋₅ θ(*N)C _{H₂O}
*N + H ⁺ + e ⁻ = *NH	r ₆ = k ₆ θ(*N)C _{H⁺} - k ₋₆ θ(*NH)
*NH + *H = *NH ₂ + * (RDS)	r ₇ = k ₇ θ(*NH)θ(*H) - k ₋₇ θ(*NH ₂)θ(*)
*NH ₂ + H ⁺ + e ⁻ = * + NH ₃	r ₈ = k ₈ θ(*NH ₂)C _{H⁺} - k ₋₈ P _{NH₃} θ(*)
H ⁺ + e ⁻ + * = *H	θ(*H) = K ₉ C _{H⁺} θ(*)

Table R7 Degrees for rate control of all intermediates and transition states of NO₂-RR process on the Cu(111) surface at 300 K under the applied electrode potential of -1.50 V.

Species (intermediates)	X _i	Species (transition-states)	X _i
*NO ₂	-1	*-NO ₂	0
*HNO ₂	-1	NO ₂ -H	0
*NO	-1	NOOH-H	0
*NOH	-1	NO-H	0
*N	-1	NOH-H	0
*NH	-1	N-H	0
*NH ₂	2.96×10 ⁻⁷¹	NH-H	1
*	2.96×10 ⁻⁷¹	*-NH ₃	0
*H	2.96×10 ⁻⁷¹	*-H	0

Table R8 Microkinetic equations of NO₂-RR process on Cu(100) surface. * indicates active site. θ_i and P_i represent the coverage and pressure of reactants respectively. C_i is the concentrations of aqueous-phase species.

NO ₂ ⁻ + * = *NO ₂ + e ⁻	θ(*NO ₂) = K ₁ C _{NO₂⁻} θ(*)
*NO ₂ + H ⁺ + e ⁻ = *HNO ₂	r ₂ = k ₂ θ(*NO ₂)C _{H⁺} - k ₋₂ θ(*HNO ₂)
*HNO ₂ + H ⁺ + e ⁻ = *NO + H ₂ O	r ₃ = k ₃ θ(*HNO ₂)C _{H⁺} - k ₋₃ θ(*NO)C _{H₂O}
*NO + H ⁺ + e ⁻ = *NOH	r ₄ = k ₄ θ(*NO)C _{H⁺} - k ₋₄ θ(*NOH)
*NOH + H ⁺ + e ⁻ = *N + H ₂ O	r ₅ = k ₅ θ(*NOH)C _{H⁺} - k ₋₅ θ(*N)C _{H₂O}
*N + *H = *NH + *	r ₆ = k ₆ θ(*N)θ(*H) - k ₋₆ θ(*NH)θ(*)
*NH + H ⁺ + e ⁻ = *NH ₂ (RDS)	r ₇ = k ₇ θ(*NH)C _{H⁺} - k ₋₇ θ(*NH ₂)
*NH ₂ + H ⁺ + e ⁻ = * + NH ₃	r ₈ = k ₈ θ(*NH ₂)C _{H⁺} - k ₋₈ P _{NH₃} θ(*)
H ⁺ + e ⁻ + * = *H	θ(*H) = K ₉ C _{H⁺} θ(*)

Table R9 Degrees for rate control of all intermediates and transition states of NO₂-RR process on the Cu(100) surface at 300 K under the applied electrode potential of -0.75 V.

Species (intermediates)	X _i	Species (transition-states)	X _i
*NO ₂	0	*-NO ₂	0
*HNO ₂	5.97×10 ⁻⁷²	NO ₂ -H	0
*NO	5.97×10 ⁻⁷²	NOOH-H	0
*NOH	1.59×10 ⁻⁵⁰	NO-H	0
*N	1.25×10 ⁻⁴²	NOH-H	0
*NH	6.24×10 ⁻¹⁸	N-H	0

*NH ₂	0	NH-H	1
*	0	*-NH ₃	0
*H	6.24×10 ⁻¹⁸	*-H	0

Table R10 Microkinetic equations of CO₂RR process on Cu(111) surface. * indicates active site. θ_i and P_i represent the coverage and pressure of reactants respectively. C_i is the concentrations of aqueous-phase species.

$\text{CO}_2 + * = *\text{CO}_2$	$\theta(*\text{CO}_2) = K_1 P_{\text{CO}_2} \theta(*)$
$*\text{CO}_2 + \text{H}^+ + \text{e}^- = *\text{COOH}$	$r_2 = k_2 \theta(*\text{CO}_2) C_{\text{H}^+} - k_{-2} \theta(*\text{COOH})$
$*\text{COOH} + \text{H}^+ + \text{e}^- = *\text{CO} + \text{H}_2\text{O}$	$r_3 = k_3 \theta(*\text{COOH}) C_{\text{H}^+} - k_{-3} \theta(*\text{CO}) C_{\text{H}_2\text{O}}$
$*\text{CO} = * + \text{CO}$ (RDS)	$r_4 = k_4 \theta(*\text{CO}) - k_{-4} \theta(*) P_{\text{CO}}$

Table R11 Microkinetic equations of CO₂RR process on Cu(100) surface. * indicates active site. θ_i and P_i represent the coverage and pressure of reactants respectively. C_i is the concentrations of aqueous-phase species.

$\text{CO}_2 + * = *\text{CO}_2$	$\theta(*\text{CO}_2) = K_1 P_{\text{CO}_2} \theta(*)$
$*\text{CO}_2 + *\text{H} = *\text{COOH} + *$	$r_2 = k_2 \theta(*\text{CO}_2) \theta(*\text{H}) - k_{-2} \theta(*\text{COOH}) \theta(*)$
$*\text{COOH} + \text{H}^+ + \text{e}^- = *\text{CO} + \text{H}_2\text{O}$	$r_3 = k_3 \theta(*\text{COOH}) C_{\text{H}^+} - k_{-3} \theta(*\text{CO}) C_{\text{H}_2\text{O}}$
$*\text{CO} = * + \text{CO}$ (RDS)	$r_4 = k_4 \theta(*\text{CO}) - k_{-4} \theta(*) P_{\text{CO}}$
$\text{H}^+ + \text{e}^- + * = *\text{H}$	$\theta(*\text{H}) = K_5 C_{\text{H}^+} \theta(*)$

Table R12 Calculated Gibbs free energy values (ΔG) for reaction steps during the urea synthesis on Cu(111) and Cu(100) surfaces with respect to the applied electrode potentials (U vs RHE), corresponding to Tables R1,R3,R4. The unit is eV.

Cu(111)	-1.50 V	-1.25 V	-1.00 V	-0.75 V	-0.50 V	-0.25 V	0.00 V
$\Delta G(*\text{NO}_2)$	-0.559	-0.611	-0.684	-0.841	-1.087	-1.335	-1.515
$\Delta G(*\text{NO}_2\text{H})$	0.351	0.311	0.271	0.228	0.182	0.132	0.079
$\Delta G(*\text{NO})$	-2.532	-2.546	-2.560	-2.529	-2.436	-2.331	-2.215
$\Delta G(*\text{NOH})$	-0.304	-0.298	-0.292	-0.283	-0.230	-0.171	-0.116
$\Delta G(*\text{N})$	-1.463	-1.501	-1.538	-1.551	-1.560	-1.569	-1.578
$\Delta G(*\text{NH})$	-1.315	-1.332	-1.349	-1.350	-1.337	-1.321	-1.300
$\Delta G(*\text{COOH})$	0.064	0.004	-0.055	-0.070	-0.069	-0.058	-0.037
$\Delta G(*\text{CO})$	-0.896	-0.902	-0.908	-0.919	-0.937	-0.961	-0.990
$\Delta G(*\text{CO-NH})$	0.041	0.024	0.008	-0.009	-0.012	-0.004	0.005
$\Delta G(*\text{NH-CO-NH})$	-1.212	-1.228	-1.243	-1.258	-1.264	-1.262	-1.258
$\Delta G(*\text{NH-CO-NH}_2)$	-0.227	-0.245	-0.264	-0.288	-0.381	-0.493	-0.601
$\Delta G(*\text{NH}_2\text{-CO-NH}_2)$	-0.533	-0.535	-0.541	-0.551	-0.556	-0.554	-0.547
$\Delta G(*\text{H})$	-0.290	-0.320	-0.350	-0.370	-0.302	-0.196	-0.09

Cu(100)	-1.50 V	-1.25 V	-1.00 V	-0.75 V	-0.50 V	-0.25 V	0.00 V
$\Delta G(*\text{NO}_2)$	-0.994	-1.006	-1.034	-1.167	-1.425	-1.702	-1.915
$\Delta G(*\text{NO}_2\text{H})$	0.539	0.476	0.414	0.312	0.198	0.103	0.026
$\Delta G(*\text{NO})$	-2.252	-2.333	-2.414	-2.435	-2.393	-2.321	-2.220
$\Delta G(*\text{NOH})$	-0.911	-0.897	-0.884	-0.867	-0.829	-0.801	-0.783
$\Delta G(*\text{N})$	-1.828	-1.846	-1.865	-1.866	-1.802	-1.736	-1.672
$\Delta G(*\text{NH})$	-0.919	-0.916	-0.913	-0.910	-0.918	-0.935	-0.957
$\Delta G(*\text{CO}_2\text{-N})$	0.092	0.036	-0.021	-0.078	-0.104		
$\Delta G(*\text{COOH-N})$	-1.339	-1.321	-1.302	-1.283	-1.262		
$\Delta G(*\text{COOH})$						0.179	0.138
$\Delta G(*\text{CO})$						-0.986	-0.993
$\Delta G(*\text{CO-N})$	-1.237	-1.215	-1.192	-1.169	-1.113	-1.058	-1.026

$\Delta G(^*CO-NH)$	-0.720	-0.773	-0.826	-0.888	-0.932	-0.951	-0.945
$\Delta G(^*NH-CO-NH)$	-1.470	-1.475	-1.480	-1.485	-1.471	-1.447	-1.424
$\Delta G(^*NH-CO-NH_2)$	-0.154	-0.179	-0.205	-0.242	-0.274	-0.299	-0.319
$\Delta G(^*NH_2-CO-NH_2)$	-0.578	-0.606	-0.635	-0.668	-0.696	-0.720	-0.740
$\Delta G(^*H)$	-0.501	-0.516	-0.531	-0.534	-0.491	-0.455	-0.426

Figure R1 Microkinetic simulations. Turnover frequencies (TOF) per site for urea synthesis on **a** Cu(111) and **d** Cu(100) surfaces as functions of applied electrode potential vs RHE at 300 K and 1 bar. TOFs per site for NH₃ synthesis on **b** Cu(111) and **e** Cu(100) surfaces as a function of applied electrode potential vs RHE at 300 K and 1 bar. TOFs per site for urea synthesis on **c** Cu(111) and **f** Cu(100) surfaces mapped with pressure (1-100 bar) and temperature (300-700 K and 300-900K, respectively).

Figure R2 Simulated coverage curves of adsorbed species for urea synthesis as a function of temperature on Cu(111) (300 to 800K) and Cu(100) (300 to 1000K) under the applied electrode potential of -1.50 and -0.50 V, respectively. The pH is set as 8.3 for urea synthesis in accordance with the experimental environments.

Figure R3 Adsorption energies of $^*\text{NO}_2$, $^*\text{CO}_2$, and $^*\text{H}$ as a function of U/RHE (from 0.50 to -1.50 V) on **a** Cu(111), **b** Cu(110) and **c** Cu(100) surfaces. **d** Excess electrons of NO_2^- in vacuum. The pH is set as 8.3 for urea synthesis in accordance with the experimental environments.

- Electric double-layer capacitance analysis:

After reading the revision, I accept the following assertion from the authors:

- The magnitude of EDL capacitance is reasonable with experiments and other work (20 to 25 $\mu\text{F}/\text{cm}^2$), so I concede that the method used must be getting that somewhat right.
- The magnitude of capacitance on Cu(111) and Cu(100) is similar, but on Cu(110) it's very different.

The authors have established more clearly that a difference in capacitance seems to correlate with a difference in urea TOF, but still did not convincingly answer **why**. If they assert an “electric double layer-tuned mechanism” in the title and state “tuning the capacitance of the EDL can be a way to optimize the electrochemical synthesis of urea.” (line 472-473 in revised manuscript), readers would expect a convincing physical explanation of why the capacitance correlates with urea TOF. Is it an interfacial electric field effect? Is it an electrode charge density effect? Etc.

Although the authors have added their physical conjectures in the revision (all physically sound), they have not proven any of such concepts. They simply observed that the capacitances are different and the urea TOFs are also different on different Cu facets. Such observations are interesting and perhaps important, but asserting that the EDL capacitance can be tuned to improve urea synthesis based on that alone seems premature.

*I'm only inclined to accept this manuscript for high-impact publication if the underlying physics of the EDL capacitance is convincingly connected to the electrokinetics.

Response: Thanks for raising the valuable concern. We are sorry for that the underlying physics of the EDL capacitance connected to the electrokinetic are not clearly stated in previous version. We would like to better illustrate the physical origin of it in the revised version.

It has been reported that kinetics are strongly dependent on the physicochemical properties of electrochemical interfaces, including the nature of the electrode material and the double layer (*ACS Catal.* 2023, 13, 11080). It underpins our approach to elucidating capacitance-TOF correlations. To better illustrate the physical origin of the relationship between EDL capacitance and electrokinetic, we studied the surface-charge density (σ) of intermediates adsorbed surfaces. Our study of σ demonstrates it could be an appropriate descriptor for electrostatic effects of the double layer on electrokinetic, since it describes the variations of the interfacial field local to the reaction site (*Nat. Commun.* 2020, 11, 33; *ACS Catal.* 2020, 10, 7826).

Figure R4 presents the changes in σ as a function of potential for surfaces with adsorbed C- and N-intermediates. The reaction sites on C- and N-intermediates adsorbed Cu(111) and Cu(100) surfaces, characterized by positive σ , show identical interfacial fields. This uniformity indicates that C- and N-intermediates can coexist on these surfaces, thereby facilitating the coupling steps. In contrast, C- and N-intermediates adsorbed Cu(110) surface show opposite interfacial fields, hindering coupling steps. Especially, the negative σ of $^*\text{CO}_2$ adsorbed Cu(110) surface could be responsible for the decrease in EDL capacitance of C-intermediate adsorbed Cu(110) surface (Figure 5).

The change in σ for intermediates adsorbed Cu(111) and Cu(100) surfaces during the urea synthesis as a function of electrode potential are shown in Figure R5. The σ follows the order Cu(100) > Cu(111), which is consistent with the order of EDL capacitance and reaction rate. To capture the influence of double layer charging on electrokinetic, we investigated the σ -dependent kinetic barriers as shown in Figure R5d,e. In the case of the RDS for urea synthesis on both Cu(111) and Cu(100) surfaces, there is an increase in the kinetic barrier as σ decreases. Briefly, we find a novel strategy to design efficient electrocatalysts at the molecular level. By regulating the EDL with the larger capacitances, we can enhance the escape of electrons during the urea synthesis on Cu electrode surface, thereby decreasing the kinetic barrier of reaction steps and eventually result in an increased reaction rate.

To incorporate these insights into the work, below three modifications have been added in the revised version:

- ◆ Figure R4 has been incorporated as Supplementary Figure 34.
- ◆ Figure R5 has been incorporated as Supplementary Figure 35.
- ◆ The discussions of “Notably, the EDL capacitances for intermediates involved in the $\text{NO}_2\text{-RR}$ and CO_2RR are comparable on Cu(111) and Cu(100) surfaces. However, Cu(110) surface presents substantial difference in capacitances between the $\text{NO}_2\text{-RR}$ and CO_2RR processes. This suggests that the coupling behaviors of N/C-intermediates on Cu(111) and Cu(100) surfaces might be enhanced, owing to their resembling physicochemical attributes and interfacial behaviors. However, Cu(110) surface might hinder the interaction of N-intermediate with C-intermediate due to the distinctions in capacitances. To better illustrate the physical origin of the relationship between EDL capacitance and electrokinetic, we studied the surface-charge density (σ) of intermediates adsorbed surfaces. σ could be an appropriate descriptor for electrostatic effects of the double layer on electrokinetic, since it describes the variations of the interfacial field local to the reaction site.^{50,51}

Supplementary Fig. 34 presents the changes in σ for surfaces with adsorbed C- and N-intermediates. The reaction sites on C- and N-intermediates adsorbed Cu(111) and Cu(100) surfaces, characterized by positive σ , show identical interfacial fields. This uniformity indicates that C- and N-intermediates can coexist on these surfaces, thereby facilitating the coupling steps. In contrast, C- and N-intermediates adsorbed Cu(110) surface shows opposite interfacial fields, hindering coupling steps. Especially, the negative σ of $^*\text{CO}_2$ adsorbed Cu(110) surface could be responsible for the decrease in EDL capacitance of C-intermediate adsorbed Cu(110) surface (Fig. 5).” and “This variation can be attributed to alterations in σ as the larger capacitances indicate more electrons on the electrode. The change in σ for intermediates adsorbed Cu(111) and Cu(100) surfaces during the urea synthesis are shown in Supplementary Fig. 35. The σ follows the order Cu(100) > Cu(111), which is consistent with the order of EDL capacitance (Fig. 5f) and reaction rate. Supplementary Fig. 35 also show the σ -dependent kinetic barriers of RDS for urea synthesis on both Cu(111) and Cu(100) surfaces, there is an increase in the kinetic barrier as σ decreases. Briefly, we find a novel strategy to design efficient electrocatalysts at the molecular level. By regulating the EDL with the larger capacitances, we can enhance the escape of electrons during the urea synthesis on Cu electrode surface, thereby decreasing the kinetic barrier of reaction steps and eventually result in an increased reaction rate.” have been added in Page 11 in the revised manuscript.

Figure R4 Change in surface-charge density (σ) for **a** N-intermediates or **b** C-intermediates adsorbed Cu(111), Cu(110) and Cu(100) surfaces as a function of applied electrode potential vs RHE. The surface charge is defined as the number of excess electrons per unit area (ACS Catal. 2022, 12, 6143).

Figure R5 Change in surface-charge density (σ) for intermediates adsorbed **a** Cu(111) and **b** Cu(100) surfaces as a function of applied electrode potential vs RHE during the urea synthesis process. **c** Comparison the surface-charge density of Cu(111) and Cu(100) surfaces during the urea synthesis. Kinetic barriers for the RDS of **d** Cu(111) and **e** Cu(100) during the urea synthesis process as a function of the surface-charge density. The surface charge is defined as the number of excess electrons per unit area.

● “Not-double-reference” DFT:

The authors corrected the “double-reference” terminology to “constant-potential,” which is indeed more sensible.

A small additional comment: The method in this paper is not technically constant-potential since the

electrode potential is only inferred/determined after running constant-charge calculations, as opposed to setting a target potential and letting the charge fluctuate. I do realize that the literature uses the term constant-potential (or grand canonical) loosely for both constant-charge (change q then determine V) and constant-potential (target V and let q fluctuate) DFT calculations. Therefore, I will not press further but invite the authors to assess their precise terminology in the future.

Response: Thanks for the insightful comments from the reviewer and pointing out the nuances in the terminology related to our DFT method. We acknowledge that we determine the electrode potential after running constant-charge calculations in our approach, which is a method to approximately simulate constant potentials in electrochemical reactions.

In light of the reviewer's suggestion, we recognize the importance of using precise terminology to avoid any potential confusion. We will be more careful in our future works to clarify this distinction and to ensure that our expression accurately reflects our methodology. We appreciate the reviewer's constructive suggestion and thanks for the opportunity to improve our work.

- VAPSol setting:

The authors laid out their arguments for the VAPSol settings (dielectric constant, Debye length, and surface tension terms) in good detail. I agree that these common settings are widely used in literature, but that does not exempt them from being physically questionable. The Noskov paper (ACS Catal. 2019, 9, 920) actually suggested that a dielectric constant of 80 is unphysical. They suggested using a lower dielectric constant (like 4) and modifying the default electron density cutoff to create a physical solvent and ion distribution. Although doing so indeed lowers the capacitance, it could entirely be the limitation of the implicit solvent to begin with.

Having said so, I'm aware that the impact of changing physical parameters like dielectric constant on the energetics of adsorbed species is not well documented. It is perhaps outside the scope of this manuscript to study the sensitivity of these VAPSol settings. I, therefore, accept using the commonly used settings and again invite the authors to study these parameters further in future work.

Response: We greatly appreciate reviewer's careful examination of our work. We agree with reviewer's point that the certain settings are widespread in the literature does not automatically validate their physical relevance.

As reviewer correctly pointed out, the impact of these parameters on the energetics of adsorbed species remains a relatively uncharted area, which could be an exciting avenue of future research. We genuinely value the reviewer's recommendation to delve deeper into these parameters in our subsequent works. This suggestion will guide our future studies on this topic, and we'll strive to shed more light on the sensitivity and implications of these VAPSol settings.

Reviewer #2:

Response: We would like to express our gratitude to Reviewer #2 for clarifying the co-reviewing process and for the contribution to the peer-review of our manuscript. We appreciate the time and effort both Reviewer #2 and the co-reviewer have invested in reviewing our work. We have further revised this manuscript as suggested by the reviewers.

Reviewer #3:

The authors have presented a substantially revised manuscript addressing the comments from reviewers. I appreciate the authors' efforts to improve their work; I however feel that some of the changes still leave a gap between the results and the conclusions. Most important among these is the treatment of charged species in calculating reaction barriers and adsorption energies. Some changes were additionally somewhat nonresponsive to aspects of my comments, which I further detail below. I believe the logical gaps in this version are too still substantial to justify the conclusions without significant qualification of the results.

- (1) I still disagree that the authors' approach to modeling activation energies of E-R steps will give physically meaningful results. The authors reference several papers purported to use a similar approach. However, two of the papers (J. Am. Chem. Soc. 2022, 144, 12800 and J. Am. Chem. Soc. 2021, 143, 9423) appear to be using grand canonical DFT/MD in which the number of electrons fluctuates and allows treatment of species such as H_3O^+ . This H_3O^+ is an important distinction from the H_3O (charge neutral) employed by the authors in this study.

In the present submitted work, the use of charge-neutral H_3O , which has already accepted an electron, creates an unstable initial state for the NEB calculation and thereby lowers the barrier calculated with NEB. This is indeed similar to the published work in Nat. Commun. 2023, 14, 3256, and perhaps also Nat. Commun. 2023, 14, 112 (the authors state they are working with H_3O^+ , though it appears to simply be H_3O). However, these treatments of H_3O^+ as H_3O still suffer from the same problems of underestimating barriers. The most significant issue is the use of these results to conclude that E-R is more favorable than L-H, since no artificially unstable initial state is being used for the L-H mechanism.

This issue is discussed in literature; one such example is work by Janik and collaborators (Catal Today 2017, 288, 63) that nicely summarizes various treatments in its introduction. More recent work includes J Phys Chem A 2022, 126, 7382 or work by Voth's group (J Chem Phys 2021, 154, 194506 or J Am Chem Soc 2021, 143, 18672).

At the very least, the authors need to acknowledge the limitations of their approach to using uncharged H_3O in their system. They should qualify that the barriers estimated with H_3O can be significantly underestimated due to the use of unstable H_3O rather than H_3O^+ , and that this may impact the qualitative mechanistic conclusions of this study.

Response: These points are really nice and important! Many thanks to the reviewer for this comment and providing highly useful references, which we have used to support our discussion about the newly added results for activation energies of Eley-Rideal (E-R) steps.

According to this review comment, we have added more testing calculation results for the activation energies of E-R steps. Before answering this comment, we need to apologize for the imprecision in our previous depiction of H_3O^+ in our previous version. In our revised analysis focusing on the NO_2^- RR of Cu(111), Cu(110), and Cu(100) surfaces, we have carefully examined the electron distribution of H_3O^+ during the proton-coupled electron transfer (PECT) steps by the Bader charge analysis. As demonstrated in Figure R1a,b,c, we observed that H_3O^+ used in our work loses more than 0.5 electrons under the applied electrode potential, thereby existing in an ionic state. This is primarily due to the calculation method in our work similar to the grand canonical density functional theory (DFT), where the number of electrons is allowed to fluctuate then the H_3O^+ could be optimized to its most stable state. We revised the H_3O as H_3O^+ in the revised version.

In addition, as highlighted in key publications such as Catal. Today, 2017, 288, 63; J. Phys. Chem. A, 2022, 126, 7382; J. Chem. Phys., 2021, 154, 194506; and J. Am. Chem. Soc., 2021, 143, 18672, representing ions in the bulk electrolyte using DFT is challenging. This is primarily due to limitations in unit cell size and the use of static solvation models. In light of this, we have re-evaluated our method based on the literature recommended by the reviewer. Figure R2 shows the optimized structures of the reactant ($^*\text{NO}_2 + 3\text{H}_2\text{O}$), reference ($^*\text{NO}_2 + 3\text{H}_2\text{O} + ^*\text{H}$), transition ($^*\text{NO}_2 + 2\text{H}_2\text{O} + \text{H}_3\text{O}^+$), and product ($^*\text{HNO}_2 + 2\text{H}_2\text{O}$) states used to calculate the potential-dependent kinetic barrier for hydrogenation step of $^*\text{NO}_2$ on Cu(111) surface. The kinetic barrier is 0.402 eV at $U_0 = 0.37$ V vs RHE with pH of 8.3 (the experimental conditions for urea synthesis).

The kinetic barrier can be further approximated as a linear function of the electrode potential using Butler-Volmer theory (Catal. Today, 2017, 288, 63):

$$\Delta G_{act} = \Delta G_{act(U_0)} + \beta F(U - U_0)$$

where F is the Faraday's constant, $F(U - U_0)$ is the kinetic barrier change due to the altered electrode potential and β is the reaction symmetry factor, approximated as 0.5 based on previous studies (Catal. Today, 2017, 288, 63). As depicted in Figure R3, the kinetic barrier calculated using the Butler-Volmer theory is lower than our initial results. This further substantiates our conclusion that Eley-Rideal (E-R) mechanisms are more favorable than Langmuir-Hinshelwood (L-H) mechanisms.

We recognize the inherent limitations of DFT in fully and accurately representing ions in bulk solutions. These challenges stem from the complexities of modeling dynamic electrochemical environments within the constraints of DFT. However, apart from these limitations, we believe that our DFT-based calculations still provide valuable insights and offer a reasonable approximation.

To address these points, the following changes have been made in the revised version:

- Figure R1 has been incorporated as **Supplementary Figure 36**.
- Figure R2 has been incorporated as **Supplementary Figure 37**.
- Figure R3 has been incorporated as **Supplementary Figure 38**.
- The discussions of "Taken NO_2 -RR of Cu(111), Cu(110), and Cu(100) surfaces as examples. Bader charge analysis is employed to investigate the electron distribution of H_3O^+ during the proton-coupled electron transfer (PECT) steps. As demonstrated in Supplementary Fig. 36, H_3O^+ loses more than 0.5 electrons under the applied electrode potential, thereby existing in an ionic state.

The ions in the bulk electrolyte using DFT is challenging due to limitations in unit cell size and the use of static solvation models. In light of this, we further compare our method with another approximating method for electrode potential-dependent activation energies proposed by Akhade, S. A., et al.¹⁻⁴ Supplementary Fig. 37 shows the optimized structures of the reactant ($^*\text{NO}_2 + 3\text{H}_2\text{O}$), reference ($^*\text{NO}_2 + 3\text{H}_2\text{O} + ^*\text{H}$), transition ($^*\text{NO}_2 + 2\text{H}_2\text{O} + \text{H}_3\text{O}^+$), and product ($^*\text{HNO}_2 + 3\text{H}_2\text{O}$) states used to calculate the potential-dependent kinetic barrier for hydrogenation step of $^*\text{NO}_2$ on Cu(111) surface. The kinetic barrier is 0.402 eV at $U_0 = 0.37$ V vs RHE with pH of 8.3 (the experimental conditions for urea synthesis).

The kinetic barrier can be approximated as a linear function of the electrode potential using Butler-Volmer theory:¹

$$\Delta G_{act} = \Delta G_{act(U_0)} + \beta F(U - U_0)$$

where F is the Faraday's constant, $F(U - U_0)$ is the kinetic barrier change due to the altered electrode potential and β is the reaction symmetry factor, approximated as 0.5 based on previous studies.¹ As depicted in Supplementary Fig. 38, the kinetic barrier calculated using the Butler-Volmer theory is lower than our initial results. This further substantiates our conclusion that Eley-Rideal (E-R) mechanisms are more favorable than Langmuir-Hinshelwood (L-H) mechanisms. Therefore, despite representing ions in the bulk electrolyte using DFT is challenging due to limitations in unit cell size and the use of static solvation models, our results still provide valuable insights and offer a reasonable approximation." have been added in Page 15 in the revised Supplementary Information.

Figure R1 Excess electrons of H_3O^+ in vacuum during the NO_2 -RR on **a** Cu(111), **b** Cu(110) and **c** Cu(100) surfaces.

Figure R2 Top and side view of the **a** reactant, **b** reference, **c** transition and **d** product state of the hydrogenation step of *NO_2 to *HNO_2 on the Cu (111) surface via E-R mechanism.

Figure R3 The comparison of the potential-dependent kinetic barrier (eV) of the hydrogenation step for *NO_2 on the Cu(111) surface via the method of this work and Butler Volmer theory with β equal to 0.5.

- (2) The authors present additional microkinetic modeling results in Figures R1-R3 to discuss coverage effects. The data are interesting, although the captions of these figures must be improved to include basic details such as which panels correspond to the items in the captions. The resolution after PDF conversion is also somewhat poor and difficult to read.

Response: Thanks for the reviewer's comments. We feel sorry for the lack of detail in the figure caption and the low resolution of the converted digital. We have revised the captions of Figures R4-R6 to clearly indicate which panels correspond to the items detailed in the captions. And we have enhanced the quality of these figures to ensure they are clear and easily readable upon conversion to PDF format.

Figure R4 TOF of urea synthesis as a function of **a d** NO_2^- concentration (0 to 1 M), **b e** CO_2 pressure (0 to 100 bar) and **c f** H^+ concentration (0 to 0.1 M) for Cu(111) and Cu(100) under the temperature of 300 K and the applied potential of -1.50 and -0.50 V on Cu(111) and Cu(100) surfaces, respectively.

Figure R5 Simulated coverage curves of adsorbed species for urea synthesis as a function of temperature on **a** Cu(111) and **b** Cu(100) under different pH values (pH = 7 to 14). The applied

potential of -1.50 and -0.50 V on Cu(111) and Cu(100) surfaces, respectively.

Figure R6 Simulated coverage curves of adsorbed species for urea synthesis as a function of temperature on **a** Cu (111) and **b** Cu(100) surfaces under different NO_2^- concentrations ($C_{NO_2^-} = 0.001$ to 1 M). The applied potential of -1.50 and -0.50 V on Cu(111) and Cu(100) surfaces, respectively. The pH is set as 8.3 for urea synthesis in accordance with the experimental environments.

My question was intended to discuss adsorbate interactions in the context of high coverages. The authors in Figure R2 show very high coverages by adsorbed species. Presumably there are no adsorbate interactions that are considered in these models, which leads to model-predicted coverages on the order of 1 ML by species. In practice, these coverages will be much lower due to repulsion of species, and this will significantly impact calculated reaction energies as well as the calculated rates. (To a point of reviewer 1, I suspect that the high coverages may play a role in the unphysical 50+ orders of magnitude difference in TOFs still being calculated in this study, which I do not believe can be explained on the energetics alone.)

Response: Many thanks for the close inspection of our results! We appreciate the chance to further clarify our findings for the huge difference in TOFs.

According to the reviewer's comment about the negative charge on NO_2^- , we further re-calculate the adsorption energy of $*NO_2$ at varying coverages as a function of U/RHE (from 0 to -1.5V) on the Cu(111) surface. As shown in the Figure R7b, the NO_2^- is indeed with an extra electron in our revised version. The adsorbate-adsorbate interactions, especially at high coverages, can indeed influence the overall adsorption energetics of $*NO_2$ (Figure R7a), and consequently may influence the reaction rates. In this work, we investigate the TOF for the microkinetic simulation under the low coverage of 1/8 ML $*NO_2$, the high coverage of $*NH$ and $*NO_2$ during the urea synthesis on Cu(111) and Cu(100) surfaces mainly caused by their RDS (Table R1,R4). The results are consistent with previous work (Nat. Commun. 2023, 14, 112 and Nat. Commun. 2023, 14, 3256). However, we find that the coverage indeed could be the reason for the huge difference in TOF as follows.

In the revised version, we have optimized our microkinetic modeling for NO_2 -RR, CO_2 RR, and urea synthesis processes, as detailed in Tables R1-11. These modifications substantially alter the TOF trends and values for urea synthesis and NO_2 -RR on the Cu(111) surface, as well as the degree of rate control (DRC) values in Tables R2 and R5. The revised modeling shows that both the NO_2 -RR and urea synthesis exhibit similar low TOF values on the Cu(111) surface, contrasting with higher TOF values on Cu(100), as depicted in Figure R8. The results can be well explained via their DRC values.

A key factor contributing to the distinct TOF trends on Cu(111) and Cu(100) surfaces is the difference in their DRC values, as detailed in Tables R2 and R5. This difference in DRC is a crucial determinant of TOF differences. On the Cu(111) surface, $*NO_2$ negatively impacts TOF, while C-intermediates have a positive effect. Conversely, $*NO_2$ exhibits a positive impact on TOF on the Cu(100) surface. Figure R9 illustrates that $*NO_2$ is stably adsorbed on both Cu(111) and Cu(100) surfaces. In contrast,

*CO₂ is physically adsorbed on these surfaces. This distinct adsorption behavior leads to a high TOF on Cu(100) due to the strong adsorption of *NO₂, and a lower TOF on Cu(111) owing to the combined effects of strong *NO₂ adsorption and weak *CO₂ adsorption. The coverage curves (Figure R10) indicate that *NH and *NO₂ are the predominant surface-covering species during the urea synthesis on Cu(111) and Cu(100), respectively. As demonstrated in Tables R2 and R5, *NH negatively impacts the reaction rate on Cu(111), while *NO₂ positively influences it on Cu(100). This further results in the distinct TOF trends on Cu(111) and Cu(100) surfaces.

To address these points, the following changes have been made in the revised version:

- Figure R7 has been revised and shown in the SI as **Supplementary Figure 14**.
- Figure R8 has been revised and shown in the manuscript as **Figure 4**.
- Figure R9 has been revised and shown in the SI as **Supplementary Figure 13**.
- Figure R10 has been revised and shown in the SI as **Supplementary Figure 29**.
- Table R1-11 has been incorporated as **Supplementary Table 13-23**.
- The discussions of **“To elucidate the huge differences in TOF on Cu(111) and Cu(100) surfaces during urea synthesis, a comprehensive degree of rate control (DRC) analysis for all intermediates and transition states is conducted, as detailed in Supplementary Tables 14,17. A positive (negative) Xi value signifies that an increase in reaction rate r need to further stabilization (destabilization) of the corresponding surface state. On the Cu(111) surface, the *NO₂ adversely affects TOF, while C-intermediates enhance it. Conversely, *NO₂ exhibits a positive impact on TOF on the Cu(100) surface. Supplementary Fig. 13 illustrates that *NO₂ can be stably adsorbed on both Cu(111) and Cu(100) surfaces while *CO₂ is physically adsorbed. This distinct adsorption behavior leads to a higher TOF on Cu(100) than Cu(111). Moreover, the coverage curves (Supplementary Fig. 29) indicate that *NH and *NO₂ are the predominant surface-covering species during the urea synthesis on Cu(111) and Cu(100), respectively. The *NH negatively impacts the reaction rate on Cu(111), while *NO₂ positively influences it on Cu(100) (Supplementary Tables 14,17). This further results in a lower TOF for Cu(111) and a higher TOF for Cu(100).”** have been added in **Page 9** in the revised manuscript.

Figure R7 a Adsorption energies of *NO₂ at varying coverages as a function of U/RHE (from 0.00 to -1.50 V) on the Cu(111) surface. **b** Excess electrons of NO₂⁻ in vacuum. The pH is set as 6.8 for NO₂⁻ RR in accordance with the experimental environments. The ML indicates the monolayer.

Figure R8 Microkinetic simulations. Turnover frequencies (TOF) per site for urea synthesis on **a** Cu(111) and **d** Cu(100) surfaces as functions of applied electrode potential vs RHE at 300 K and 1 bar. TOFs per site for NH₃ synthesis on **b** Cu(111) and **e** Cu(100) surfaces as a function of applied electrode potential vs RHE at 300 K and 1 bar. TOFs per site for urea synthesis on **c** Cu(111) and **f** Cu(100) surfaces mapped with pressure (1-100 bar) and temperature (300-700 K and 300-900K, respectively).

Figure R9 Adsorption energies of *NO₂, *CO₂, and *H as a function of U/RHE (from 0.50 to -1.50 V) on **a** Cu(111), **b** Cu(110) and **c** Cu(100) surfaces. **d** Excess electrons of NO₂⁻ in vacuum. The pH is set as 8.3 for urea synthesis in accordance with the experimental environments.

Figure R10 Simulated coverage curves of adsorbed species for urea synthesis as a function of temperature on Cu(111) (300 to 800K) and Cu(100) (300 to 1000K) under the applied electrode potential of -1.50 and -0.50 V, respectively. The pH is set as 8.3 for urea synthesis in accordance with the experimental environments.

Table R1 Microkinetic equations of urea synthesis process on the Cu(111) surface. * denotes active site. θ_i and P_i represent the coverage and pressure of reactants respectively. C_i is the concentrations of aqueous-phase species.

Reaction steps	Reaction rate (r)
$\text{NO}_2^- + * = *\text{NO}_2 + \text{e}^-$	$\theta(*\text{NO}_2) = K_1 C_{\text{NO}_2^-} \theta(*)$
$*\text{NO}_2 + \text{H}^+ + \text{e}^- = *\text{HNO}_2$	$r_2 = k_2 \theta(*\text{NO}_2) C_{\text{H}^+} - k_{-2} \theta(*\text{HNO}_2)$
$*\text{HNO}_2 + \text{H}^+ + \text{e}^- = *\text{NO} + \text{H}_2\text{O}$	$r_3 = k_3 \theta(*\text{HNO}_2) C_{\text{H}^+} - k_{-3} \theta(*\text{NO}) C_{\text{H}_2\text{O}}$
$*\text{NO} + \text{H}^+ + \text{e}^- = *\text{NOH}$	$r_4 = k_4 \theta(*\text{NO}) C_{\text{H}^+} - k_{-4} \theta(*\text{NOH})$
$*\text{NOH} + \text{H}^+ + \text{e}^- = *\text{N} + \text{H}_2\text{O}$	$r_5 = k_5 \theta(*\text{NOH}) C_{\text{H}^+} - k_{-5} \theta(*\text{N}) C_{\text{H}_2\text{O}}$
$*\text{N} + \text{H}^+ + \text{e}^- = *\text{NH}$	$r_6 = k_6 \theta(*\text{N}) C_{\text{H}^+} - k_{-6} \theta(*\text{NH})$
$\text{CO}_2 + * + \text{H}^+ + \text{e}^- = *\text{COOH}$	$r_7 = k_7 \theta(*) C_{\text{H}^+} P_{\text{CO}_2} - k_{-7} \theta(*\text{COOH})$
$*\text{COOH} + \text{H}^+ + \text{e}^- = *\text{CO} + \text{H}_2\text{O}$	$r_8 = k_8 \theta(*\text{COOH}) C_{\text{H}^+} - k_{-8} \theta(*\text{CO}) C_{\text{H}_2\text{O}}$
$*\text{NH} + *\text{CO} = *\text{CO-NH} + *$ (RDS)	$r_9 = k_9 \theta(*\text{NH}) \theta(*\text{CO}) - k_{-9} \theta(*\text{CO-NH}) \theta(*)$
$*\text{CO-NH} + *\text{NH} = *\text{NH-CO-NH} + *$	$r_{10} = k_{10} \theta(*\text{CO-NH}) \theta(*\text{NH}) - k_{-10} \theta(*\text{NH-CO-NH}) \theta(*)$
$*\text{NH-CO-NH} + *\text{H} = *\text{NH-CO-NH}_2 + *$	$r_{11} = k_{11} \theta(*\text{NH-CO-NH}) \theta(*\text{H}) - k_{-11} \theta(*\text{NH-CO-NH}_2) \theta(*)$
$*\text{NH-CO-NH}_2 + \text{H}^+ + \text{e}^- = \text{CO}(\text{NH}_2)_2 + *$	$r_{12} = k_{12} \theta(*\text{NH-CO-NH}_2) C_{\text{H}^+} - k_{-12} C_{\text{CO}(\text{NH}_2)_2} \theta(*)$
$\text{H}^+ + \text{e}^- + * = *\text{H}$	$\theta(*\text{H}) = K_{13} C_{\text{H}^+} \theta(*)$

Table R2 Degrees for rate control of all intermediates and transition states of urea synthesis process on the Cu(111) surface at 300 K under the applied electrode potential of -1.50 V.

Species (intermediates)	X_i	Species (transition-states)	X_i
*NO ₂	-1	*-NO ₂	0
*HNO ₂	-1	NO ₂ -H	0
*NO	-1	NOOH-H	0
*NOH	-1	NO-H	0
*N	-1	NOH-H	0
*NH	-1	N-H	0
*COOH	1	COO-H	0
*CO	1	COOH-H	0
*CO-NH	0	CO-NH	1
*NH-CO-NH	0	NHCO-NH	0
*NH-CO-NH ₂	9.19×10^{-61}	NHCONH-H	0
*	3.15×10^{-60}	*-CO(NH ₂) ₂	0
*H	-3.64×10^{-59}	*-H	0

Table R3 Microkinetic equations of urea synthesis process on the Cu(100) surface under the high applied electrode potential (0.00 to -0.42 V). * indicates active site. θ_i and P_i represent the coverage and pressure of reactants respectively. C_i is the concentrations of aqueous-phase species.

$\text{NO}_2^- + * = *\text{NO}_2 + e^-$	$\theta(*\text{NO}_2) = K_1 C_{\text{NO}_2^-} \theta(*)$
$*\text{NO}_2 + \text{H}^+ + e^- = *\text{HNO}_2$	$r_2 = k_2 \theta(*\text{NO}_2) C_{\text{H}^+} - k_{-2} \theta(*\text{HNO}_2)$
$*\text{HNO}_2 + \text{H}^+ + e^- = *\text{NO} + \text{H}_2\text{O}$	$r_3 = k_3 \theta(*\text{HNO}_2) C_{\text{H}^+} - k_{-3} \theta(*\text{NO}) C_{\text{H}_2\text{O}}$
$*\text{NO} + \text{H}^+ + e^- = *\text{NOH}$	$r_4 = k_4 \theta(*\text{NO}) C_{\text{H}^+} - k_{-4} \theta(*\text{NOH})$
$*\text{NOH} + \text{H}^+ + e^- = *\text{N} + \text{H}_2\text{O}$	$r_5 = k_5 \theta(*\text{NOH}) C_{\text{H}^+} - k_{-5} \theta(*\text{N}) C_{\text{H}_2\text{O}}$
$*\text{N} + *\text{H} = *\text{NH} + *$	$r_6 = k_6 \theta(*\text{N}) \theta(*\text{H}) - k_{-6} \theta(*\text{NH}) \theta(*)$
$\text{CO}_2 + *\text{H} = *\text{COOH}$	$r_7 = k_7 \theta(*\text{H}) P_{\text{CO}_2} - k_{-7} \theta(*\text{COOH})$
$*\text{COOH} + \text{H}^+ + e^- = *\text{CO} + \text{H}_2\text{O}$	$r_8 = k_8 \theta(*\text{COOH}) C_{\text{H}^+} - k_{-8} \theta(*\text{CO}) C_{\text{H}_2\text{O}}$
$*\text{N} + *\text{CO} = *\text{CO-N} + *$	$r_9 = k_9 \theta(*\text{N}) \theta(*\text{CO}) - k_{-9} \theta(*\text{CO-N}) \theta(*)$
$*\text{CO-N} + \text{H}^+ + e^- = *\text{CO-NH}$	$r_{10} = k_{10} \theta(*\text{CO-N}) C_{\text{H}^+} - k_{-10} \theta(*\text{CO-NH})$
$*\text{CO-NH} + *\text{NH} = *\text{NH-CO-NH} + *$ (RDS)	$r_{11} = k_{11} \theta(*\text{CO-NH}) \theta(*\text{NH}) - k_{-11} \theta(*\text{NH-CO-NH}) \theta(*)$
$*\text{NH-CO-NH} + \text{H}^+ + e^- = *\text{NH-CO-NH}_2$	$r_{12} = k_{12} \theta(*\text{NH-CO-NH}) C_{\text{H}^+} - k_{-12} \theta(*\text{NH-CO-NH}_2)$
$*\text{NH-CO-NH}_2 + \text{H}^+ + e^- = \text{CO}(\text{NH}_2)_2 + *$	$r_{13} = k_{13} \theta(*\text{NH-CO-NH}_2) C_{\text{H}^+} - k_{-13} C_{\text{CO}(\text{NH}_2)_2} \theta(*)$
$\text{H}^+ + e^- + * = *\text{H}$	$\theta(*\text{H}) = K_{14} C_{\text{H}^+} \theta(*)$

Table R4 Microkinetic equations of urea synthesis process on the Cu(100) surface under the low applied electrode potential (-0.42 to -1.30 V). * indicates active site. θ_i and P_i represent the coverage and pressure of reactants respectively. C_i is the concentrations of aqueous-phase species.

$\text{NO}_2^- + * = *\text{NO}_2 + e^-$	$\theta(*\text{NO}_2) = K_1 C_{\text{NO}_2^-} \theta(*)$
$*\text{NO}_2 + \text{H}^+ + e^- = *\text{HNO}_2$ (RDS)	$r_2 = k_2 \theta(*\text{NO}_2) C_{\text{H}^+} - k_{-2} \theta(*\text{HNO}_2)$
$*\text{HNO}_2 + \text{H}^+ + e^- = *\text{NO} + \text{H}_2\text{O}$	$r_3 = k_3 \theta(*\text{HNO}_2) C_{\text{H}^+} - k_{-3} \theta(*\text{NO}) C_{\text{H}_2\text{O}}$
$*\text{NO} + \text{H}^+ + e^- = *\text{NOH}$	$r_4 = k_4 \theta(*\text{NO}) C_{\text{H}^+} - k_{-4} \theta(*\text{NOH})$
$*\text{NOH} + \text{H}^+ + e^- = *\text{N} + \text{H}_2\text{O}$	$r_5 = k_5 \theta(*\text{NOH}) C_{\text{H}^+} - k_{-5} \theta(*\text{N}) C_{\text{H}_2\text{O}}$
$*\text{N} + *\text{H} = *\text{NH} + *$	$r_6 = k_6 \theta(*\text{N}) \theta(*\text{H}) - k_{-6} \theta(*\text{NH}) \theta(*)$
$*\text{N} + \text{CO}_2 = *\text{CO}_2\text{-N}$	$r_7 = k_7 \theta(*\text{N}) P_{\text{CO}_2} - k_{-7} \theta(*\text{CO}_2\text{-N})$
$*\text{CO}_2\text{-N} + \text{H}^+ + e^- = *\text{COOH-N}$	$r_8 = k_8 \theta(*\text{CO}_2\text{-N}) C_{\text{H}^+} - k_{-8} \theta(*\text{COOH-N})$
$*\text{COOH-N} + \text{H}^+ + e^- = *\text{CO-N} + \text{H}_2\text{O}$	$r_9 = k_9 \theta(*\text{COOH-N}) C_{\text{H}^+} - k_{-9} \theta(*\text{CO-N}) C_{\text{H}_2\text{O}}$
$*\text{CO-N} + \text{H}^+ + e^- = *\text{CO-NH}$	$r_{10} = k_{10} \theta(*\text{CO-N}) C_{\text{H}^+} - k_{-10} \theta(*\text{CO-NH})$
$*\text{CO-NH} + *\text{NH} = *\text{NH-CO-NH} + *$	$r_{11} = k_{11} \theta(*\text{CO-NH}) \theta(*\text{NH}) - k_{-11} \theta(*\text{NH-CO-NH}) \theta(*)$
$*\text{NH-CO-NH} + \text{H}^+ + e^- = *\text{NH-CO-NH}_2$	$r_{12} = k_{12} \theta(*\text{NH-CO-NH}) C_{\text{H}^+} - k_{-12} \theta(*\text{NH-CO-NH}_2)$
$*\text{NH-CO-NH}_2 + \text{H}^+ + e^- = \text{CO}(\text{NH}_2)_2 + *$	$r_{13} = k_{13} \theta(*\text{NH-CO-NH}_2) C_{\text{H}^+} - k_{-13} C_{\text{CO}(\text{NH}_2)_2} \theta(*)$
$\text{H}^+ + e^- + * = *\text{H}$	$\theta(*\text{H}) = K_{14} C_{\text{H}^+} \theta(*)$

Table R5 Degrees for rate control of all intermediates and transition states of urea synthesis process on the Cu(100) surface at 300 K under the applied electrode potential of -0.50 V.

Species (intermediates)	X_i	Species (transition-states)	X_i
*NO ₂	2.17×10 ⁻²³	*-NO ₂	0
*HNO ₂	0	NO ₂ -H	1
*NO	0	NOOH-H	0
*NOH	0	NO-H	0
*N	0	NOH-H	0
*NH	3.12×10 ⁻³²	N-H	0
*CO ₂ -N	-3.12×10 ⁻³²	CO ₂ -N	0
*COOH-N	-3.12×10 ⁻³²	NCOO-H	0
*CO-N	-3.12×10 ⁻³²	CO-N	0
*CO-NH	-3.12×10 ⁻³²	CON-H	0
*NH-CO-NH	3.12×10 ⁻³²	NHCO-NH	0
*NH-CO-NH ₂	2.33×10 ⁻²⁷	NHCONH-H	0

*	2.33×10^{-27}	*-CO(NH ₂) ₂	0
*H	-1.02×10^{-23}	*-H	0

Table R6 Microkinetic equations of NO₂⁻RR process on Cu(111) surface. * indicates active site. θ_i and P_i represent the coverage and pressure of reactants respectively. C_i is the concentrations of aqueous-phase species.

NO ₂ ⁻ + * = *NO ₂ + e ⁻	θ(*NO ₂) = K ₁ C _{NO₂⁻} θ(*)
*NO ₂ + H ⁺ + e ⁻ = *HNO ₂	r ₂ = k ₂ θ(*NO ₂)C _{H⁺} - k ₋₂ θ(*HNO ₂)
*HNO ₂ + H ⁺ + e ⁻ = *NO + H ₂ O	r ₃ = k ₃ θ(*HNO ₂)C _{H⁺} - k ₋₃ θ(*NO)C _{H₂O}
*NO + H ⁺ + e ⁻ = *NOH	r ₄ = k ₄ θ(*NO)C _{H⁺} - k ₋₄ θ(*NOH)
*NOH + H ⁺ + e ⁻ = *N + H ₂ O	r ₅ = k ₅ θ(*NOH)C _{H⁺} - k ₋₅ θ(*N)C _{H₂O}
*N + H ⁺ + e ⁻ = *NH	r ₆ = k ₆ θ(*N)C _{H⁺} - k ₋₆ θ(*NH)
*NH + *H = *NH ₂ + * (RDS)	r ₇ = k ₇ θ(*NH)θ(*H) - k ₋₇ θ(*NH ₂)θ(*)
*NH ₂ + H ⁺ + e ⁻ = * + NH ₃	r ₈ = k ₈ θ(*NH ₂)C _{H⁺} - k ₋₈ P _{NH₃} θ(*)
H ⁺ + e ⁻ + * = *H	θ(*H) = K ₉ C _{H⁺} θ(*)

Table R7 Degrees for rate control of all intermediates and transition states of NO₂⁻RR process on the Cu(111) surface at 300 K under the applied electrode potential of -1.50 V.

Species (intermediates)	X _i	Species (transition-states)	X _i
*NO ₂	-1	*-NO ₂	0
*HNO ₂	-1	NO ₂ -H	0
*NO	-1	NOOH-H	0
*NOH	-1	NO-H	0
*N	-1	NOH-H	0
*NH	-1	N-H	0
*NH ₂	2.96×10^{-71}	NH-H	1
*	2.96×10^{-71}	*-NH ₃	0
*H	2.96×10^{-71}	*-H	0

Table R8 Microkinetic equations of NO₂⁻RR process on Cu(100) surface. * indicates active site. θ_i and P_i represent the coverage and pressure of reactants respectively. C_i is the concentrations of aqueous-phase species.

NO ₂ ⁻ + * = *NO ₂ + e ⁻	θ(*NO ₂) = K ₁ C _{NO₂⁻} θ(*)
*NO ₂ + H ⁺ + e ⁻ = *HNO ₂	r ₂ = k ₂ θ(*NO ₂)C _{H⁺} - k ₋₂ θ(*HNO ₂)
*HNO ₂ + H ⁺ + e ⁻ = *NO + H ₂ O	r ₃ = k ₃ θ(*HNO ₂)C _{H⁺} - k ₋₃ θ(*NO)C _{H₂O}
*NO + H ⁺ + e ⁻ = *NOH	r ₄ = k ₄ θ(*NO)C _{H⁺} - k ₋₄ θ(*NOH)
*NOH + H ⁺ + e ⁻ = *N + H ₂ O	r ₅ = k ₅ θ(*NOH)C _{H⁺} - k ₋₅ θ(*N)C _{H₂O}
*N + *H = *NH + *	r ₆ = k ₆ θ(*N)θ(*H) - k ₋₆ θ(*NH)θ(*)
*NH + H ⁺ + e ⁻ = *NH ₂ (RDS)	r ₇ = k ₇ θ(*NH)C _{H⁺} - k ₋₇ θ(*NH ₂)
*NH ₂ + H ⁺ + e ⁻ = * + NH ₃	r ₈ = k ₈ θ(*NH ₂)C _{H⁺} - k ₋₈ P _{NH₃} θ(*)
H ⁺ + e ⁻ + * = *H	θ(*H) = K ₉ C _{H⁺} θ(*)

Table R9 Degrees for rate control of all intermediates and transition states of NO₂⁻RR process on the Cu(100) surface at 300 K under the applied electrode potential of -0.75 V.

Species (intermediates)	X _i	Species (transition-states)	X _i
*NO ₂	0	*-NO ₂	0
*HNO ₂	5.97×10^{-72}	NO ₂ -H	0
*NO	5.97×10^{-72}	NOOH-H	0
*NOH	1.59×10^{-50}	NO-H	0
*N	1.25×10^{-42}	NOH-H	0

*NH	6.24×10^{-18}	N-H	0
*NH ₂	0	NH-H	1
*	0	*-NH ₃	0
*H	6.24×10^{-18}	*-H	0

Table R10 Microkinetic equations of CO₂RR process on Cu(111) surface. * indicates active site. θ_i and P_i represent the coverage and pressure of reactants respectively. C_i is the concentrations of aqueous-phase species.

$\text{CO}_2 + * = *\text{CO}_2$	$\theta(*\text{CO}_2) = K_1 P_{\text{CO}_2} \theta(*)$
$*\text{CO}_2 + \text{H}^+ + \text{e}^- = *\text{COOH}$	$r_2 = k_2 \theta(*\text{CO}_2) C_{\text{H}^+} - k_{-2} \theta(*\text{COOH}) \theta(*)$
$*\text{COOH} + \text{H}^+ + \text{e}^- = *\text{CO} + \text{H}_2\text{O}$	$r_3 = k_3 \theta(*\text{COOH}) C_{\text{H}^+} - k_{-3} \theta(*\text{CO}) C_{\text{H}_2\text{O}}$
$*\text{CO} = * + \text{CO}$ (RDS)	$r_4 = k_4 \theta(*\text{CO}) - k_{-4} \theta(*) P_{\text{CO}}$

Table R11 Microkinetic equations of CO₂RR process on Cu(100) surface. * indicates active site. θ_i and P_i represent the coverage and pressure of reactants respectively. C_i is the concentrations of aqueous-phase species.

$\text{CO}_2 + * = *\text{CO}_2$	$\theta(*\text{CO}_2) = K_1 P_{\text{CO}_2} \theta(*)$
$*\text{CO}_2 + *\text{H} = *\text{COOH} + *$	$r_2 = k_2 \theta(*\text{CO}_2) \theta(*\text{H}) - k_{-2} \theta(*\text{COOH}) \theta(*)$
$*\text{COOH} + \text{H}^+ + \text{e}^- = *\text{CO} + \text{H}_2\text{O}$	$r_3 = k_3 \theta(*\text{COOH}) C_{\text{H}^+} - k_{-3} \theta(*\text{CO}) C_{\text{H}_2\text{O}}$
$*\text{CO} = * + \text{CO}$ (RDS)	$r_4 = k_4 \theta(*\text{CO}) - k_{-4} \theta(*) P_{\text{CO}}$
$\text{H}^+ + \text{e}^- + * = *\text{H}$	$\theta(*\text{H}) = K_5 C_{\text{H}^+} \theta(*)$

My second question related to the adsorption of charged species also seems to not have been addressed. The authors present figures R4 and R5, which are interesting data. It is however very surprising to me that NO₂(-) would bind more strongly at more negative potentials, given the negative charge on NO₂(-). It seems based on Table R1 that the authors are not accounting for the negative charge on NO₂(-) when calculating these adsorption energies, which represents a significant issue as the potential-dependent behavior depends on the treatment of this charge. The quantitative and qualitative behaviors of NO₂(-) adsorption will differ significantly from that for NO₂(neutral) as shown.

Response: Many thanks to the reviewer for this comment that helped us get more accurate results about the adsorption energies of charged species like NO₂⁻. This is a really nice and important point indeed deserving a detailed discussion and analysis, which we believe will also be highly useful for numerous related studies in the future.

We completely re-examined our approach and the associated data, particularly in relation to Figures R7,R9 and Table R12. We realized that our initial analysis did not sufficiently consider the negative charge of NO₂⁻ when calculating the adsorption energies. Different from the H₃O⁺, NO₂⁻ can not be automatically optimized to the negative charged state due to NO₂ has the other stable neutral state. This oversight result in inaccuracies in both the quantitative and qualitative interpretations of our previous findings.

To address this issue, we have revised our calculations to more accurately account for the negative charge on NO₂⁻. We have updated our computational models to include the negative charge effect on NO₂⁻. This was achieved by adding an electron to the systems, enabling us to more precisely simulate the interaction between NO₂⁻ and the charged surface under various electrode potentials. As shown in Figures R7,R9, NO₂⁻ could maintain the negative charged state under the applied potentials. Contrary to our initial findings, the results suggest that NO₂⁻ binds more strongly at more positive potentials.

To address these points, the following changes have been made in the revised version:

- Table R12 has been incorporated as **Supplementary Table 28**.
- The discussions of "Different from the H₃O⁺, NO₂⁻ can not be automatically optimized to the negative charged state due to NO₂ has the other stable neutral state. We add an electron to the systems, enabling to more precisely simulate the interaction between NO₂⁻ and the charged surface under various electrode potentials. As shown in Supplementary Fig. 13,14, the NO₂⁻

could maintain the negative charged state under the applied potentials. The results suggest that NO₂⁻ binds more strongly at more positive potentials.” have been added in Page 6 in the revised Supplementary Information.

Table R12 Calculated Gibbs free energy values (ΔG) for reaction steps during the urea synthesis on Cu(111) and Cu(100) surfaces with respect to the applied electrode potentials (vs RHE), corresponding to Supplementary Tables 13,15,16. The unit is eV.

Cu(111)	-1.50 V	-1.25 V	-1.00 V	-0.75 V	-0.50 V	-0.25 V	0.00 V
$\Delta G(*NO_2)$	-0.559	-0.611	-0.684	-0.841	-1.087	-1.335	-1.515
$\Delta G(*NO_2H)$	0.351	0.311	0.271	0.228	0.182	0.132	0.079
$\Delta G(*NO)$	-2.532	-2.546	-2.560	-2.529	-2.436	-2.331	-2.215
$\Delta G(*NOH)$	-0.304	-0.298	-0.292	-0.283	-0.230	-0.171	-0.116
$\Delta G(*N)$	-1.463	-1.501	-1.538	-1.551	-1.560	-1.569	-1.578
$\Delta G(*NH)$	-1.315	-1.332	-1.349	-1.350	-1.337	-1.321	-1.300
$\Delta G(*COOH)$	0.064	0.004	-0.055	-0.070	-0.069	-0.058	-0.037
$\Delta G(*CO)$	-0.896	-0.902	-0.908	-0.919	-0.937	-0.961	-0.990
$\Delta G(*CO-NH)$	0.041	0.024	0.008	-0.009	-0.012	-0.004	0.005
$\Delta G(*NH-CO-NH)$	-1.212	-1.228	-1.243	-1.258	-1.264	-1.262	-1.258
$\Delta G(*NH-CO-NH_2)$	-0.227	-0.245	-0.264	-0.288	-0.381	-0.493	-0.601
$\Delta G(*NH_2-CO-NH_2)$	-0.533	-0.535	-0.541	-0.551	-0.556	-0.554	-0.547
$\Delta G(*H)$	-0.290	-0.320	-0.350	-0.370	-0.302	-0.196	-0.09

Cu(100)	-1.50 V	-1.25 V	-1.00 V	-0.75 V	-0.50 V	-0.25 V	0.00 V
$\Delta G(*NO_2)$	-0.994	-1.006	-1.034	-1.167	-1.425	-1.702	-1.915
$\Delta G(*NO_2H)$	0.539	0.476	0.414	0.312	0.198	0.103	0.026
$\Delta G(*NO)$	-2.252	-2.333	-2.414	-2.435	-2.393	-2.321	-2.220
$\Delta G(*NOH)$	-0.911	-0.897	-0.884	-0.867	-0.829	-0.801	-0.783
$\Delta G(*N)$	-1.828	-1.846	-1.865	-1.866	-1.802	-1.736	-1.672
$\Delta G(*NH)$	-0.919	-0.916	-0.913	-0.910	-0.918	-0.935	-0.957
$\Delta G(*CO_2-N)$	0.092	0.036	-0.021	-0.078	-0.104		
$\Delta G(*COOH-N)$	-1.339	-1.321	-1.302	-1.283	-1.262		
$\Delta G(*COOH)$						0.179	0.138
$\Delta G(*CO)$						-0.986	-0.993
$\Delta G(*CO-N)$	-1.237	-1.215	-1.192	-1.169	-1.113	-1.058	-1.026
$\Delta G(*CO-NH)$	-0.720	-0.773	-0.826	-0.888	-0.932	-0.951	-0.945
$\Delta G(*NH-CO-NH)$	-1.470	-1.475	-1.480	-1.485	-1.471	-1.447	-1.424
$\Delta G(*NH-CO-NH_2)$	-0.154	-0.179	-0.205	-0.242	-0.274	-0.299	-0.319
$\Delta G(*NH_2-CO-NH_2)$	-0.578	-0.606	-0.635	-0.668	-0.696	-0.720	-0.740
$\Delta G(*H)$	-0.501	-0.516	-0.531	-0.534	-0.491	-0.455	-0.426

Further, my last point about transport of species to the surface is not necessarily related to the temperature effects and adsorption, but rather to the treatment of the complex gradients involved with transporting charged species from the bulk electrolyte to the local environment of the surface. Such modeling is significantly more involved than for comparable gas-phase contexts, and the quantitative nature of results will be affected by choices of appropriate chemical potentials and, e.g., collision theory in the condensed phase context.

Again, at the very least these items need to be discussed as limitations of the present study.

Response: Thanks for the valuable concern. We understand the concern regarding the transport processes and agree that the modeling of charged species transport in the electrolyte involves intricate gradient considerations that are distinct from gas-phase processes. Such phenomena are indeed more complex due to the interactions between the charged species and the local environment at the electrode-electrolyte interface. A fully explicit description of the electrolyte is needed to provide

the most accurate description of the transporting charged species from the bulk electrolyte to the local environment of the surface. Such a method would involve complex ab initio molecular dynamics (AIMD) simulations incorporating both water molecules and ions explicitly. Although complex gradients study can make the calculation more accurate, it is usually more expensive and not affordable for a large system and complex reaction mechanisms like urea synthesis as we are studying in this work.

We acknowledged this limitation of our study in the revised version and added discussions of “Modeling the charged species transport in the electrolyte is important while complex due to the interactions between the charged species and the local environment at the electrode-electrolyte interface. A fully explicit description of the electrolyte is needed to provide the most accurate description of the transporting properties. Such a method would involve complex AIMD simulations incorporating both water molecules and ions explicitly.⁶⁶⁻⁶⁹ However, complex gradients study is more expensive and not affordable for a large system, particularly when exploring complex reaction mechanisms like urea synthesis in our study, it exceeds the capabilities of our current computational resources.” in Page 13 in revised manuscript.

- (3) Minor point, but it would be most typical to present the free energies of Supplementary Table 20-20 relative to a uniform reference (e.g., clean surface and reactants), rather than using the arbitrary energy reference provided by the DFT code. Such an approach would enable easier comparison of energies across the various intermediates and surfaces.

Response: Thanks for the reviewer’s comments. The calculated energetics of adsorbed species have been revised as Supplementary Tables 20-23 in the revised version.

Table R13 Calculated Gibbs free energy values (ΔG) for reaction steps during the NO_2 -RR on Cu(111), Cu(110) and Cu(100) surfaces without extra charges added, corresponding to Supplementary Table 18. The unit is eV.

Cu(111)	*NO ₂	*HNO ₂	*NO	*NOH	*N	*NH	*NH ₂	*NH ₃
ΔG (eV) (L-H)	-0.929	0.315	-1.552	0.145	-0.968	-1.131	-0.461	-0.543
ΔG (eV) (E-R)	-1.567	0.554	-1.864	0.112	-1.010	-0.951	-0.385	-0.951
Cu(110)	*NO ₂	*HNO ₂	*NO	*NOH	*N	*NH	*NH ₂	*NH ₃
ΔG (eV) (L-H)	-1.992	0.688	-1.401	0.350	-1.556	-0.640	-0.987	-0.381
ΔG (eV) (E-R)	-1.086	0.806	-1.138	0.224	-1.495	-0.716	-1.046	-1.039
Cu(100)	*NO ₂	*HNO ₂	*NO	*NOH	*N	*NH	*NH ₂	*NH ₃
ΔG (eV) (L-H)	-1.197	0.548	-1.265	-0.344	-1.423	-0.857	-0.246	-0.484
ΔG (eV) (E-R)	-1.879	0.705	-1.357	-0.440	-1.211	-0.767	-0.315	-0.737

Table R14 Calculated Gibbs free energy values (ΔG) for reaction steps during the CO_2 RR on Cu(111), Cu(110) and Cu(100) surfaces without extra charges added, corresponding to Supplementary Table 19. The unit is eV.

Cu(111)	*CO ₂	*COOH	*CO	*HCO	CO (g)
ΔG (eV) (L-H)	-0.037	0.383	-0.372	0.882	1.022
ΔG (eV) (E-R)	0.414	0.382	-0.270	0.660	
Cu(110)	*CO ₂	*COOH	*CO	*HCO	CO (g)
ΔG (eV) (L-H)	0.089	0.118	-0.447	0.931	1.342
ΔG (eV) (E-R)	0.653	0.024	-0.179	-0.474	
Cu(100)	*CO ₂	*COOH	*CO	*HCO	CO (g)
ΔG (eV) (L-H)	0.028	0.279	-0.376	0.668	1.069
ΔG (eV) (E-R)	0.566	0.115	-0.256	-0.108	

Table R15 Calculated Gibbs free energy values (ΔG) for the possible coupling steps (the first coupling step) during the urea synthesis on Cu(111), Cu(110) and Cu(100) surfaces without extra charges added.

The unit is eV.

Cu(111)			*NO	*NOH	*N	*NH	
ΔG (eV) (*CO ₂)			0.447	-0.341	-0.450	-0.362	
ΔG (eV) (*CO)			0.744	0.040	-1.591	0.019	
Cu(110)	*NO ₂	*HNO ₂	*NO	*NOH	*N	*NH	*NH ₂
ΔG (eV) (*CO ₂)	-0.391	-0.601	-0.599	-1.461	-1.134	-1.106	-0.570
ΔG (eV) (*CO)			0.427	-0.836	-2.060	-0.678	0.134
Cu(100)	*NO ₂	*HNO ₂	*NO	*NOH	*N	*NH	
ΔG (eV) (*CO ₂)	-0.211	-0.217	-0.003	-0.681	0.035	-0.006	
ΔG (eV) (*COOH)	-0.471	-0.131	-1.123	-0.962	-1.057	-1.157	
ΔG (eV) (*CO)			0.138	-0.402	-1.021	-0.213	

Table R16 Calculated Gibbs free energy values (ΔG) for the possible coupling steps (the second coupling step) during the urea synthesis on Cu(111) and Cu(100) surfaces without extra charges added. The unit is eV.

Cu(111)	*NH-CO-NOH	*NH-CO-NH
ΔG	-0.512	-1.250

Cu(100)	*N-COOH-N	*N-CO-NH	*NH-CO-NH
ΔG	-1.248	-0.948	-1.413

REVIEWER COMMENTS

Reviewer #1 (Remarks to the Author):

Generally the authors have addressed my comments, but I have concerns regarding the premise that this is an "EDL-tuned urea mechanism" based on the microkinetic modeling analysis.

Re: Electric double-layer capacitance analysis:

In this revision, the authors have clarified the correlation of CEDL vs. urea TOF through an analysis of surface charge densities. I'm satisfied with this physical explanation for explaining the difference in RDS activation barriers. Some of the new data is somewhat hard to interpret (Fig. R4 and R5 a, b, c) but the overall picture and the argument is clear. The RDS activation barriers vary with surface charges, as shown in Fig. R5 d and e, by roughly 0.05 to 0.1 eV across the potential range.

However, the follow-up conclusion/assertion that this is an "EDL-tuned urea mechanism" is not supported based on my understanding. The authors' sequential arguments are:

1. Cu(100) have higher EDL capacitance than Cu(111).
2. Cu(100) have higher surface charges than Cu(111).
3. Cu(100) have different potential-dependent RDS activation barriers from Cu(111) (Fig. R5).
4. Cu(100) have higher urea TOFs than Cu(111) (microkinetic model).
5. Therefore, higher EDL capacitance is a descriptor for higher urea TOF

 an EDL-tune urea mechanism.

Nevertheless, as already shown from the microkinetic model, the differences in RDS activation barriers alone can not explain the urea TOF between Cu(111) and Cu(100). This means point 3 does not connect with point 4, so point 1 can not connect with point 5.

The presented surface-charge vs. activation barrier argument (although indeed physically sound) can not support the broad assertion that this is an "EDL-tuned urea mechanism."

The findings in this work still perhaps deserve publication after revision regarding the story. I'd suggest the authors to reconsider their main premise (i.e., a new title and a new punch-line). For example, points 1-3 and point 4 can be made independently and separately, and they would still be interesting and important findings

re: Microkinetic model

In this revision, the authors have pointed out a very plausible origin for the large difference in urea TOFs between Cu(100) and Cu(111). The difference in the RDS coupled with the difference in surface coverages conceivably yields many orders of magnitude difference in the TOF. These are important findings.

However, these findings and interpretations could be elaborated more clearly in my opinion. In the rebuttal and manuscript revision, the facts presented are all reasonable but a bit poorly organized, failing to highlight important points. I'd suggest adding straight-to-the-point arguments, something like:

"On Cu(111), *CO---NH is the RDS; the surface is covered fully with *NH but with very little *CO.

On Cu(100), *NO2---H is the RDS; the surface is covered fully with *NO2.

Cu(111) struggles to get *CO while Cu(100) gets H+ from the electrolyte (assuming ER step).

(To support, the authors can then explain the computed Xi values of *CO and *N-species.)"

And finally, some broad concluding sentences like:

"The reason for large difference in urea TOFs between 100 and 111 is not activation barriers alone.

It is the different RDS and the availability of reactants (*NH/*CO for 111; *NO2/H+ for 100)".

But overall, I find the authors have adequately addressed my concerns in this microkinetic section.

Reviewer #2 (Remarks to the Author):

Reviewer #3 (Remarks to the Author):

The authors have satisfactorily responded to my comments; I recommend this work for publication.

Response to the comments of Reviewers

We are deeply grateful to the reviewers for their critical and constructive comments, which contribute to improving the quality of this manuscript. We have carefully addressed all the comments and revised the manuscript accordingly. The revisions are highlighted in the manuscript. The point-by-point responses are listed below, all the changes are highlighted in red color in the revised manuscript.

Reviewer #1 (Remarks to the Author):

Generally the authors have addressed my comments, but I have concerns regarding the premise that this is an "EDL-tuned urea mechanism" based on the microkinetic modeling analysis.

Re: Electric double-layer capacitance analysis:

In this revision, the authors have clarified the correlation of CEDL vs. urea TOF through an analysis of surface charge densities. I'm satisfied with this physical explanation for explaining the difference in RDS activation barriers. Some of the new data is somewhat hard to interpret (Fig. R4 and R5 a, b, c) but the overall picture and the argument is clear. The RDS activation barriers vary with surface charges, as shown in Fig. R5 d and e, by roughly 0.05 to 0.1 eV across the potential range.

However, the follow-up conclusion/assertion that this is an "EDL-tuned urea mechanism" is not supported based on my understanding. The authors' sequential arguments are:

1. Cu(100) have higher EDL capacitance than Cu(111).
2. Cu(100) have higher surface charges than Cu(111).
3. Cu(100) have different potential-dependent RDS activation barriers from Cu(111) (Fig. R5).
4. Cu(100) have higher urea TOFs than Cu(111) (microkinetic model).
5. Therefore, higher EDL capacitance is a descriptor for higher urea TOF

 an EDL-tune urea mechanism.

Nevertheless, as already shown from the microkinetic model, the differences in RDS activation barriers alone can not explain the urea TOF between Cu(111) and Cu(100). This means point 3 does not connect with point 4, so point 1 can not connect with point 5.

The presented surface-charge vs. activation barrier argument (although indeed physically sound) can not support the broad assertion that this is an "EDL-tuned urea mechanism."

The findings in this work still perhaps deserve publication after revision regarding the story. I'd suggest the authors to reconsider their main premise (i.e., a new title and a new punch-line). For example, points 1-3 and point 4 can be made independently and separately, and they would still be interesting and important findings.

Response: Thanks for the thorough review and constructive comment regarding on electric double-layer capacitance analysis. We are grateful for the acknowledgment of our efforts to elucidate the correlation between CEDL and RDS kinetic barrier and we appreciate the insights for enhancing our presentation and conclusions.

- ◆ According to the reviewer's suggestion, we have revised the main premise of our work, including the title and the main conclusions of our manuscript.
 - We updated the title to "Potential and electric double-layer effect in electrocatalytic urea synthesis" which will be more accurately reflects the scope and findings of our research.
 - We removed the expression that the EDL capacitance could be an indicator for urea TOF, and revised the presentation of the abstract and conclusion as "The electric double-layer

capacitance also plays a key role in urea synthesis. Based on these findings, we propose two essential strategies to promote the efficiency of urea synthesis on Cu electrodes: increasing Cu(100) surface ratio and elevating the reaction temperature.” and “Moreover, the capacitance of EDL is proved to be critical for urea synthesis on Cu surfaces. It is worth noting that while the EDL capacitance is effective in regulating the kinetic barrier of RDS, its impact on the reaction rate of complex reactions should be evaluated in conjunction with other factors such as coverage. Based on these findings, we propose the design principles for promoting the efficiency of urea synthesis, i.e., increasing (100) surface ratio and elevating the reaction temperature.”.

- ◆ According to the reviewer’s concern, we refrained from implying a direct causal relationship between EDL capacitance and urea TOF. We removed the expression of “EDL-tuned urea mechanism” and any suggestion that EDL alone determines the reaction rate. The corresponding discussions have been revised as “Therefore, tuning the capacitance of the EDL could be a way to optimize the electrochemical synthesis of urea. However, it needs to be evaluated with other factors to comprehensively assess the impact on the reaction rate, due to the TOF is not influenced by the kinetic barriers alone as discussed above.” in Page 12 in the revised version. This adjustment ensures that points 1-3 (regarding EDL capacitance, surface charges, and kinetic barriers of RDS) and point 4 (concerning the reaction rate) are presented independently, highlighting their individual significance and relevance to our research.
- ◆ As for the Reviewer’s concern about Supplementary Fig. 34 and Supplementary Fig. 35a,b,c, we further illustrated them to make the discussion more clarity.
 - In Supplementary Fig. 34, we calculated the surface-charge density (σ) for N-intermediates and C-intermediates adsorbed Cu(111), Cu(110) and Cu(100) surfaces. The C- and N-intermediates adsorbed Cu(111) and Cu(100) surfaces are characterized by positive σ , showing identical interfacial fields. Therefore, the preferred electrochemical environment for adsorbed C- and N-intermediates on Cu(111) and Cu(100) surfaces are similar. This uniformity indicates that C- and N-intermediates can coexist on these surfaces, thereby providing steric possibilities for the coupling steps. In contrast, C- and N-intermediates adsorbed Cu(110) surface show opposite interfacial fields. The C-intermediates prefer formed on the electrode surface with negative σ , while the N-intermediates prefer formed on the electrode surface with positive σ . This disparity suggests a challenging coexistence of C- and N- intermediates on Cu(110), which may hinder coupling steps.
 - In Supplementary Fig. 35a,b,c, we further revised the discussions as: “Notably, all intermediates adsorbed Cu(100) surface exhibit a larger capacitance than the Cu(111) surface (Fig. 5f). This variation can be attributed to alterations in σ as the larger capacitances indicate more electrons on the electrode. As shown in Supplementary Fig. 35a-c, the σ follows the order Cu(100) > Cu(111), which is consistent with the order of EDL capacitance (Fig. 5f).” in Page 12 in the revised version.

re: Microkinetic model

In this revision, the authors have pointed out a very plausible origin for the large difference in urea TOFs between Cu(100) and Cu(111). The difference in the RDS coupled with the difference in surface coverages conceivably yields many orders of magnitude difference in the TOF. These are important findings.

However, these findings and interpretations could be elaborated more clearly in my opinion. In the rebuttal and manuscript revision, the facts presented are all reasonable but a bit poorly organized, failing to highlight important points. I’d suggest adding straight-to-the-point arguments, something like:

“On Cu(111), *CO---NH is the RDS; the surface is covered fully with *NH but with very little *CO.

On Cu(100), *NO₂---H is the RDS; the surface is covered fully with *NO₂.

Cu(111) struggles to get *CO while Cu(100) gets H⁺ from the electrolyte (assuming ER step).

(To support, the authors can then explain the computed X_i values of *CO and *N-species.)”

And finally, some broad concluding sentences like:

“The reason for large difference in urea TOFs between 100 and 111 is not activation barriers alone.

It is the different RDS and the availability of reactants (*NH/*CO for 111; *NO₂/H⁺ for 100)”.

But overall, I find the authors have adequately addressed my concerns in this microkinetic section.

Response: Thanks for the insightful and constructive comments. We appreciate the reviewer recognition of the importance of our findings in the microkinetic section and suggestions for improving the clarity and organization of our presentation.

To elaborate more clearly on our findings and interpretations, we reorganize the relevant sections to highlight the key points more effectively. The discussions of “To elucidate the huge differences in TOF on Cu(111) and Cu(100) surfaces during urea synthesis, the comprehensive degree of rate control (DRC) (Supplementary Tables 14,17) and coverage analysis (Supplementary Fig. 29) are conducted. A positive (negative) X_i value signifies that an increase in reaction rate (r) need to further stabilization (destabilization) of the corresponding surface state. On Cu(111), CO-NH coupling is the RDS (Supplementary Table 14); the surface is covered fully with *NH but with very little *CO (Supplementary Fig. 29). On Cu(100), NO₂-H hydrogenation is the RDS (Supplementary Table 17); the surface is covered fully with *NO₂ (Supplementary Fig. 29). Cu(111) struggles to get *CO, while Cu(100) gets H⁺ mainly from the electrolyte (E-R step). Thus, the urea synthesis reaction can proceed smoothly on Cu(100) surface, but is difficult on Cu(111) surface. Moreover, the *NH with $X_i = -1$ negatively impacts the r on Cu(111), while the surface is covered fully with *NH. On Cu(100), *NO₂ with $X_i = 1$ positively influences r and *NO₂ is the predominant surface-covering species during the urea synthesis on Cu(100). Therefore, the r for urea synthesis is further weaken on Cu(111) surface and enhanced on Cu(100) surface. Consequently, the reason for large difference in urea TOFs between Cu(100) and Cu(111) is not kinetic barriers alone. It need to be attributed to the different RDS and the availability of reactants (*NH/*CO for Cu(111), *NO₂/H⁺ for Cu(100)).” have been revised in Page 9 in the revised version.

Reviewer #2 (Remarks to the Author):

Response: We would like to express our gratitude to Reviewer #2 for clarifying the co-reviewing process and for the contribution to the peer-review of our manuscript. We appreciate the time and effort both Reviewer #2 and the co-reviewer have invested in reviewing our work. We have further revised this manuscript as suggested by the reviewers.

Reviewer #3 (Remarks to the Author):

The authors have satisfactorily responded to my comments; I recommend this work for publication

Response: We greatly appreciate the reviewer's approval of our manuscript for publication, and we would like to express our sincere gratitude to the reviewer for the time and efforts dedicated to thoroughly reviewing our work and providing constructive and expert comments that have contributed to greatly improving the quality of this manuscript.

REVIEWERS' COMMENTS

Reviewer #1 (Remarks to the Author):

The authors have addressed my comments

Reviewer #2 (Remarks to the Author):
